# Evolution of cell size control is canalized towards adders or sizers by cell cycle structure and selective pressures

Felix Proulx-Giraldeau[1], Jan M Skotheim[2,3], Paul François[1]*

[1]Department of Physics, McGill University, Montreal, Canada; [2]Department of Biology, Stanford University, Stanford, United States; [3]Chan Zuckerberg Biohub, San Francisco, San Francisco, United States

**Abstract** Cell size is controlled to be within a specific range to support physiological function. To control their size, cells use diverse mechanisms ranging from 'sizers', in which differences in cell size are compensated for in a single cell division cycle, to 'adders', in which a constant amount of cell growth occurs in each cell cycle. This diversity raises the question why a particular cell would implement one rather than another mechanism? To address this question, we performed a series of simulations evolving cell size control networks. The size control mechanism that evolved was influenced by both cell cycle structure and specific selection pressures. Moreover, evolved networks recapitulated known size control properties of naturally occurring networks. If the mechanism is based on a G1 size control and an S/G2/M timer, as found for budding yeast and some human cells, adders likely evolve. But, if the G1 phase is significantly longer than the S/G2/M phase, as is often the case in mammalian cells in vivo, sizers become more likely. Sizers also evolve when the cell cycle structure is inverted so that G1 is a timer, while S/G2/M performs size control, as is the case for the fission yeast *S. pombe*. For some size control networks, cell size consistently decreases in each cycle until a burst of cell cycle inhibitor drives an extended G1 phase much like the cell division cycle of the green algae *Chlamydomonas*. That these size control networks evolved such self-organized criticality shows how the evolution of complex systems can drive the emergence of critical processes.

*For correspondence:
paul.francois2@mcgill.ca

**Competing interest:** The authors declare that no competing interests exist.

## Editor's evaluation

This paper develops evolutionary simulations to identify the type of molecular networks that can give rise to size control. The authors propose an evolutionary framework to find which factors select for particular mechanisms in cell size control. They show that the evolution of a specific cell size control mechanism is dependent on the cell cycle structure.

## Introduction

Cell size is fundamental to cell physiology and function because it sets the scale of subcellular compartments, cellular biosynthetic capacity, metabolism, mechanical properties, surface-to-volume ratios, and molecular transport (*Chan and Marshall, 2010*; *Ginzberg et al., 2015*; *Neurohr et al., 2019*; *Zatulovskiy and Skotheim, 2020*). While different types of cells vary enormously in size to perform their functions, cells within a particular type are generally uniform in size indicating that cell growth may be accurately coupled to division and differentiation processes. On a phenomenological level, there are many commonalities in how cells regulate their size even though the molecules controlling cell division vary across the tree of life with the most striking differences separating eukaryotes and

bacteria. This extreme molecular diversity in the regulatory proteins raises the question as to what are the common features of the control systems that evolved to implement size control.

Most generally, cell size control can be viewed as a return map where the division size is a function of the cell size at birth. The examination of proliferating cells in laboratory conditions has revealed a variety of size control phenomena that can be characterized quantitatively by plotting the size of a cell at birth against the amount of mass added before it divides (*Amir, 2014*; *Facchetti et al., 2017*; *Jun and Taheri-Araghi, 2015*). A 'sizer' has a slope of –1 so that all variation in cell mass at birth is compensated for in one cell cycle, whereas an 'adder' has a slope of 0 so that each cell adds the same amount of mass during the cell cycle regardless of initial size. In the case of an adder, control is weaker so that multiple cell cycles are required for a particularly large or small cell to return to the average cell size. Importantly, the slope relating size at birth with the amount of growth in the cell cycle is a metric that quantifies the amount of size control occurring in a particular condition.

Studies of cell size control have revealed a diverse set of phenomena. Fission yeast and mouse epidermal stem cells exhibit 'sizers', and most bacteria, archaea, and cultured human cell lines exhibit behavior closer to adders (*Cadart et al., 2018*; *Eun et al., 2018*; *Jun et al., 2018*; *Sveiczer et al., 1996*; *Westfall and Levin, 2017*; *Willis and Huang, 2017*; *Xie and Skotheim, 2020*). Thus, while diverse size control behaviors have been observed, adders have been observed more often than sizers. This raises the question of why adders are more frequently observed if sizers, by definition, are more effective at controlling cell size (*Barber et al., 2017*; *Willis et al., 2020*).

To address the question of why adders are the most often observed form of cell size control, we used evolutionary algorithms (*Holland, 1992*) to identify commonalities between networks evolved to control cell size. Evolutionary algorithms are a class of machine learning techniques aiming at mimicking evolutionary processes (*Crombach, 2021*; *François, 2014*; *François and Hakim, 2004*; *Xiong et al., 2019*). Because of the nature of evolution, results of evolutionary computations are often more efficient and more creative than expected (*Lehman et al., 2020*). Furthermore, solutions found by evolutionary algorithms are constrained by their evolutionary paths followed and present similar characteristics to biologically evolved systems (*Schaerli et al., 2018*). Cell size is regulated through the cell cycle control network that governs transitions from one phase of the cell cycle to the next. The division cycle can be broken up into distinct phases that are characterized by different molecular activities (*Morgan, 2007*). While it is typically considered that there are 4 phases of the cell cycle (G1, S, G2, and M), we here consider a two phase model based on a G1 phase and a composite S/G2/M phase. This is because size control in general has been associated with either the G1/S transition or mitosis at the end of the cell cycle.

We start with a simple model of the cell cycle with a timer for an S/G2/M phase (*Figure 1*) that we evolve to optimize homeostatic cell size control (*Figure 1B*). We discovered that different control mechanisms could perform cell size control based on protein quantity or concentration. Simulations in which size control takes place in G1 phase converge toward an adder mechanism for the entire cell cycle and identified an active quantity sensing mechanism similar to dilution-based mechanisms previously identified experimentally (*Chen et al., 2020*; *Qu et al., 2019*; *Schmoller et al., 2015*). The relative durations of G1 and S/G2/M were important in determining size control properties. A relatively shorter S/G2/M phase favors sizer mechanisms, while longer S/G2/M phases favor adders. Moreover, inverting the model so that cell size controls S/G2/M and G1 is a timer, like in fission yeast, results in more sizer-like control. Thus, we anticipate adders arise when cell size is controlled at a point intermediate in the cell cycle, like the G1/S transition, while sizers will appear when cell size regulates a point later in the cell cycle, as is the case when G1 is proportionally longer, or control takes place at the transition to mitosis. We finally identify a self-organized mechanism based on fluctuation sensing where size control occurs on average over multiple cycles. While there is no one-to-one correspondence between a specific size control mechanism and a given evolutionary pressure, our work identifies clear evolutionary principles that shed light on the diverse cell size control phenomena previously observed experimentally.

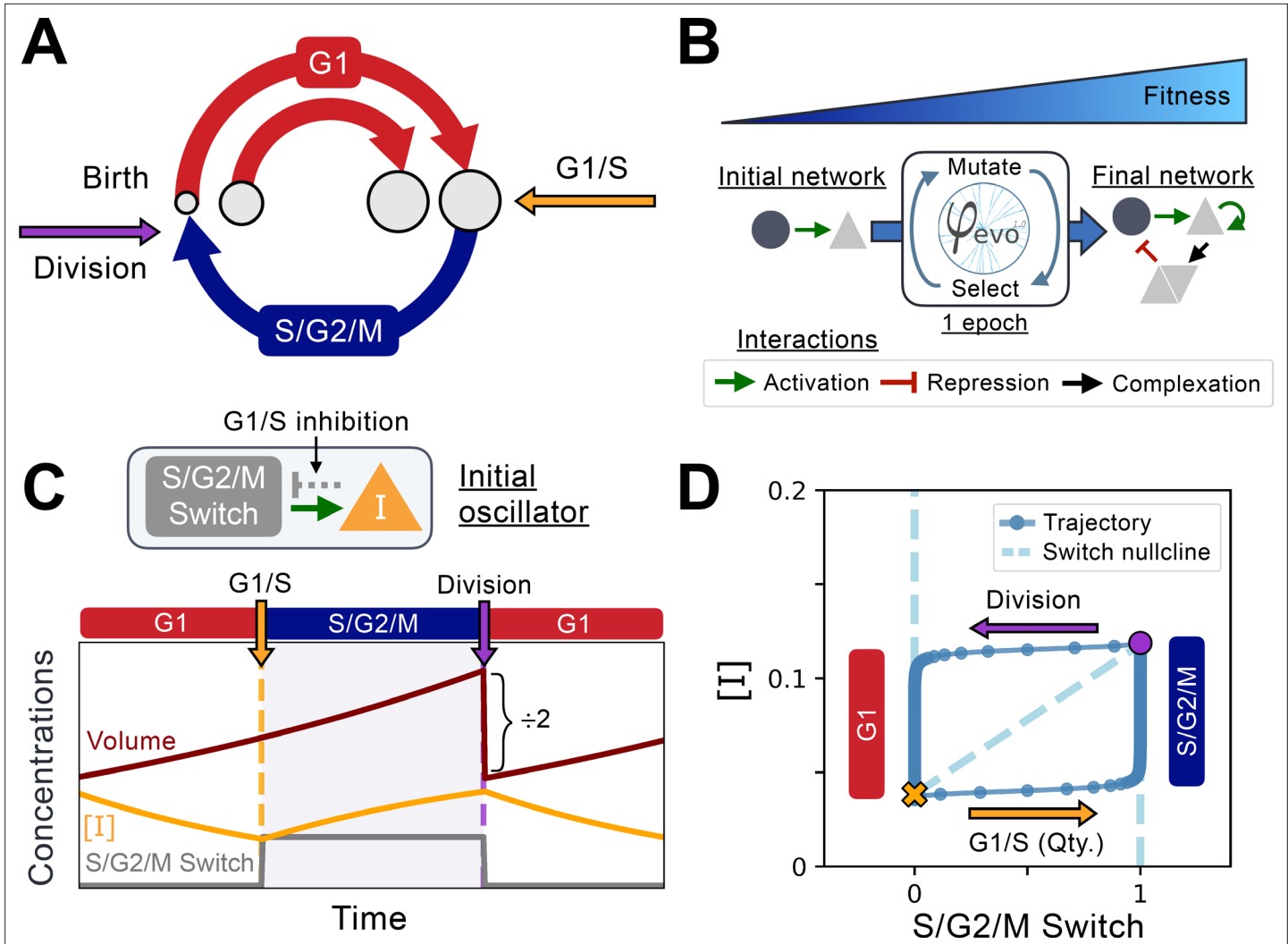

**Figure 1.** Implementation of the cell cycle seed network and evolution algorithm. (**A**) Schematic representation of the coupling between cell size and cell cycle progression. The transition between G1 (red) and S phases of the cell cycle at the G1/S transition (orange) can evolve to depend on cell size, while the duration of the S/G2/M (blue) phase is independent of size. Cell division takes place instantaneously following mitosis (purple). (**B**) Schematic representation of the φ-evo algorithm implementation. The network generating tool takes an initial network topology as its starting point for evolution as well as a user-defined fitness function. φ-evo then goes through successive epochs of mutation and selection to extract a final optimized network. At each selection step, the fittest half of the networks are retained and duplicated for evolution in the subsequent epoch. Interactions permitted to be mutated by φ-evo include transcriptional activation (green arrow), transcriptional repression (red arrow) and protein-protein interactions responsible for complex formation (black arrow). (**C**) Schematic of our seed network topology implementing a simplified relaxation oscillator. Cell cycle state (G1 or S/G2/M) is encoded via a binary switch called 'S/G2/M Switch' that is 0 in G1 and 1 in S/G2/M. Transition between G1 and S is controlled by the quantity of an inhibitor of the G1/S transition that we call $I$. The lower the quantity $I$, the higher the chance of progression through the G1/S transition. This interaction is represented as the grey arrow in the network topology and cannot be mutated by φ-evo. After progressing through G1/S, cells enter S/G2/M which we model as a pure timer of fixed duration with some uniform noise. Cell volume grows exponentially and is divided symmetrically following mitosis. We then follow one of the daughter cells and disregard the other one. (**D**) Phase-space representation of the initial relaxation oscillator. The X-coordinate shows the S/G2/M Switch variable, and the Y-coordinate shows the concentration of $[I]$. The oscillator runs counterclockwise with the left branch (x=0) corresponding to G1 and the right branch (x=1) corresponding to S/G2/M. The G1/S transition and division events are instantaneous in our simulations but are smoothly represented here for visualization purposes. From these transitions, we can extract the approximate shape of the $[I]$ nullcline that we plot under the oscillator with a dashed line. We note that the position of the G1/S transition in phase-space will vary as a function of the volume of the cell as it depends on the *quantity* of $I$ rather than its *concentration* $[I]$.

## Results

### Initial cell-cycle model

In general, there are two classes of mechanisms that cells use to control their size that can be separated in terms of whether cell division or cell growth per se is regulated by cell size. Note that in this work, we will use mass, size, and volume interchangeably. In the first class, it is crucial that the growth rate per unit mass of a cell depends on cell size so that cells that are significantly larger than the optimum cell size grow slower (*Cadart et al., 2019*; *Ginzberg et al., 2018*; *Miettinen and Björklund, 2016*; *Nordholt et al., 2020*; *Tzur et al., 2009*). Such slower growing cells are then outcompeted by cells closer to the optimum size even when divisions occur purely by chance (*Conlon and Raff, 2003*). While size-dependent growth mechanisms exist and do support size homeostasis, such mechanisms rely on inefficient growth in all the cells away from the optimum size (*Ginzberg et al., 2018*; *Miettinen and Björklund, 2016*; *Nordholt et al., 2020*). To avoid such inefficient growth, many types of cells use active size control mechanisms to accelerate progression through the cell cycle in larger cells (*Zatulovskiy and Skotheim, 2020*). In our simulations, we keep cell growth rates constant over a physiological range of cell sizes. This allows us to focus on the common features of the molecular networks in which increasing cell size drives changes in molecular activities to trigger cell division. We assume that cell volume $V$ grows at a rate $\lambda(V) \times V$, so that growth is exponential when $\lambda$ is a constant. Volume is divided by 2 at each division after which we follow one of the two daughter cells. The growth rate sets the time scale for the system dynamics as it defines the doubling time $\tau = \ln(2)/\lambda$. Any interdivision time shorter than $\tau$ will see the cell volume shrink at the next generation while any interdivision time larger than $\tau$ will see the cell volume grow. We also use the chemistry square bracket convention such that any protein X's *concentration* is denoted by $[X]$. Correspondingly, its *quantity* is denoted by $X$ only and is defined as $X = [X] \times V$.

We initialize our network evolution simulations with a very simplified model of the cell-cycle (*Figure 1A*). We model two independent phases of a symmetrically dividing cell, G1 and S/G2/M, separated by a commitment point at the end of G1 and division at the end of S/G2/M (*Figure 1A*, *Chandler-Brown et al., 2017*). We encode this cell cycle state information via a binary switch variable we call 'S/G2/M Switch' that is 0 in G1 and 1 in S/G2/M. In all simulations, we follow an inhibitor model (*Heldt et al., 2018*; *Schmoller et al., 2015*; *Zatulovskiy et al., 2020*) and assume that the probability of passing the G1/S transition is controlled by the *quantity* of a transcription regulator $I$. One way the quantity rather than the concentration of a molecule could be sensed is through its titration against a fixed cellular quantity such as the genome, which is part of a general class of titration-based cell size sensing mechanisms (*Amodeo et al., 2015*; *Heldt et al., 2018*; *Si et al., 2019*; *Wang et al., 2009*). A lower quantity of this inhibitor $I$ means a higher probability of a G1/S transition at the current time step of the simulation (*Appendix 1—figure 2*). Like all other proteins, the quantity $I$ is produced with a rate proportional to volume, degraded at a constant rate, diluted by cell growth, and equally partitioned between mother and daughter cells at division (see Materials and Methods). We found that due to the volume scaling assumption, $[I]$'s concentration alone was largely independent of volume and could not trigger a size-dependent G1/S transition, which is why we opted for the quantity of $I$ instead (*Appendix 1—figure 3*). Upon passing the G1/S transition, we assume cells are committed to division and there is a fixed time delay before they divide thus modeling S/G2/M as a timer (*Chandler-Brown et al., 2017*; *Doncic et al., 2015*). We initially fixed the timer duration to be roughly equal to 50% of the doubling time $\tau$ with some uniform noise such that G1 and S/G2/M durations would be the same at equilibrium. Regulation of the quantity $I$ during the cell cycle thus controls the precise timing of the G1/S transition, but it is not always perfect since the transition is probabilistic. This, along with the noise in S/G2/M timer duration, creates natural cell to cell variability in volume that needs to be compensated for by the evolved mechanism. We note that we initialized most of our simulations with one added interaction in which production of $[I]$ is activated by the S/G2/M Switch variable to reset its concentration to a higher level before the next generation. We initially ran simulations without this specific interaction but found that it systematically appears in the initial stages of evolution simulations. We therefore included it in the initial network to speed up our simulations. We refer the reader to the Appendix 1 for more details. The models used in this study are publicly available (*Proulx-Giraldeau and François, 2022*).

Typical dynamics of this simple cell-cycle model are represented in *Figure 1*. These dynamics are similar to models of cell cycles based on relaxation oscillators (*Cross, 2003*; *Tsai et al., 2008*). The left

and right slow branches correspond to G1 and S/G2/M, respectively, and the fast horizontal branches represent G1/S and division. An intermediate fictitious nullcline is shown as a line that connects the average concentration $[I]$ at G1/S and at division. Starting with cell birth, the system goes down the left-G1 branch because of degradation, then jumps to the right-S/G2/M branch below the threshold for the G1/S transition, stays there while moving up due to production by the S/G2/M Switch, until it jumps back to the left branch at the end of the timer phase. We note that there is no explicit volume control in this initial model since the only control comes from the quantity of $I$ which does not initially depend in any way on the volume. This initial quantity sensing oscillator does not perform size control and instead results in unstable growth where size deviations are amplified at each generation instead of being corrected (*Appendix 1—figure 3*) as had been previously described for a size scaling inhibitor dilution model (*Barber et al., 2017*; *Willis et al., 2020*). Thus, the network needs to evolve some other interactions and/or parameters to go beyond a simple G1 inhibitor driven by production in S/G2/M to create a viable cell lineage.

## Evolution of quantity-based size control mechanisms

To examine how networks controlling cell size could evolve, we ran evolutionary simulations that optimize both the number of divisions $N_{Div}$ and the coefficient of variation of the size distribution at birth $CV_{Birth}$ (see Materials and Methods for algorithm details). *Figure 2A* illustrates the behavior and results of a typical evolutionary run, with axes defined by both fitnesses used. Simulations successfully evolving size control mechanisms typically follow the same pattern. Networks initially cluster in two regions: region [ii] where cells have low $CV_{Birth}$ but grow too small and die after a few divisions, and region [i] where cells grow too big and reach our cut-off for fitness 2 (y-axis). Notice that our Pareto evolutionary algorithm maximizes network diversity, so that those two clusters are at first maintained during evolution (rank 1 Pareto networks *Warmflash et al., 2012*). As the number of epochs increases, networks in cluster [ii] have more and more divisions, but still grow too big, so that those cells are therefore penalized (see details in Appendix 1). At some point in evolution (around epoch 700 for this particular simulation), some weak control mechanism suddenly evolves, preventing cells from becoming too big without imposing a tight control on the average volume (see also Figure 5 for an explicit example of how this is done). Thus, fitness 2 collapses and the number of divisions is optimized simultaneously. Cells later optimize the control to give a lower $CV_{Birth}$ . The optimal networks, at the right most end of this line, both maximize $N_{Div}$ and minimize $CV_{Birth}$ .

Evolution simulations are in part reproducible and most often lead to similar network topologies. The evolution trajectory leading to Model A1 is a typical example (*Figure 2B*, see also variations of this network in Model A2 in *Figure 2H* and models A3-6 in *Appendix 1—figures 8–11*). The minimal network common to all those models is very simple. One gene, $R$, is added to the seed network and is both repressed and titrated by $I$ forming the network motif known as a Mixed Feedback Loop (*François and Hakim, 2005*). Size control can then be understood intuitively as follows.$[I]$ represses $[R]$, which is thus only produced in the narrow window of the cycle when $[I]$ is low, *i.e.*, when the cell is close to the G1/S transition and in early S/G2/M. But, since the *quantity* $I = [I] \times V$ is fixed at G1/S by design, the *concentration* $[I]$ is inversely proportional to the volume of the cell at the G1/S transition ($V_{G1/S}$) as shown in our simulations (*Figure 2C*). Because of this, the $[I]$ dependent synthesis rate of $[R]$ and therefore its subsequent concentration are (linear) functions of the volume of $V_{G1/S}$ (*Figure 2C*), allowing for the cell to keep a memory of its volume at G1/S via the $[R]$ variable (this holds even once $[R]$ is constantly degraded for the remainder of the cycle). This has two effects. First, during S/G2/M, $I$ synthesis rate is proportional to volume by hypothesis (and thus to $V_{G1/S}$), and $I$ is titrated by $[R]$, also proportional to $V_{G1/S}$. Both effects even out so that cells are born with a fixed *quantity* of inhibitor $I$ that is independent of volume (*Figure 2D*). Second, after division, production of $I$ is 0 by hypothesis, but $I$ still is titrated in G1 by the remaining $[R]$ (still proportional to $V_{G1/S}$). Because $I$ quantity at the beginning of G1 is size independent, this ensures that daughter cells reach the $I$ *quantity* threshold of G1/S earlier if they were born larger, thus ensuring size control. To confirm this, we examine the change in quantity of $I$ as a function of time spent in G1 and of $V_{G1/S}$, and we see the slope of these two quantities is volume-dependent (*Figure 2E*). We notice that this evolved size mechanism likely is the simplest possible allowed by our formalism: on the one hand, it entirely captures the volume dependency in a single variable $[R]$, and on the other hand, it ensures proper scaling of $I$ both at birth and at G1/S for size control with the help of a single titration. Notice that such simple control

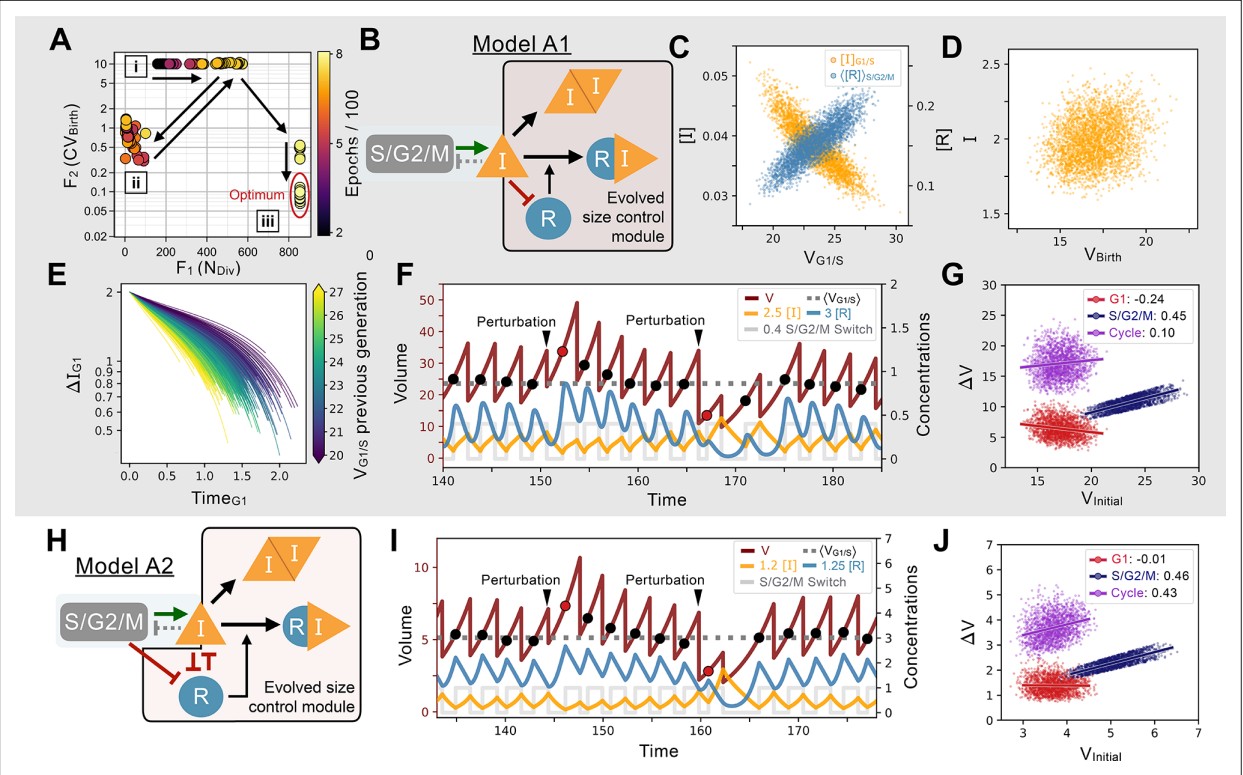

**Figure 2.** Evolution of feedback-based size control. (**A**) Typical 2D fitness trajectory for an evolutionary run. Individual networks are dots color coded by their epoch within the evolutionary trajectory. Fitness function of the number of divisions of a cell lineage during a time interval of fixed length ($F_1 \left(N_{Div}\right)$, X-coordinate) and fitness function of the coefficient of variation of the volume distribution at birth ($F_2 \left(CV_{Birth}\right)$, Y-coordinate). Optimal model behavior is located in the bottom right corner of the figure where networks produce cell lineages with many offspring and strong size control. First, there are several epochs without any size control; networks cluster in two regions of the Pareto front corresponding to volume going to the maximum allowed value (cluster [i]) or to the minimum value (cluster [ii]). Both cases are highly penalized in their fitness score. Evolution goes back and forth between the [i] and [ii] clusters with a slow increase in the number of divisions (X-coordinate). Eventually, some volume control evolves and networks transition in the [iii] cluster where their $CV_{Birth}$ is slowly optimized further until the end of the run. (**B**) Core network topology of the evolved Model A1 network that employs a feedback-based mechanism described in detail in panels C-F. (**C**) Concentration of $[I]$ at the G1/S transition (Y-coordinate, left axis, orange) and average concentration of the repressor protein $[R]$ in S/G2/M (Y-coordinate, right axis, light blue) as a function of the volume of the cell at G1/S (X-coordinate), that is, the beginning of S phase. We see here that $[R]$ acts as a direct size sensor of the volume at G1/S. (**D**) *Quantity* of inhibitor $I$ at birth as a function of volume of the cell at birth, which is independent of size due to titration by $[R]$ during S/G2/M. (**E**) Trajectories of quantity of inhibitor $I$ in G1 as a function of time. Trajectories are color coded as a function of the volume at G1/S during the previous generation's cell cycle. For visualization purposes, trajectories are offset vertically to all begin at the average *quantity* of $I$ at birth (t=0) shown to be on average independent of volume in panel D. Larger cell volumes lead to greater titration of $[I]$ in G1 by $[R]$. In turn, this ensures that G1 duration of the daughter cell cycle is shorter, which underpins the size control mechanism. (**F**) Characteristic dynamics of Model A1. Circles indicate volume at G1/S. Extrinsic perturbations are applied to the model by temporarily changing the division ratios which kicks the system out of equilibrium at the subsequent cycle such that $V_{NextG1/S} = \left(1 \pm 0.5\right) \langle V_{G1/S} \rangle$. The volume relaxation back to its homeostatic value takes ~2–3 generations, almost insensitive to the fact that the perturbation is applied towards higher or lower volumes. (**G**) Amount of volume added $\Delta V$ in G1 (red), S/G2/M (dark blue), and over the whole cycle (purple) as a function of their initial volume at the beginning of these phases, *that is*, birth for G1 and cycle, and G1/S for S/G2/M, with the slope of linear fits indicated in legend. Slope of 1 corresponds to a Timer, slope of –1 to a Sizer and slope of 0 to an Adder. (**H**) Network topology of Model A2, a second evolved network that is similar to Model A1 albeit with different kinetic parameters and 2 additional interactions (see text). (**I**) Characteristic dynamics of Model A2. Extrinsic perturbations are applied like in (F). The volume relaxation back to its homeostatic value takes ~3–4 generations when applied towards the higher volumes but only 1 generation when applied towards the lower volumes. (**J**) Amount of volume added for different periods of the cell cycle for Model A2.

also explains the sudden evolutionary "jump" of Pareto front around epoch 700 on *Figure 2A*, which corresponds to when the Mixed Feedback Loop motif first appears.

These evolved size control networks, while relying on *quantity* sensing, are conceptually similar to the budding yeast network relying on *concentration* sensing of the cell cycle inhibitor Whi5 since there is a constant quantity of $I$ present right after division (just like Whi5). In budding yeast, the Whi5 protein is passively diluted in G1 to increase the stochastic rate of progression through the G1/S

transition. The time spent in G1 depends on the initial concentration of Whi5 at birth, which scales as $\frac{1}{V}$, to promote a sizer mechanism. Here, the concentration of $[I]$ at birth scales as $\frac{1}{V}$, but so does the threshold concentration of $[I]$ regulating the G1/S transition. This is precisely why an active titration mechanism is required to obtain G1 size control in our setup. Such homeostatic control ensures cell size returns to its steady state distribution following an artificial perturbation as soon as a volume deviation is detected at G1/S (*Figure 2F*). When we plot the amount of volume added $\Delta V$ for different phases of the cell cycle as a function of the initial volume at the beginning of these phases, we find an approximate adder over the whole cycle that results from weak sizer in G1 followed by a timer in S/G2/M as has been found in budding yeast (*Chandler-Brown et al., 2017*; *Di Talia et al., 2007*; *Soifer et al., 2016*, *Figure 2G*).

While we chose one simple model to illustrate the control mechanism common to our set of evolved networks (Model A1 shown in *Figure 2B*), other evolved networks were more elaborate but illustrated a similar principle. For example, Model A2 contains extra interactions for the volume sensing gene $[R]$, where $[R]$ is repressed by the S/G2/M Switch (meaning its production is completely shut down in S/G2/M leading to sawtooth-like dynamics). Furthermore, $[R]$ represses the synthesis of $[I]$, adding another layer of repression to promote size control beyond the previously described titration by $[R]$ (*Figure 2H*). If we perturb cell size to examine the dynamics of the return to steady state and look at the added volume during the cell cycle, we again see overall a weak adder behavior similar to that found in Model A1 (*Figure 2I–J*). We give additional examples of similarly evolved networks in *Appendix 1—figures 8–11* where we can see the sensing and the feedback mechanism being implemented in slightly different ways. Yet, despite these mechanistic differences in feedback regulation the resulting function of the evolved networks were similar as indicated by their $CV_{Birth}$ (*Appendix 1—table 1*).

## Quantifying size control

To study the mechanisms implicated in cell size control, we modify the control at G1/S and introduce the control volume $V_C$. This control variable is independent from the biochemical network and is maintained fixed allowing us to disconnect the actual cell volume $V$ from the biochemical network and by forcing the G1/S transition to be triggered once $I_C = [I] \times V_C$ is low enough. We then numerically integrate the differential equations of the model and measure the period $T(V_C)$ of the simulated cell-cycle for this control volume. Use of the control volume allows us to break the size feedback system and distinguish its input, $V_C$, from its output, the induced cycle period T (*Angeli et al., 2004*). We compute $T(V_C)$ for Models A1 and A2 and compare their responses with the analytical curves of the archetypical timer, adder, and sizer (*Figure 3A and C*). Those curves intersect at the point where the induced period is exactly equal to the population doubling time ($\tau = \ln(2)/\lambda$), which defines the equilibrium volume achieved by our cell size control network corresponding to $\langle V_{G1/S} \rangle$. Examination of the size control in different cell cycle phases indicates the contributions of G1 and S/G2/M to the overall system behavior (*Figure 3B and D*). We note that we later examine statistics of ensembles of evolved models but that Models A1 and A2 are both typical examples of evolved feedback-based models.

Quantifying precisely how the cell cycle period depends on the control volume at G1/S allows us to see that the $T(V_C)$ for Model A1 overlaps with the theoretical adder curve over a broad range of volumes. In contrast, Model A2 behaves as a sizer for volumes smaller than the equilibrium volume. However, at higher volumes it behaves more like an adder/timer similar to Model A1. Model A2's equilibrium is thus 'tuned' by evolution to be in a regime corresponding to the minimum of the $\Delta V_{Cycle}$ curve extrapolated from the $T(V_C)$ as shown in *Figure 3D*, precisely when the system transits from a sizer at small volumes to an adder-like behavior at higher volumes. Similar behavior was observed experimentally in budding yeast (*Chandler-Brown et al., 2017*; *Delarue et al., 2017*). Thus, both Models A1 and A2 approximate an adder near the equilibrium size, but their behavior differs further from equilibrium where for smaller volumes, Model A1 is still an adder while Model A2 is a sizer.

## Modulating cell cycle structural constraints selects for adders or sizers

So far, our evolutionary algorithm selects networks that implement adders rather than sizers near the equilibrium size. This is surprising because sizers are in principle better than adders at controlling cell size and reducing the CV of the size distribution at birth, which is one of our fitness functions. That adders are more frequently observed in nature than sizers (*Cadart et al., 2018*; *Eun et al., 2018*; *Jun*

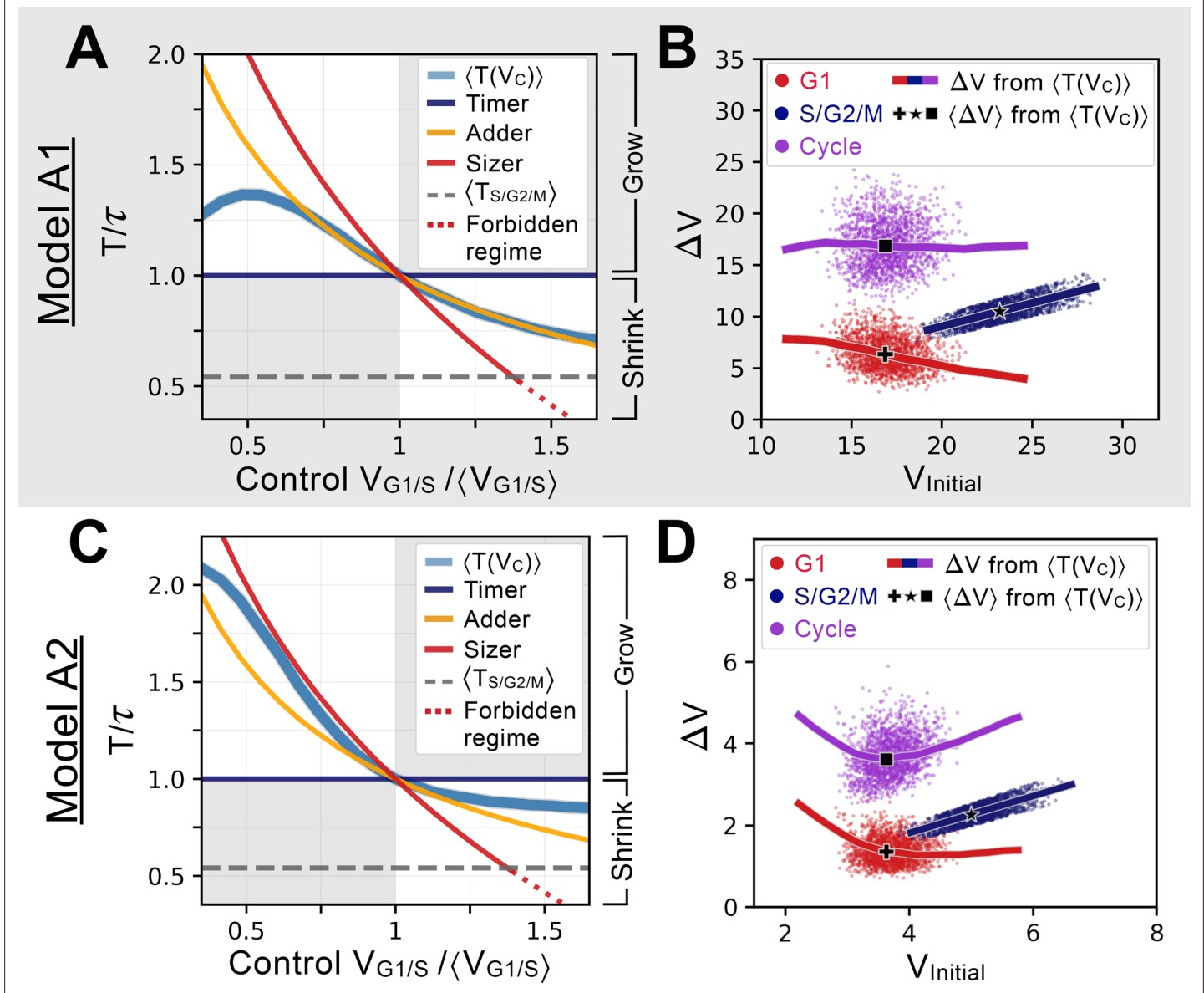

**Figure 3.** Characterizing and comparing evolved size control mechanisms. (**A**) Average period $T(V_C)$ of the oscillator of Model A1 as a function of the control volume $V_C$ at the G1/S transition. Period is normalized by $\tau$ the doubling time of the cell, and volume is rescaled by $\langle V_{G1/S} \rangle$, which corresponds to $T(\langle V_{G1/S} \rangle) = \tau$. Normalized periods larger than 1 indicate cell lineages that grow over time whereas normalized periods smaller than 1 indicate lineages that shrink over time. Periods for the sizer (red), adder (orange) and timer (dark blue) are shown for comparison. The S/G2/M timer period is incompressible and prevents a perfect sizer from existing in the large volume range as indicated by the red dotted line. Model A1 follows approximately the adder archetype over a large range of control volumes. (**B**) Added volumes $\Delta V$ for different phases of the cell cycles for simulations of Model A1. Individual dots correspond to different cell cycles for a simulation at steady-state. The full line corresponds to the extrapolation from the $T(V_C)$ curve shown in A for a restricted range of $V_C$ relevant to the scatter. The black cross, star and square indicate the average added volumes corresponding to when the system senses a volume corresponding to $\langle V_{G1/S} \rangle$ at the G1/S transition. We see that the model is predicted to follow an adder over a large range of volumes. (**C**) Average period $T(V_C)$ of the oscillator of Model A2, with similar conventions as for panel A. We note that the $T(V_C)$ curve of this model is closer to the sizer at lower volumes and closer to a weak adder/timer at higher volumes relative to $\langle V_{G1/S} \rangle$. (**D**) Added volumes $\Delta V$ for different phases of the cell cycles for simulations of Model A2, with similar conventions as for panel B. Here we see the predicted sizer behavior at lower volumes and the weak adder/timer behavior at higher volumes.

et al., 2018; *Westfall and Levin, 2017*; *Willis and Huang, 2017*; *Zatulovskiy and Skotheim, 2020*), is consistent with our evolution simulations, but adds to the mystery as to why this takes place.

To gain insight into the underlying reason for the prevalence of adders, we considered what might be exceptional in the cases where sizers occur. The best studied, and highly accurate sizer, is found

in the fission yeast *S. pombe* (*Fantes, 1977*; *Sveiczer et al., 1996*). In contrast to budding yeast and human cells, where the size control takes place largely in G1 phase, fission yeast exerts size control later in the cell cycle at the G2/M transition. That size control took place later in the cell cycle in fission yeast suggested that this structural feature of its cell cycle might be responsible for its stronger size control. To test this, we inverted our seed network so that the G1 phase was a timer, while cell size control could evolve in S/G2/M (see Materials and Methods). We compare these results to the evolutionary simulations starting with the 'control' seed network with G1 size control and an S/G2/M timer (*Figure 2*). We note that *S. pombe* growth rates have been reported to deviate from exponential (*Wood and Nurse, 2015*). While slower than exponential growth would aid cell size control, our analysis here is restricted to exponential growth.

To determine how the seed network structure influences the subsequent evolution of cell size control, we performed 120 independent evolutionary simulations for the two network structures initialized with the Model A1 topology and parameters (*Figure 4*). Sixty simulations were performed using Pareto optimization and another 60 simulations were performed using individual fitness optimization based on the number of cell divisions $N_{Div}$. For each simulation's most fit model after 500 epochs, we calculate the CV of the volume distribution at birth, $CV_{Birth}$, and the slope of the linear fit of the volume added in the entire cell cycle as a function of the cell size at birth, Slope $\Delta V_{Cycle}$ (we remind the reader that a slope of –1 corresponds to a sizer, 0 to an adder, and 1 to a timer). We chose to use 500 epochs in our simulations because in our previous experience this was sufficient for networks to evolve to be near the optimum, but not so much that they were forced to extensively explore the effects of neutral mutations near the optimum. Model A1's initial and Slope before evolution are indicated by the dashed black line. In the control experiment where G1 is a sizer and S/G2/M is a timer, most evolutionary simulations with two fitness functions (Pareto) yield models close to the adder regime with a low $CV_{Birth}$. However, when only the number of divisions ($N_{Div}$) is used as a fitness, the evolutionary simulations are closer to the sizer regime, albeit with a slightly higher $CV_{Birth}$ (*Figure 4A*, Welch's *t*-test p<10⁻⁴). When the cell cycle structure is inverted so that G1 is a timer and S/G2/M is a sizer like it is in the fission yeast *S. pombe*, we found that more sizer-like networks evolve than in the control experiment (*Figure 4B*, Welch's *t*-test on agglomerated data p=0.03). This shows that having a network structure like the fission yeast *S. pombe* promotes sizers, while having the size control portion of the cell cycle earlier, as in the budding yeast *S. cerevisiae*, promotes adders. Thus, performing simulations using cell cycle network structures of these two yeasts results in the evolution of size control mechanisms that reflect those that are naturally occurring.

The general notion that having cell size control in G1 results in adder-like mechanisms, while control later in the cell cycle results in more sizer-like mechanisms fits most observations of human cell lines grown in culture, budding yeast, and fission yeast. However, a recent study examining mouse epidermal stem cells growing and dividing in the skin found both a strong sizer and that this sizer was largely due to the size-dependent regulation of G1 (*Mesa et al., 2018*; *Xie and Skotheim, 2020*). This raised the question as to how a network performing size control in G1 could result in a sizer for the entire cell cycle. One important difference between mammalian cells grown in culture and the mouse epidermal stem cells growing in an animal (*in vivo*) is the change in the relative durations of the G1 and S/G2/M phases of the cell cycle. While the S/G2/M phase of the cell cycle is similar in duration in cultured cells and the epidermal stem cells *in vivo* at ~12 hr, the G1 phase extends ~fivefold from ~10 hr in culture to ~50 hr *in vivo* (*Cadart et al., 2018*; *Xie and Skotheim, 2020*). This suggests the hypothesis that the overall size control behavior can be dominated by the relatively longer cell cycle phase, as is likely the case for the G1 phase of epidermal stem cells. To test this hypothesis, we performed evolutionary simulations with size control in G1 and a timer in S/G2/M but where we changed the duration of the S/G2/M timer phases of the cell cycle to be significantly shorter or longer than the G1 phase at equilibrium. When a timer in S/G2/M is relatively shorter compared to G1, we generally see more sizer-like behavior can evolve (*Figure 4C*, Welch's *t*-test on agglomerated data p=4 x 10⁻³), while when it is relatively longer, we see more adder-like or even timer-like behavior (*Figure 4D*, Welch's *t*-test on agglomerated data p<10⁻⁴).

We next considered the effect of changing the amount of noise in the timer phase of the cell cycle. To do this, we examined the evolution of networks performing size control in G1 and where the S/G2/M phase with an increasing amount of noise. Increasing the noise in the timer progressively reduced the amount of size control done by the network (*Appendix 1—figure 5*). This is likely

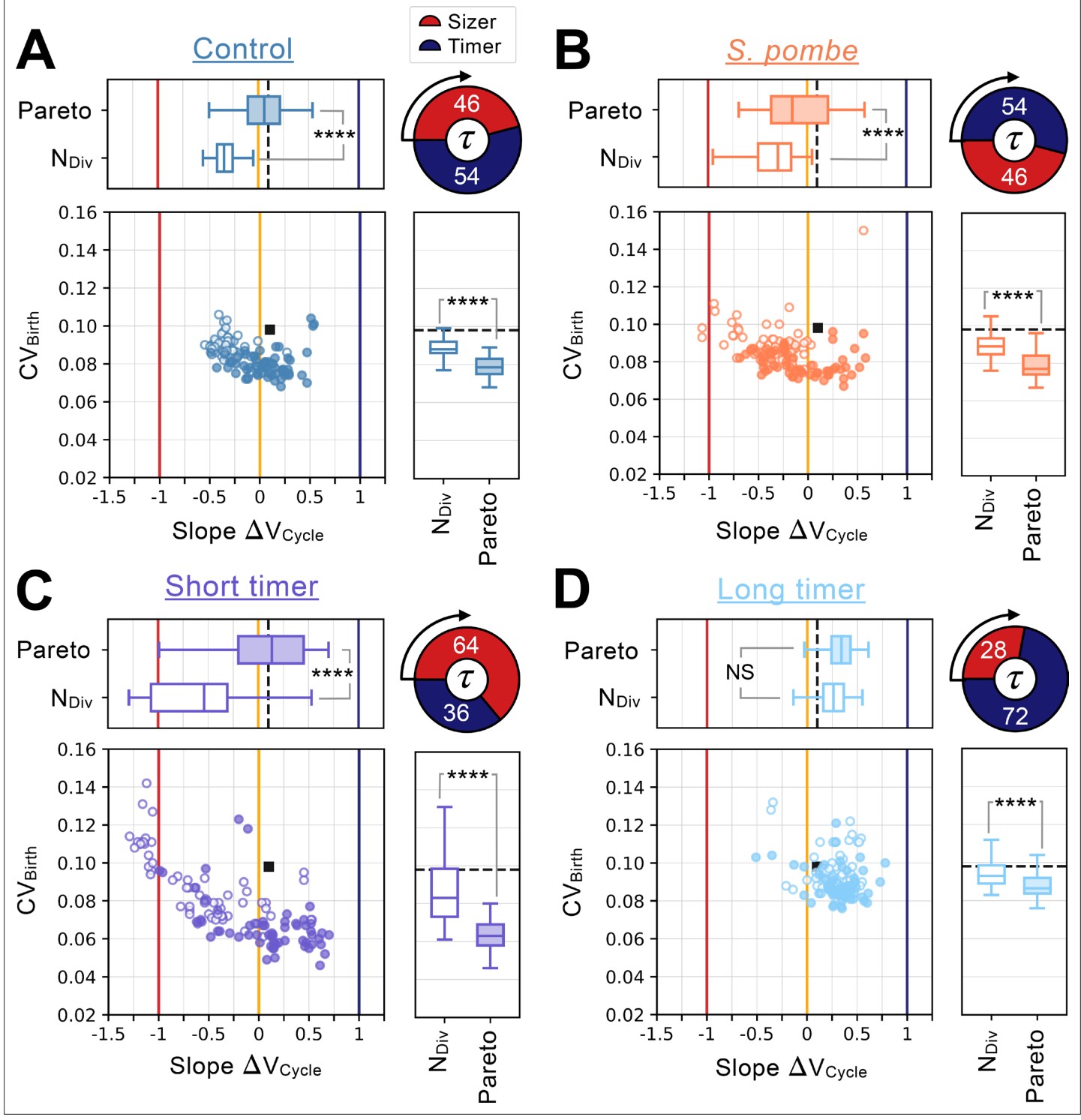

**Figure 4.** Distinct network constraints and selection pressures bias size control evolution towards adders or sizers. Summary statistics for evolutionary simulations each having 500 epochs. Model A1 shown in *Figure 2A-G* was used as the initial seed network. 60 simulations were performed using Pareto optimization of the number of divisions ($N_{Div}$) and the CV of cell size at birth ($CV_{Birth}$), are labeled *Pareto* and are shown in full colors. 60 more simulations were performed using only the number of divisions as the fitness function, are labeled $N_{Div}$ and are shown in colored outlines only. Scatter plots show the coefficient of variation of the size distribution at birth ($CV_{Birth}$, Y-coordinate) as a function of the fitted added volume slope over the whole cycle as a function of volume at birth (Slope $\Delta V_{Cycle}$, X-coordinate) for the most fit models evolved during each of the 120 independent simulations. Horizontal box plots above the scatter plots display the distributions of the added volume slopes for the *Pareto* and $N_{Div}$ simulations. Timer (dark blue), adder (orange) and sizer (red) slopes are shown respectively at 1, 0, and –1 for comparison. Vertical box plots on the right of the scatter plots

*Figure 4 continued on next page*

*Figure 4 continued*

show the distributions of $CV_{Birth}$ for the *Pareto* and $N_{Div}$ simulations. Asterisks represent p-values for the Welch's t-Test between the distributions. For reference, *NS* indicates $p > 0.05$, * indicates $p < 0.05$, ** indicates $p < 10^{-2}$, *** indicates $p < 10^{-3}$ and **** indicates $p < 10^{-4}$. The values of $CV_{Birth}$ and Slope $\Delta V_{Cycle}$ for the initial seed Model A1 are shown as a black square in the scatter plot or as a dashed black line in the box plots. Each panel explores different cell cycle structures which are summarized by the pie charts. Cycles begin on the left of the pie charts and rotate clockwise, indicating the order of the sizer (red) and timer (dark blue) phases. The labels indicate each phase's duration at equilibrium as a percentage of the doubling time $\tau$. (**A**) Identical evolutionary parameters as for Model A1 evolution shown in *Figure 2A–G*. G1 performs size control and has a duration $0.46\tau$ at equilibrium and S/G2/M is a timer of duration $0.54\tau$. (**B**) Evolution results for a cell cycle structure where the sizer and the timer phases of the cell cycle are inverted akin to *S. pombe*. G1 is a timer of duration $0.54\tau$ and S/G2/M performs size control and has duration $0.46\tau$ at equilibrium. (**C**) Evolution results for a G1 size control of average duration $0.64\tau$ at equilibrium where S/G2/M is a timer of duration $0.36\tau$. (**D**) Evolution results for a G1 size control of average duration $0.28\tau$ at equilibrium where S/G2/M is a timer of duration $0.72\tau$.

because the fixed duration of S/G2/M allows the system to accurately reset protein concentrations for the subsequent cell cycle to promote accurate G1 control (*Willis et al., 2020*). We also examined the effects of adding noise to the cellular growth rate and to volume partitioning at division and found similar results (*Appendix 1—figures 6–7*).

Taken together, our simulations show how the structural features of the cell cycle are important for determining what type of size control ultimately evolves. G1 control is more conducive to the evolution of adders, while S/G2/M control is more conducive to sizers. Moreover, size control can be dominated by the cell cycle phase of longest duration and is modulated by the specific selection criteria.

## A two-step evolutionary pathway for cell size control

From our series of evolution simulations, we found a somewhat paradoxical inverse correlation between the added volume slope quantifying the degree of size control (sizer vs. adder) and the $CV_{Birth}$ (*Figure 4B–C*). This was surprising because it means that there is a broader distribution of volumes in the sizer regime where control should be more effective in theory. To better understand this inverted correlation between Slope $\Delta V_{Cycle}$ and $CV_{Birth}$, we revisited our evolutionary simulations to examine the evolutionary pathways through which the networks progressed through the simulated epochs.

A typical evolutionary pathway for a *Pareto* simulation of the *S. pombe*-like network structure presented in *Figure 4B* is shown in detail in *Figure 5*. Here, Model A1 topology is conserved throughout evolution although individual parameter values change. In the early stages of the evolution (epoch 650), we typically see dynamics where small cell size triggers an overshoot to a large cell size, which is then reduced through a series of rapid divisions (*Figure 5A*). Because of this small-size-triggered overshoot, the system behaves more like a sizer when the average behavior is analyzed (*Figure 5B*). However, the high degree of variability in the cell cycles also leads to a broad distribution of volumes (*Figure 5C*). The variability in volume is attenuated in later epochs where there are fewer and smaller volume overshoots triggered by small cell size (*Figure 5D–I*). Since small cell size no longer triggers a dramatic amount of cell growth, the slope of $\Delta V_{Cycle}$ is increased and the system converges towards a weak adder in which the distribution of volume at birth is more Gaussian and the $CV_{Birth}$ is lower (*Figure 5H–I*). Taken together, these analyses suggest a two-step evolutionary pathway, consistent with the evolutionary dynamics first seen in *Figure 2A*. First, a strong but imprecise sizer mechanism evolves where, because of noise in the system, small variations in volume lead to a dramatic overcorrection and overshoot of the target volume. The variability in volume produced by this overshoot is then reduced by attenuating the strength of the size control response. Indeed, the overall weaker size control allows the system to respond more mildly to size deviations, thus yielding a lower $CV_{Birth}$ overall which we select for. Thus, selecting for a smaller $CV_{Birth}$, *i.e.* better size control, can end up selecting for adders rather than sizers. This paradoxical result is consistent with the fact that when we select only for the number of cell divisions ($N_{Div}$), one sees that more sizer-like behavior can evolve (*Figure 4A–C*). The typical behavior before optimization of $CV_{Birth}$ is a strong sizer as illustrated in *Figure 5A*.

## Fluctuation sensing and the evolution of self-organized criticality

One of the main features of smaller cells is that they have fewer proteins and mRNA. If some aspects of protein synthesis and degradation are subject to Poisson fluctuations, we expect such fluctuations

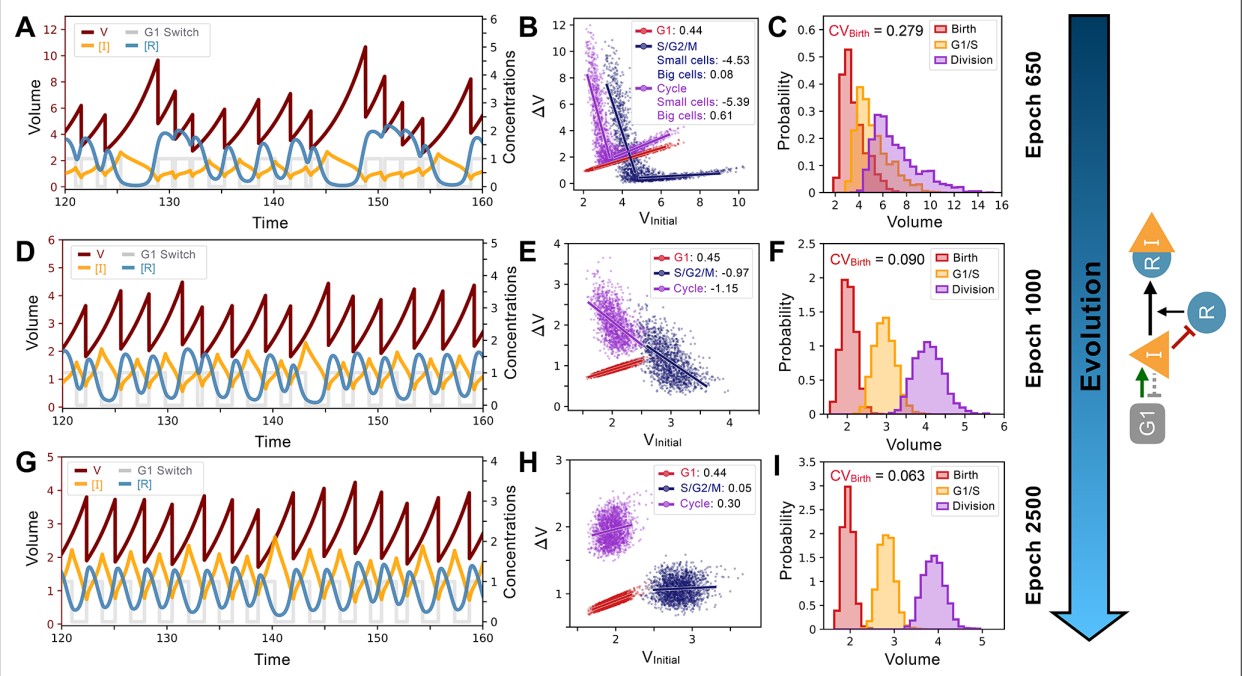

**Figure 5.** System and evolutionary dynamics of cell size control networks. Snapshots of an evolutionary simulation of 2500 epochs initialized with the Model A1 network topology along with an *S. pombe*-like cell cycle structure with a timer in G1 followed by a sizer in S/G2/M (see *Figure 4B*). Pareto fitness optimization was performed using $N_{Div}$ and $CV_{Birth}$ as fitness functions. Rows indicate simulation results for the fittest networks from evolutionary epochs 650 (Panels A-C), 1000 (Panels D-F), and 2500 (Panels G-I). Network topology remains the same throughout the evolutionary simulation and is shown on the right. Evolutionary dynamics continually reduce the selected for $CV_{Birth}$ and proceed through a noisy sizer to a less noisy adder. (**A**) Typical dynamics of the most fit model from epoch 650. (**B**) Added volumes $\Delta V$ for different phases of the cell cycle for the most fit model from epoch 650. Fitted slopes are indicated in the legend. Fits for the S/G2/M and Cycle added volumes were split in two separate the size control for small and large cells. (**C**) Size distributions at birth (red), G1/S (orange), and division (purple) for the most fit model from epoch 650. The coefficient of variation of the volume distribution at birth $CV_{Birth} = 0.279$. (**D**) Typical dynamics of the most fit model from epoch 1000. (**E**) Added volumes $\Delta V$ for different phases of the cell cycle for the most fit model from epoch 1000. Fitted slopes are indicated in the legend. (**F**) Size distributions at birth (red), G1/S (orange), and division (purple) for the most fit model from epoch 1000. The coefficient of variation of the volume distribution at birth $CV_{Birth} = 0.090$. (**G**) Typical dynamics of the most fit model from epoch 2500. (**H**) Added volumes $\Delta V$ for different phases of the cell cycle for the most fit model from epoch 2500. Fitted slopes are indicated in the legend. (**I**) Size distributions at birth (red), G1/S (orange), and division (purple) for the most fit model from epoch 2500. We see here that the sizer behavior from epoch 1000 was abandoned for a weaker adder overall yielding lower $CV_{Birth} = 0.063$.

to produce larger concentration fluctuations in smaller cells. For example, let us assume that the balance of synthesis and degradation of a generic protein results into a Poisson distribution with parameter $\frac{\rho V}{\delta}$, where $\rho$ is the synthesis rate in number of proteins per unit of time for a reference volume of 1, $V$ is the volume, and $\delta$ is the degradation rate. The average concentration of this protein in an exponentially growing cell will be $\frac{\rho}{\delta}$, which is independent of the cell volume $V$ as expected from the production rate scaling. However, following the Bienaymé formula, the variance in the concentration is $\frac{\rho V}{\delta} \times \frac{1}{V^2} = \frac{\rho}{\delta V}$ (*Figure 6A*), which decreases with volume. This result makes intuitive sense because bigger cells have to produce more proteins to keep concentrations constant, so that the fluctuations in the relative number of proteins (and thus concentration) are smaller (see *Jia et al., 2021* for a complete analytical study of how in general variance scales differently from mean when volume varies). Thus, if the cell could sense the size of *concentration fluctuations* in some way, it would be able to harness the cell size-dependence of such stochastic fluctuations to regulate cell division and control cell size.

To test if we can evolve networks that control cell size through sensing Poissonian fluctuations, we first initialized our simulation with a network similar to that shown in *Figure 1C*, but with an added self-activating gene $A$ that can activate the production of the $I$ inhibitor. We then ran evolutionary simulations using the cell cycle structure of a sizer controlling G1 and timer in S/G2/M, but where the G1/S transition is regulated by the *concentration* $[I]$ instead of its *quantity*. We also used Pareto fitness optimization of $N_{Div}$ and of $CV_{Birth}$. Importantly, we use the stochastic version of our equations with

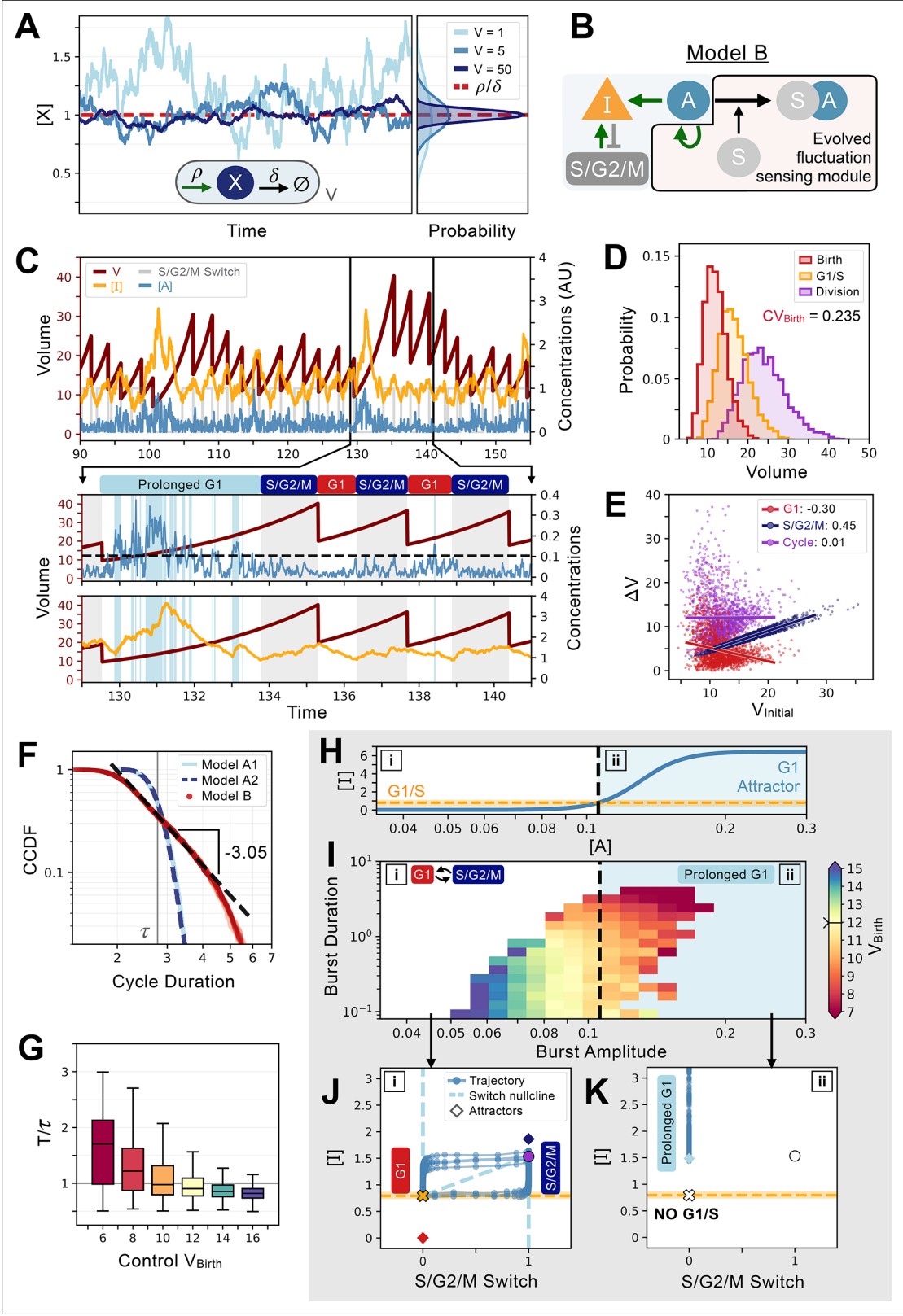

**Figure 6.** An evolved concentration fluctuation sensing size control model exhibits self-organized criticality. (**A**) Size-dependent molecular noise arises due to Poissonian fluctuations in molecule number. Consider a protein production-degradation scheme for protein *quantity X* with production rate $\rho$ and degradation rate $\delta$ contained in a volume *V*. At equilibrium, the concentration of $[X]$ will be given by a distribution with mean $\frac{\rho}{\delta}$ and variance $\propto \frac{1}{V}$. The effect of *V* on the molecular noise is shown on the time trajectories for simulations in a constant volume of *V*=1 (light blue), *V*=5 (medium blue),

*Figure 6 continued on next page*

*Figure 6 continued*

and *V*=50 (dark blue). Corresponding concentration distributions are shown on the right-hand side of the panel. (**B**) Network topology of Model B which evolved to sense fluctuations. Here, the G1/S transition is controlled by the *concentration* of $[I]$ and not its *quantity*. (**C**) Characteristic cell cycle dynamics of Model B. Trajectories of $[I]$, $[A]$, and S/G2/M Switch are rescaled with arbitrary units (AU) for visualization purposes. Below, we zoom-in on three cycles to show how a low-volume induced burst in $[A]$ leads to a massive production of $[I]$ inducing a temporarily prolonged G1 phase. Subsequently, cells become bigger and display lower molecular noise inducing a comparatively shorter G1 phase. (**D**) Volume distributions at birth (red), G1/S (orange), and division (purple) for Model B. The coefficient of variation of the volume distribution at birth $CV_{Birth} = 0.235$. (**E**) Amount of volume added $\Delta V$ in G1 (red), S/G2/M (dark blue), and over the whole cycle (purple) as a function of their initial volume at the beginning of these phases, *i.e.*, birth for G1 and cycle, and G1/S for S/G2/M, with the slope of linear fits indicated in legend. (**F**) Complementary cumulative distribution functions of the cycle duration (CCDF; probability that the cell cycle duration is larger than the value on the X-axis) for three models discussed in the main text: Model A1 (light blue), Model A2 (dashed dark blue), and Model B (red). The light grey line indicates the doubling time $\tau$. We see that Model B exhibits a long tail past the doubling time, which is consistent with a power-law scaling of the cycle duration probability. We find a criticality indicative scaling exponent of –3.05 for the CCDF after fitting the tail of the distributions of 5 independent realizations of the dynamics of Model B. (**G**) Box plots of the cycle length distributions as a function of control volume $V_C$ at birth. Cycle lengths are normalized by the doubling time $\tau$. Here, $V_C$ sets the molecular noise level to be equivalent to that of an exponentially growing cell born at $V_C$ but whose volume is reset to $V_C$ at each division. Note the very long tail of the distributions at small $V_C$ . (**H**) Position of the G1 attractor for inhibitor $[I]$ as a function of activator $[A]$. The black dashed line corresponds to the level of $[A]$ which triggers a transition between two modes of growth and division as shown by the position of the G1 attractor for $[I]$ becoming equal to the concentration required to induce the G1/S transition. The two modes of growth are labeled [i] and [ii] and are also indicated in panels I-K. (**I**) Activator protein $[A]$ is synthesized in bursts whose amplitude and duration are a function of volume. We define the burst duration as the total time during which $[A]$ >0 for a cycle. The burst amplitude corresponds to the average level of $[A]$ during each G1 phase. Each burst is then color-coded as a function of the birth volume of the cell that induced it. We use a divergent colormap whose center value (light yellow) corresponds to the average volume of the cells at birth and is indicated by a notch on the colorbar. Here, [i] corresponds to the deterministic regime when volume is high and [ii] corresponds to the noisy regime when volume is low. Note that the average volume of the cells at birth is positioned close to the black dashed line. (**J**) Phase-space representation of the relaxation oscillator in the deterministic regime [i]. The X-coordinate shows the S/G2/M Switch variable, and the Y-coordinate shows the concentration of $[I]$. Here, when volume is high, the position of the G1 attractor is *below* the $[I]$ concentration at which the G1/S transition happens. Thus, G1/S takes place and cells are in the cell cycle with a period of ~0.85. (**K**) Phase-space representation of the noisy regime [ii]. Here, when volume is low, the position of the G1 attractor becomes *greater* than the $[I]$ concentration at which the G1/S transition happens, and cells remain temporarily stuck in a prolonged G1 state and are unable to trigger the G1/S transition.

the molecular noise modeled using a Langevin noise term with a variance inversely proportional to the volume as explained above. We then extracted the most fit network and optimized it further using Pareto optimization of $N_{Div}$ and of the $\Delta V_{Cycle}$ slope to push the models towards the sizer regime. The most fit network of those combined evolutionary simulations is presented in *Figure 6B* and demonstrates that we can indeed evolve fluctuation-based cell size control (Model B).

The mechanism for size control that evolved based on size-dependent fluctuation sensing is remarkably similar to what we observed for models without size-dependent fluctuations (*Figure 5A*). For large volumes, the cycle has a constant period which corresponds to approximately 85% of the doubling time $\tau$. This ensures that in the high-volume regime, the system shrinks over time. When the volume is small however, fluctuations allow the concentration $[A]$ to cross the threshold of the highly non-linear transcriptional activation of $[I]$ by $[A]$. This results in a massive increase of $[I]$ that needs to be degraded to progress further into the cell cycle. Thus, the low volume regime occasionally leads to a considerable increase in G1 length and a correspondingly very large cell at division. These very large cells then reliably and deterministically re-enter multiple, rapid cell cycles with short G1 until the cell is small again and the concentration fluctuations again become large enough to trigger the activation of $[I]$ by $[A]$. This mechanism thus appears very similar to the early sizer mechanism observed in other quantity sensing simulations shown in *Figure 5*. However, here the mechanism is based only on size-dependent fluctuations in protein concentration and the overall behavior is closer to an adder (*Figure 6E*).

The system dynamics that evolved to perform fluctuation-based cell size control produce volume distributions that are long-tailed due to the stochastic occurrence of occasional exceptionally long G1 phases. Interestingly, the probability distribution of cell cycle durations follows a power law (*Figure 6F*), which is due to the very broad distributions of G1 duration at lower cell volumes. A more controlled analysis specifying the initial conditions showed that cell cycles get increasingly long, and their distributions widen with decreasing control volume $V_C$ (*Figure 6G*). Such non-Gaussianity is the hallmark of critical behavior, suggesting that the evolution of fluctuation-based cell size control is based on self-organized criticality (SOC). SOC is defined as a system where an order parameter feeds back on a control parameter (*Sornette et al., 1995*; *Vidiella et al., 2021*). The canonical example of

SOC is the sandpile to which grains of sand are added on top. As the sand accumulates, the slope steepens, and the angle of the pile (control parameter) increases. Eventually, this triggers avalanches (order parameter) that feedback to dramatically reduce the angle of the pile. This ensures that the system dynamically tunes itself at the critical value of the angle of the pile where avalanches can occur.

We conclude that our evolved size control network exhibits SOC based on several observations. Starting from a high volume, multiple divisions at a rate faster than it takes to double the biomass reduce cell volume $V$ just like the addition of grains of sand gradually increases the slope of the pile. Then, for small enough volumes, bursts of $[A]$ drive an extended G1 that greatly increases cell size, which, like the sandpile avalanches, resets the system's control parameter (volume of the cell or angle of the sandpile). Interestingly, evolution tuned the system to be near a bifurcation (*Figure 6I*). If we consider the deterministic regime, in which the fluctuations in $[A]$ are small, the cycle is unperturbed and oscillates with a period roughly equal to 85% of the doubling time (*Figure 6J*). In contrast, if we consider the noisy regime, in which the fluctuations in $[A]$ are large, the cycle disappears, and the system stays locked in a prolonged G1 state with a high value of $[I]$ which is akin to a bifurcation destroying the cycle (*Figure 6K*). This bifurcation takes place because the position of the G1 attractor for $[I]$ becomes larger with increasing $[A]$ and eventually overcomes the concentration required to induce the stochastic G1/S transition (*Figure 6H*). Then, the system remains stuck in a state where G1/S cannot be triggered, and cells effectively exit the cycle. As growth occurs, noise dies down and so does the position of the G1 attractor, eventually becoming smaller than the $[I]$ concentration required to induce the G1/S transition which allows cells to re-enter the cell cycle. Thus, the system is critical from a dynamical systems standpoint and also fits the general observation that SOC systems tune themselves to be right at the point where the order parameter is non zero, but infinitesimal (*Sornette et al., 1995*). In our case, the bifurcation corresponds exactly to the point where $[A]$ can sufficiently activate the production of $[I]$ to prevent the G1/S transition.

## Discussion

The last decade saw an explosion of time lapse microscopy studies measuring how cells control their size. These studies revealed diverse phenomena that are characterized by the correlation between cell size at birth and cell size at division. Size control ranged from sizers, where the size at division is uncorrelated from the size at birth, to adders, which add a constant volume in each cell division cycle, to timers, whose cell cycle duration is size-independent (*Cadart et al., 2018*; *Eun et al., 2018*; *Jun et al., 2018*; *Willis and Huang, 2017*; *Wood and Nurse, 2015*; *Zatulovskiy and Skotheim, 2020*). The presence of these diverse phenomena raises the question as to why the underlying control networks evolve one rather than another type of cell size control?

To explore the evolution of cell size control networks subject to distinct selection pressures, we used computational evolution simulations. We initially examined the evolution of a seed cell cycle model consisting of G1 and S/G2/M phases of similar duration, where the G1 phase was free to evolve size-dependence, but the S/G2/M phase was constrained as a timer. Our simulations reliably evolved a control mechanism based on a Mixed Feedback Loop (*François and Hakim, 2005*, *Figure 2*). This network is centered on a cell cycle regulator ($I$) that inhibits the G1/S transition in proportion to its quantity. $[I]$ is titrated away into an inactive complex by an increasing amount of another protein $[R]$ that is synthesized in proportion to cell size. This results in a size-dependent decrease in the effective cell cycle inhibitor (free $[I]$). Thus, our evolved network implements an effective dilution of a cell cycle inhibitor that is conceptually similar to the well-described inhibitor dilution models of budding yeast, human cells, and *Arabidopsis* plants (*D'Ario et al., 2021*; *Schmoller et al., 2015*; *Xie and Skotheim, 2020*; *Zatulovskiy et al., 2020*). We note that we did not allow the synthesis of our proteins, such as $[I]$, to be size-independent as has been found for budding yeast (*Chen et al., 2020*; *Schmoller et al., 2015*; *Swaffer et al., 2021*) as this could result in a one-step implementation of cell size control through the pure dilution of a cell cycle inhibitor. It is therefore interesting that given the constraint that all proteins be made in proportion to cell size, the network still evolved an effective 'dilution' of the active form of the cell cycle inhibitor molecule $[I]$.

Our evolution simulations gave insight into factors that bias evolution towards sizer or adder type control mechanisms (*Figure 4*). First, it is worth noting that our evolution simulations were not deterministic. There was no one-to-one correspondence between a given evolutionary pressure and any one specific cell size control mechanism. Rather, our claims represent an average behavior observed

over the course of many simulations. Size control, as measured by the CV at a particular point in the cell cycle, has contribution both from the slope of the correlation between cell size and the amount of cell growth, and from the amount of noise characterizing the differences between cells that are initially the same size (*Di Talia et al., 2007*). It is therefore possible that a low noise adder can produce a lower CV than a higher noise sizer. This is reflected in the evolutionary paths of some of our simulations, which traverse from a noisy sizer to a less noisy adder (*Figure 5*). However, we anticipate even noisy sizers will be better than adders at controlling cell size in response to large deviations away from the steady state distribution. This is because sizers will always return the cell size to be within the steady state distribution within a cell cycle. We note that these generic results of how sizers and adders can govern cell size homeostasis can be derived from more traditional analytical methods (*Barber et al., 2017*; *Willis et al., 2020*). However, our evolution simulations are particularly useful because the molecular networks that evolved give non-trivial insights into how the observed size homeostasis dynamics can be regulated (e.g. via a Mixed Feedback Loop or using a system close to criticality). They are also suggestive of evolutionary pathways: despite different evolutionary modalities and control types, a natural step in many of our simulated evolutions is a system with strong sizers at very small volume only (*Figures 5 and 6*). This is practically reflected in a strongly negative slope on the very left side of the $\Delta V_{Cycle}$ plots, and a positive slope at higher volume corresponding to timers (*Figures 5B and 6E*). Similar non-monotonicity of $\Delta V_{Cycle}$ has been identified in models of various realism and complexity (*Chandler-Brown et al., 2017*; *Delarue et al., 2017*) and we provide here an evolutionary explanation for such an effect. We thus predict that this will be observed in systems where CV at birth does not need to be tightly controlled.

In the selection of a size controlling G1 network followed by a timer in S/G2/M, we observed a prevalence of adders that is consistent with the prevalence of adders reported in the literature. While fewer in number, sizers have also been observed. That the most accurate sizers have been observed in the fission yeast *S. pombe* (*Fantes, 1977*; *Sveiczer et al., 1996*; *Wood and Nurse, 2015*), and that this organism performs cell size control at G2/M rather than at G1/S led us to explore the effect of cell cycle structure on the evolution of cell size control. We found that controlling cell size later in the cycle in S/G2/M biases evolution away from adders and towards sizers. In retrospect, this result can be rationalized since any size deviations incurred earlier during the timer period can be compensated for by the end of the cycle with the sizer. However, when the order is inverted, any size deviations escaping a G1 control mechanism would only be amplified by exponential volume growth during the S/G2/M timer period. A second recent case exhibiting sizer control was found in mouse epidermal stem cells, which exhibit a greatly elongated G1 phase and a relatively short S/G2/M phase (*Mesa et al., 2018*; *Xie and Skotheim, 2020*). We found that if we increased the relative duration of G1 in our simulations by shortening the S/G2/M timer, we also see a bias towards sizer control. In essence, by extending G1 to a larger and larger fraction of the cell cycle the control system is gradually approaching a size control taking place at the end of the cell cycle, that is, an S/G2/M size control. Taken together, these simulations suggest the principle that having size-dependent transitions later in the cell cycle selects for sizers, while having such transitions earlier selects for adders.

In addition to identifying cell cycle structural features that canalize evolution towards sizers and adders as described above, we also observed an intriguing mechanism relying on molecular fluctuations. In this case, small cells would trigger an abnormally long G1 that would result in very large cells that decrease in size through a series of rapid cell divisions. This type of size control is reminiscent of that found in the green algae *Chlamydomonas* where a series of rapid, size-reducing cell divisions cease when cells go below a target size (*Heldt et al., 2020*). In our case, small size results in larger concentration fluctuations due to Poisson noise in the number of molecules. These concentration fluctuations, when large enough, are then able to trigger a burst of G1/S inhibitor that leads to an extended G1 phase and massive cell size growth before another series of rapid cell divisions is initiated (*Figure 6*). Interestingly, the system thus performs statistical size control over many generations. Intriguingly, this size control mechanism exhibits hallmarks of self-organized criticality (SOC). Just like adding grains of sand to a pile eventually triggers avalanches, the consistent decrease of cell size in the rapid division cycles eventually triggers a greatly extended G1 phase. To our knowledge, this is the first example where self-organized criticality is obtained in artificially evolved models of gene networks performing a well-defined function and is consistent with the idea that evolution of complex systems can favor the emergence of critical processes.

## Materials and methods
### Mathematical formalism
To model gene networks, we follow a standard ODE based formalism, where we simulate dynamics of the concentrations of proteins. We use Hill functions for transcriptional interactions, and standard mass action kinetics for protein-protein interactions. We also assume that all proteins are degraded at a constant rate. For most of the simulations presented in the paper, we use deterministic ODEs for simplicity. Importantly, cell-to-cell variability arises from the precise timing of cell cycle progression events. This allows for a natural way to generate noise on cell volume that should then be compensated for by the evolved network. In the last part of the paper, we explicitly include Langevin noise for biochemical reactions that are modeled using a classical tau-leaping formalism (*Gillespie, 2007*). Thus, each biochemical reaction takes place with a rate that corresponds to the deterministic rate, to which we add one white Gaussian noise with a variance equal to that rate. For example, given a deterministic biochemical rate $k$ and a time interval of size $\Delta t$, we consider a tau-leaping change of $k\Delta t + \mathcal{N}(0, k\Delta t)$ where $\mathcal{N}(0, k\Delta t)$ is a random gaussian variable of mean 0 and variance $k\Delta t$.

Volume influences protein dynamics in three ways. First, protein production rates are generally proportional to cell volume so that proteins reach and maintain a constant concentration that is independent of the cell volume (*Chen et al., 2020*; *Elliott and McLaughlin, 1978*; *Newman et al., 2006*; *Swaffer et al., 2021*). We note that we are not allowing the cell to employ proteins such as Whi5 in budding yeast whose production is independent of cell size so that its concentration is a direct readout of cell size (*Schmoller et al., 2015*; *Swaffer et al., 2021*). We chose to do this because we want to explore how cell size control can be done by a network with multiple feedbacks rather than just the concentration of a single protein with a special dedicated synthesis mechanism. Thus, the only deterministic influence of volume on concentration dynamics is on the dilution rate, which is proportional to the cell growth rate $\lambda(V)$ (see details in Appendix 1). At cell division, we also assume that proteins are equally partitioned between the daughter cells, that is, the concentration is the same before and after division. Note that we scale all our variables so that a concentration of one arbitrary unit corresponds roughly to 1000 proteins in a 100fL cell (*Milo et al., 2010*). Additionally, we scale the time variable so that 1 arbitrary time unit corresponds roughly to 30 min (*Di Talia et al., 2007*).

In this study, we chose a hierarchical way of introducing noise in the system, starting with the biggest contributing factor and incrementally adding additional sources of noise in subsequent analyses. All simulations presented include noise (stochastic control of G1/S transition and timing of S/G2/M, see below) in the cell cycle phases, whose CV has been found to be as high as 50% (*Di Talia et al., 2007*). Then, we introduced protein production noise via Langevin noise because the CV of regulatory protein concentrations is typically 20–30% (*Newman et al., 2006*). Importantly, the cell volume also contributes to stochastic effects, which are larger in smaller cells with fewer molecules. Thus, for stochastic simulations, we include a multiplicative $\frac{1}{\sqrt{V}}$ contribution to the added Gaussian noise term (see more complete description in the Appendix 1).

We also checked that our results are largely invariant when adding other sources of noise (see *Appendix 1—figures 5–7*). In these simulations, we also included noise in cell growth rate (CV ~15%; e.g. *Di Talia et al., 2007*), and in mass partitioning at cytokinesis (CV ~10%; e.g. *Zatulovskiy et al., 2020*).

### Evolutionary procedure
To evolve networks regulating cell size, we use the $\varphi$-evo software (*Henry et al., 2018*) with a modified numerical integrator accounting for volume dynamics and volume dependencies as described above (*Figure 1B*). $\varphi$-evo simulates the Darwinian evolution of a population of gene networks. A network is encoded with the help of a bipartite graph connecting biochemical species (typically proteins) and interactions between them (we use a custom-made Python library). Networks are converted into an ensemble of stochastic differential equations using a Python to C interpreter. This code is then compiled and integrated on the fly to compute the behavior of the networks.

Each selection step in the algorithm is referred to as an *epoch* rather than the more commonly used term *generation* because we use the term generation to refer to cell divisions in the simulations. At each epoch of the algorithm, each gene network is simulated, and its fitness computed. Based on the fitness function(s) (see below), half of the networks are selected and duplicated, while the other half is discarded to maintain a constant population size. The duplicated networks are then randomly

mutated. From the most to least probable, mutations consist in random changes of parameters of the network, random removal of interactions, and random additions of interactions or new proteins. Absolute mutation rates are adjusted as a function of the number of evolutionary epochs so that all networks in a population are mutated on average once per epoch. This implements a numerical equivalent of the biological Drake's rule that mutation rates adjust with genome size (*Lynch, 2007*). Practically, this prevents the known phenomenon of code-bloating in evolutionary simulations (*Foster, 2001*) and also means that the total number of epochs is a good proxy of the number of mutations (in random directions) needed to evolve the best networks. All of this is easily made with our customized Python library encoding networks. For more details on technical aspects and implementations of the $\varphi$-evo software, we refer the reader to *Henry et al., 2018*.

Realistic evolutionary processes select for multiple phenotypes in parallel. While trade-offs between those phenotypes are non-trivial, it has been observed that phenotypes typically define an evolutionarily Pareto front (*Shoval et al., 2012*; *Warmflash et al., 2012*). We thus perform network selection using a Pareto mode (*Warmflash et al., 2012*), in which two distinct fitness functions are computed. During the selection step, networks are first Pareto ranked. For example, consider two networks A and B and two fitness functions $f^1$ and $f^2$. $f^1_A$ refers to the fitness of network A calculated with function $f^1$. Assuming fitness functions are to be maximized, we say network A Pareto-dominates network B if both $f^1_A > f^1_B$ and $f^2_A \geq f^2_B$. Rank 1 networks are networks which are not dominated by any other networks, Rank 2 networks are networks dominated only by Rank 1 networks, and Rank 3 networks are only dominated by Rank 1 and Rank 2 networks and so on. The algorithm then selects half of the population of Rank 1 networks using a fitness sharing algorithm to maximize population diversity (see details in *Warmflash et al., 2012*). One advantage of Pareto selection is the increased flexibility of the evolutionary process. Multiple fitness functions can provide different optimization paths in parameter space, which prevents the selection process from getting stuck in a local optimum of a single fitness function. We also perform a few simulations with only one fitness function, in which case networks are simply ranked based on their fitness.

We impose two evolutionary selection pressures in the form of two fitness functions. The first fitness function is simply the number of cell divisions during a long period, which we call $N_{Div}$ . This is consistent with the classical definition of fitness as optimizing the number of offspring and is to be maximized by the algorithm. The second fitness function is the coefficient of variation of the volume distribution at birth for those $N_{Div}$ generations, which we call $CV_{Birth}$ and is to be minimized by the algorithm. This penalizes broad distributions of volume at birth, which are detrimental to cell size homeostasis, which is what we aim to examine here. We further imposed fitness penalties to prevent way too small or too big cells, see Appendix 1. There, we also study alternative fitness functions, such as least-square residual function to minimize volume variation about a target size, and the fitted slope of the amount of volume added at each cycle to be minimized to drive models toward being a sizer.

## Varying cell cycle structure

We also ran evolutionary simulations with different cell cycle structures. For evolutionary simulations where the G1/S transition was controlled by the concentration of $[I]$ , we simply change the probability to pass the G1/S transition to depend on concentration $[I]$ instead of its quantity $I$. For evolutionary simulations with a cell cycle structure similar to that found in the fission yeast *S. pombe*, we invert the cell cycle network structure. In this case, $I$ quantity controls division and the Switch is turned on for a fixed amount of time in G1. In terms of the relaxation oscillator, this means that the left branch is now S/G2/M and the right branch is G1.

## Acknowledgements

We thank Rodrigo Reyes-Lamothe, Nicolas E Buchler, and Lucas Fuentes Valenzuela for helpful comments on the manuscript. JS was supported by the NIH (R35 GM134858), PF was supported by Natural Sciences and Engineering Research Council of Canada (NSERC), Discovery Grant Program, FPG was supported by a Fonds de Recherche du Québec Nature et Technologies (FRQNT) Doctoral scholarship (B2X) and by a Natural Sciences and Engineering Research Council of Canada (NSERC) Doctoral Canada Graduate Scholarship (CGS-D).

## Additional information

### Funding

| Funder | Grant reference number | Author |
|---|---|---|
| Natural Sciences and Engineering Research Council of Canada | Discovery Grant | Paul François |
| Natural Sciences and Engineering Research Council of Canada | Alexander Graham Bell Canada Graduate Scholarship | Felix Proulx-Giraldeau |
| Fonds de recherche du Québec – Nature et technologies | Doctoral research scholarship | Felix Proulx-Giraldeau |
| National Institutes of Health | NIH R35 GM134858 | Jan M Skotheim |
| Chan Zuckerberg Initiative | Biohub Investigator Award | Jan M Skotheim |

The funders had no role in study design, data collection and interpretation, or the decision to submit the work for publication.

### Author contributions

Felix Proulx-Giraldeau, Conceptualization, Software, Validation, Investigation, Visualization, Methodology, Writing - original draft; Jan M Skotheim, Conceptualization, Supervision, Methodology, Writing - original draft, Project administration, Writing - review and editing; Paul François, Conceptualization, Software, Supervision, Funding acquisition, Methodology, Writing - original draft, Project administration, Writing - review and editing

### Author ORCIDs

Felix Proulx-Giraldeau (D) http://orcid.org/0000-0003-0238-5410
Paul François (D) http://orcid.org/0000-0002-2223-839X

### Decision letter and Author response

Decision letter https://doi.org/10.7554/eLife.79919.sa1
Author response https://doi.org/10.7554/eLife.79919.sa2

## Additional files

### Supplementary files

• MDAR checklist

### Data availability

This is a theory paper, so there is no experimental data, and all results were generated by the code. The code used is freely available at https://github.com/FelixPG/PhiEvo_SizeControl, (copy archived at swh:1:rev:afa7f16a2f8a9d793aa3685116c2436faae100dd). Reference to the code has been added in the text.

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

# Appendix 1

In section Mathematical implementation, we first describe how to modify deterministic and stochastic ODEs to account for a exponentially growing cell volume. In section Size control, we describe the three size control archetypes, namely the timer, the adder, and the sizer, as well as our implementation of the initial seed cell cycle model. Then, in section Evolutionary algorithm, we give details on the $\varphi$-evo evolutionary algorithm. In section Analysis of sources of noise, we present evolutionary simulations investigating the effects of noise on the evolved networks. Finally, in sections Model descriptions and Additional models, we provide parameter values and equations for the models presented in the main text and for additional models produced by our evolutionary simulations.

## Mathematical implementation

### Deterministic

All biochemical concentrations can be described as a quantity of molecules of a biochemical species divided by the volume that contains it. The fundamental equation describing all concentrations is thus:

$$[X](t) = \frac{X(t)}{V(t)} \tag{1}$$

Here, $X(t)$ is the quantity of molecules of an arbitrary biochemical species $X$ as a function of time and $V(t)$ is the volume of the cell containing said species as a function of time. We will use the bracket notation $[X](t)$ for concentrations. In this project, we will model all biochemical species directly at the concentration level and assume proteins are uniformly distributed in an exponentially growing cell volume. The absolute growth rate of the cell $\lambda$ is chosen to be constant over a large viable range of cell volume and is otherwise 0. Consequently, volume grows over time with a fixed absolute growth rate $\lambda = 0.25$ over a viable volume range following the equation:

$$\frac{dV}{dt} = \lambda V(t) \tag{2}$$

Deterministic rate equations describing the dynamics of the biochemical species at the concentration level have to be adjusted to take into consideration a time-varying volume. From *Equation 1* and the derivative chain rule, we get:

$$\frac{d[X]}{dt} = \frac{d}{dt}\frac{X(t)}{V(t)} = \frac{1}{V(t)}\frac{dX}{dt} - \frac{X(t)}{V(t)^2}\frac{dV}{dt}$$

which we can combine with *Equation 2* to give:

$$\frac{d[X]}{dt} = \frac{1}{V(t)}\left(\frac{dX}{dt}\bigg|_{V=\mathrm{Cst}}\right) - \lambda[X](t) = f([X](t)) - \lambda[X](t) \tag{3}$$

The first term in the *Equation 3*, $f([X](t))$, corresponds to the usual biochemical reaction rates that occur when the volume of the cell is fixed. The second term, $-\lambda[X](t)$, is a dilution term that we can interpret as an effective degradation of the concentration $[X](t)$ due to the exponential growth of the cell volume over time.

To further simplify the expression for $f([X](t))$, we make an assumption about the production rates of all biochemical species in our models. Let us consider the rate equation describing the dynamics of the quantity of an arbitrary protein $X$ with generic production rate $\rho$ and degradation rate $\delta$ contained in a fixed cell volume $V$.

$$\frac{dX(t)}{dt} = \rho - \delta X(t)$$

Here, $f([X](t))$ is given by:

$$f([X](t)) = \frac{1}{V}\frac{dX(t)}{dt} = \frac{\rho}{V} - \delta[X](t)$$

We assume that all the proteins in our models are constitutively expressed by the cell. In other words, the production rates $\rho$ are linear functions of the cell volume $\rho = \rho(V(t)) = \rho_0 V(t)$. This ensures that the protein production rates scale with the volume such that concentrations stay constant over

time which is a general feature of most proteins in *S. cerevisiae* (*Chen et al., 2020*; *Newman et al., 2006*; *Swaffer et al., 2021*). This yields: $f([X](t)) = \rho_0 - \delta[X](t)$

Taken altogether with *Equation 3*, the deterministic dynamics of a constitutively expressed arbitrary protein concentration $[X](t)$ contained in an exponentially growing cell volume are given by:

$$\frac{d[X]}{dt} = \rho_0 - (\delta + \lambda)[X](t) \tag{4}$$

## Stochastic

To simulate molecular noise, we follow a classical tau-leaping formalism (*Gillespie, 2007*). Specifically, we choose the Euler-Maruyama implementation to generate approximate solutions to stochastic differential equations (*Kloeden and Platen, 1992*).

As we have done before in the deterministic case, let us first consider the quantity of an arbitrary protein $X(t)$ in order to extract the equation for the concentration $[X](t)$. Let's assume $X(t)$ is changing via a *single* biochemical reaction rate $\frac{dX}{dt} = g(X(t))$ over the time interval $[0, T]$ given $X(t = 0) = X_0$. We will show later how this approach can be generalized to include multiple reaction rates. We begin by partitioning the time interval in $N$ equal segments of length $\Delta t$ such that $0 < t_1 < t_2 < ... < t_N = T$ with $t_n = n \cdot \Delta t$, $n \in \{1, N\}$.

The Euler-Murayama approximate solution to the stochastic differential equation at the discrete time points $t_n$ is then recursively given by the following equation for $n \in \{1, N-1\}$ where the single biochemical reaction is assumed to happen with a Poisson rate (corresponding to the deterministic rate), which adds one white Gaussian noise to the differential equations with a variance equal to that rate:

$$X_{n+1} = X_n + g(X_n)\Delta t + \sqrt{|g(X_n)|\Delta t} \cdot \mathcal{N}(0, 1) \tag{5}$$

Here, $X_n = X(t_n)$, $g(X_n)$ is the drift term, $\sqrt{|g(X_n)|}$ is the diffusion term and $\mathcal{N}(0, 1)$ is a random Gaussian variable of mean 0 and variance 1. In other words, $X_{n+1}$ is a random Gaussian variable of mean $X_n + g(X_n)\Delta t$ and variance $|g(X_n)|\Delta t$. Since this describes the quantity of proteins, we also have to consider the change in volume over time to recover the equation for the concentration of proteins, $[X](t)$. Thus, let's consider the volume of the cell $V(t)$ at the discrete time points $t_n$, which is given recursively by the equation:

$$V_{n+1} = V_n + \lambda V_n \Delta t = V_n(1 + \lambda \Delta t) \tag{6}$$

Here, we define $V_n = V(t_n)$ and recover the $\lambda V_n$ term from *Equation 2*. We assume that the volume time evolution is noiseless for simplicity. To recover, the differential equation describing the protein concentration, we evaluate the expression $[X]_{n+1} = \frac{X_{n+1}}{V_{n+1}}$. Thus, combining *Equations 5 and 6*, we get:

$$[X]_{n+1} = \frac{X_n + g(X_n)\Delta t + \sqrt{|g(X_n)|\Delta t} \cdot \mathcal{N}(0,1)}{V_n(1 + \lambda \Delta t)}$$

$$[X]_{n+1} = \frac{1}{1 + \lambda \Delta t}\left([X]_n + g([X]_n)\Delta t + \sqrt{\frac{|g([X]_n)|\Delta t}{V_n}} \cdot \mathcal{N}(0, 1)\right)$$

where we identify $\frac{1}{V_n}g(X_n) = g([X]_n)$ as the rate equation describing the *concentration* of the consitutively expressed protein $[X](t)$ when the volume $V$ is fixed as described in the deterministic case. Then, we compute the derivative as:

$$\left.\frac{d[X]}{dt}\right|_{t_n} \approx \frac{[X]_{n+1} - [X]_n}{\Delta t}$$

$$= \frac{1}{\Delta t(1 + \lambda \Delta t)}\left([X]_n - (1 + \lambda \Delta t)[X]_n + g([X]_n)\Delta t + \sqrt{\frac{|g([X]_n)|\Delta t}{V_n}} \cdot \mathcal{N}(0, 1)\right)$$

$$= \frac{1}{1 + \lambda \Delta t}\left(g([X]_n) - \lambda[X]_n + \sqrt{\frac{|g([X]_n)|}{V_n \Delta t}} \cdot \mathcal{N}(0, 1)\right)$$

Finally, we recover the approximate full differential equation for the protein concentration by expanding the prefactor in the last equation to the 0$^{\text{th}}$ order in $\Delta t$ assuming it to be small to give:

$$\left.\frac{d[X]}{dt}\right|_{t_n} \approx g([X]_n) - \lambda[X]_n + \sqrt{\frac{|g([X]_n)|}{V_n \Delta t}} \cdot \mathcal{N}(0, 1) \tag{7}$$

So far, we have assumed that there is only a *single* biochemical reaction rate $g([X](t))$. We can however easily generalize our approach to include additional reaction rates by summing up the contribution of each rate to the total differential equation. Given $M$ independent reaction rates $g_i([X](t))$ and the properties of random Gaussian variables, we can easily generalize:

$$\left.\frac{d[X]}{dt}\right|_{t_n} \approx \sum_{i=1}^{M} g_i([X]_n) - \lambda[X]_n + \sum_{i=1}^{M}\left(\sqrt{\frac{|g_i([X]_n)|}{V_n \Delta t}} \cdot \mathcal{N}_i(0,1)\right) \tag{8}$$

Importantly, the $\mathcal{N}_i$ are Gaussian vectors accounting for the noise correlations associated with single reactions. For instance, imagine one protein $X_k$ turns into another protein $X_p$, then the corresponding Gaussian vector for this interaction takes the form $\mathcal{N}(0,1)(\vec{x_k} - \vec{x_p})$ where $\vec{x_k}$ is vector of length corresponding with the number of variables in the system whose $k$-th component is equal to 1 with 0s elsewhere. This indicates that the molecular fluctuation due to this reaction should have opposite signs for $X_k$ and $X_p$ as expected.

The term $\lambda[X]_n$ is a dilution term that corresponds to an effective degradation of protein concentration $[X]_n$ as seen in the deterministic case. Interestingly, we highlight the $\frac{1}{V_n}$ dependency in the noise term. We can understand this dependency intuitively by considering a protein production process with a Poisson parameter $\theta$. In this scenario, the mean and the variance of the protein quantity distribution is given by the parameter $\theta$. Going back to concentration space, there are an infinite number of combinations of protein quantity and volume that can give the same concentration $[X] = X/V$. Thus, we need to specify both the protein number $X$ and the volume $V$ to correctly model the molecular noise contributing to fluctuations in concentrations.

## Size control

### Initial seed network

To guide the evolutionary process, we begin with an initial seed network. We base our first seed network on the phenomenology of the budding yeast *S. cerevisiae*'s cell cycle, where cell size primarily regulates the timing of the START transition in late G1 (*Schmoller et al., 2015*). This regulation allows small daughter cells to delay the G1/S transition allowing them to catch-up in size by extending the G1 phase. S/G2/M duration on the other hand is largely independent of cell size. We note that while budding yeast divide asymmetrically, our simulated cells divide symmetrically. In our simple initial seed network, the cell cycle consists of two phases, G1 and S/G2/M, which respectively denote the pre-G1/S and post-G1/S phases of the cell cycle. The transition between these two phases is controlled by the level of a transcription regulator we call $I$. Like the Whi5 protein in *S. cerevisiae*, $I$ is an inhibitor of the G1/S transition such that the lower its level, the higher the chances of cell cycle progression. Since protein production rates were assumed to be dependent on volume (as described in section Mathematical implementation), we found that $I$'s concentration alone was largely independent of volume and could not trigger a size-dependent G1/S transition as Whi5 does in budding yeast. Thus, we chose the quantity of $I$ defined as $\mathbf{I}(t) = [\mathbf{I}](t) \times V(t)$ as the control variable for this transition. We chose to model the probability of the G1/S transition occurring at the next time point of the simulation with a sigmoid-shaped curve given by *Equation 9* that can be visualized in *Appendix 1—figure 1*. We maintain $\theta = 0.8$ and $n_\theta = 8$ fixed throughout this project. We chose these values because they give a similar amount of noise in the G1/S transition as observed experimentally (*Di Talia et al., 2007*; *Chandler-Brown et al., 2017*).

$$\mathcal{P}_{\text{G1/S}}(t) = \frac{2}{1 + \exp\left((\mathbf{I}(t)/\theta)^{n_\theta}\right)} \tag{9}$$

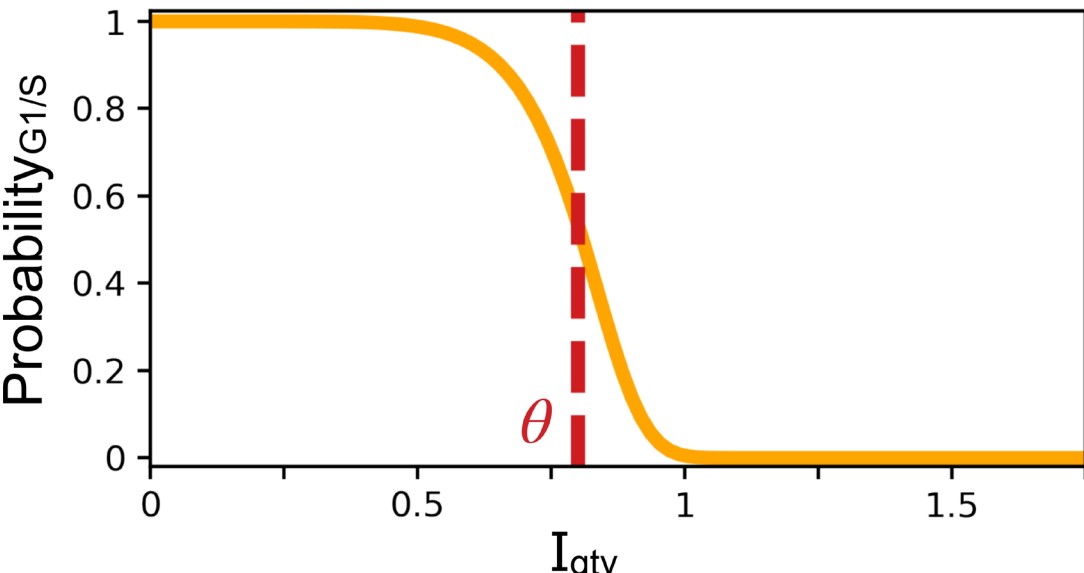

**Appendix 1—figure 1.** Probability of the G1/S transition occurring at the next time step. X-coordinate is the quantity of the transcriptional regulator $I$. Y-coordinate is the probability of the G1/S transition occurring at the next time step $\Delta t$. Parameters $\theta = 0.8$ and $n_\theta = 8$.

In our seed model, we encode cell cycle state using a binary variable **S/G2/M Switch**, which is 0 in G1 and turns to 1 in S/G2/M once the G1/S transition takes place. Following *S. cerevisiae*'s cell cycle structure where S/G2/M duration is independent of cell size, we fix S/G2/M duration to be $\sim 50\%$ of the doubling time $\tau = \frac{1}{\lambda} \ln 2$ with uniform noise unless stated otherwise. This way, cells can tune the length of their cell cycle by adjusting G1 length while being constrained by the incompressible length of the timer in S/G2/M, $T_{\text{S/G2/M}}$.

Following S/G2/M, cells divide such that $V_{\text{Birth}}^{(n+1)} = V_{\text{Division}}^{(n)}/f$ with $f$ the division fraction. The $n$ exponent here is referencing the $n$-th generation in the cell lineage. We choose $f = 2$ for all simulations performed in this study unless explicitly mentioned otherwise. We assume perfect partitioning of all proteins between the two daughter cells such that the proteins' concentrations remain the same before and after division. After division, we follow one of the two daughter cells during their own subsequent cycle. If we simulate the cell lineage for a long time, ergodicity guarantees that all volume states will be visited given stable growth and we can extract population statistics from the lineage data itself. Here, cell growth is exponential on the single cell level since we were assessing size control mechanisms that take size as an input to cell cycle control. We are not exploring the very interesting case where growth deviates from the exponential. In that case, size homoeostasis would have a contribution from some cells in the population outcompeting others in terms of their growth and we would have to simulate the entire cell population and not disregard one of the daughter cells as we do here.

Inspired by the dynamics of Whi5, which is produced in S/G2/M and diluted in G1, we chose an initial seed network where S/G2/M Switch activates the transcription of the $I$ inhibitor. This ensures that the concentration [**I**] is 'reset' to a higher value following S/G2/M and prevents cells from skipping entirely the G1 phase of the subsequent cycle which would quickly send the volume of the cell converging quickly towards 0. We note that this interaction systematically appeared anyway in our early evolution simulation so we chose to include it in the initial seed network to accelerate the evolutionary process.

## Size control archetypes

Size control mechanisms are often compared to three well-characterized models or archetypes in order to quantify the strength of the size control mechanism under study. Specifically, there are timers, adders and sizers. In this subsection, we will define each archetype and show that we can summarize them via a control volume response curve $T(V_C)$ as described in the main text.

First, let's consider the timescale of growth. Given $V(t = 0) = V_0$, the solution $V(t)$ of the volume *Equation 2* is $V(t) = V_0 e^{\lambda t}$. From this equation, we can easily recover the doubling time $\tau$ defined as the time required to double a cell's volume ($V(\tau) = 2V_0$) which yields:

$$\tau = \tfrac{1}{\lambda} \ln 2 \qquad (10)$$

The doubling time $\tau$ is only a function of the growth rate $\lambda$. Since we are only considering symmetrical division events, fixed interdivision times shorter than the doubling time will yield progressively smaller daughter cells. Similarly, fixed interdivision times longer than the doubling time will yield progressively larger daughter cells. Thus, the absolute growth rate $\lambda$ sets the timescale for cell cycle dynamics if we want to simulate a stable cell lineage.

With the quantity sensing of the inhibitor I at the G1/S transition (see *Equation 9*), we find that the instantaneous volume at G1/S sets the concentration of the biochemical species for the rest of the cycle and until the next G1/S transition in the daughter's cell cycle. Consequently, it regulates the timing of the G1 phase of the daughter cell and thus creates a return map for the volume at G1/S. We find that the volume at G1/S at the $(n + 1)$-th generation $V_{\mathrm{G1/S}}^{(n+1)}$ is given recursively by:

$$V_{\mathrm{G1/S}}^{(n+1)} = \frac{V_{\mathrm{G1/S}}^{(n)} \cdot e^{\lambda(T_{\mathrm{S/G2/M}} + T_{\mathrm{G1}}(V_{\mathrm{G1/S}}^{(n)}))}}{2} = \frac{V_{\mathrm{G1/S}}^{(n)} \cdot e^{\lambda T(V_{\mathrm{G1/S}}^{(n)})}}{2}$$

To study the mechanisms of cell size control, we choose to define a useful new variable: the control volume $V_C$. This control variable is independent from the biochemical network and maintained fixed allowing us to break the size feedback, and distinguish its input, the volume $V$, from its output, the induced cycle period $T$. With this new variable, we can modify the control at G1/S by forcing the transitions to trigger once $\mathbf{I}_C = [\mathbf{I}] \times V_C$ is low enough. We can then extract the response curve of the system, that is, the cell cycle period induced from sensing this control volume at G1/S $T(V_C)$. We represent this process schematically in *Appendix 1—figure 2A*.

The control volume at which the response curve $T(V_C)$ is equal to the doubling time $\tau$ corresponds to the equilibrium volume, $V_{eq}$, for this network. Indeed, if $T(V_{\mathrm{eq}}) = \tau$, then the cell cycle length will ensure that this cell exactly doubles its volume during its cell cycle and returns to the same $V_{\mathrm{eq}}$ at the next generation. This volume is a fixed point of the volume return map and can be either stable or unstable. Theoretically, there could be size control mechanisms with multiple fixed points of the response curve, but practically we have not seen this emerge from any of our evolution experiments and therefore assume that $V_{\mathrm{eq}}$ is unique. In the main text, we have substituted $V_{\mathrm{eq}}$ by the average value of the volume at the time where volume is sensed as both of these values are essentially identical. This corresponds to $\langle V_{\mathrm{G1/S}} \rangle$ for models with a sizer in G1 and a timer in S/G2/M and $\langle V_{\mathrm{Division}} \rangle$ for networks with a timer in G1 and a sizer in S/G2/M.

Size variation naturally occurs in our models due to the precise timing of G1 and S/G2/M cycle phases which are both noisy, so the volume does not stay at $V_{\mathrm{eq}}$ for very long. The stability of growth around this equilibrium volume however will depend on the sign of the local derivative with respect to control volume of the $T(V_C)$ response curve and we will consider the following three cases:

- is strictly increasing with $V_C$. In this case, small deviations around $V_{\mathrm{eq}}$ are amplified over successive generations and the volume quickly shrinks to 0 or explodes to ∞. In this case, we say that $V_{\mathrm{eq}}$ is an unstable fixed point of the response curve.
- $T(V_C)$ is constant with $V_C$. In this special case and assuming exponential growth of the volume, the only stable mode of growth corresponds to the response curve $T(V_C) = \tau$. This corresponds to the only stable timer archetype. In this particular scenario, $V_{\mathrm{eq}}$ is not well defined as there are an infinite number of volumes where the response curve intersects the doubling time.
- $T(V_C)$ is strictly decreasing with $V_C$. In this case, cells correct for size deviations over successive generations and perform size control. In this case, we say that $V_{\mathrm{eq}}$ is a stable fixed point of the response curve.

Here, we found the $T(V_C)$ curves that were selected by the evolutionary algorithm were all decreasing with $V_C$ as expected for stable size control mechanisms. Thus, for the remainder of this document, we will assume that $V_{\mathrm{eq}}$ is uniquely defined and corresponds to a stable fixed point of the control volume response curve.

## Timer

The timer archetype describes mechanisms that monitor time rather than size. If the cycle duration of the timer is tuned precisely to the doubling time $\tau = \frac{1}{\lambda} \ln 2$, cells will double their mass over the course the cell cycle to ensure that newborn daughter cells have the same volume at birth as their mothers did when they were born. Consequently, this category of mechanisms is notoriously bad at correcting for size deviations given exponential cell volume. If growth was linear however, this mechanism would allow for size control to take place.

$$T_{\text{Timer}}(V_C) = \tau = \frac{1}{\lambda} \ln 2 \tag{11}$$

## Adder

The adder archetype describes mechanisms where cells add a constant amount of cell volume during each cycle. We define $\Delta$ the increment of added volume between birth and division, that is $\Delta = V_{\text{Division}} - V_{\text{Birth}}$. For adder mechanisms, the added volume at each cycle is constant and does not depend on cell size. In this case, initial size deviations are reduced by a factor of 2 at each division such that the volume at birth geometrically converges to the added volume $\Delta$ over successive generations. To recover the adder response curve $T_{\text{Adder}}(V_C)$, we consider that by definition, $V^{\text{Division}} = V^{\text{Birth}} e^{\lambda T_{\text{Adder}}}$. From this equation and the definition of the adder, we can recover the cycle period $T_{\text{Adder}}$:

$$T_{\text{Adder}}(V_{\text{Birth}}) = \frac{1}{\lambda} \ln \left( \frac{\Delta}{V^{\text{Birth}}} + 1 \right)$$

We would like write this equation as a function of the control volume $V_C$ at G1/S and the equilibrium volume $V_{\text{eq}}$ alone. Assuming that the G1/S transition is followed by a timer in S/G2/M, we can write $V_{\text{Birth}} = V_C e^{\lambda T_{\text{S/G2/M}}}/2$. Similarly, since we know by definition that $T_{\text{Adder}}(V_{\text{eq}}) = \tau$, we can recover that the added volume increment is $\Delta = V_{\text{eq}} e^{\lambda T_{\text{S/G2/M}}}/2$. Finally, we can combine these two expressions with to recover the final expression of the adder response curve:

$$T_{\text{Adder}}(V_C) = \frac{1}{\lambda} \ln \left( \frac{V_{\text{eq}}}{V_C} + 1 \right) \tag{12}$$

We note here that if the control volume was measured at division or at birth, the response curve of the adder would be unchanged. The only difference would be that both $V_C$ and $V_{\text{eq}}$ would correspond to volumes at division or birth volumes instead of volume at the G1/S transition. Here for example, the volume increment $\Delta$ corresponds to the $V_{\text{eq}}$ at birth.

## Sizer

The sizer archetype describes mechanisms that measure size directly and allow a cell to return to a target volume $V_{\text{Target}}$ after a single generation, irrespective of how big or small a cell was initially. For sizers, $V^{\text{Division}} = V_{\text{Target}} = V^{\text{Birth}} e^{\lambda T_{\text{Sizer}}}$ by definition.

We can then extract:

$$T_{\text{Sizer}}(V_{\text{Birth}}) = \frac{1}{\lambda} \ln \left( \frac{V_{\text{Target}}}{V_{\text{Birth}}} \right)$$

We can then write $V_{\text{Birth}}$ and $V_{\text{Target}}$ as a function of the control volume at G1/S $V_C$ and the equilibrium volume $V_{\text{eq}}$. Using the same definitions as before $V_{\text{Birth}} = V_C e^{\lambda T_{\text{S/G2/M}}}/2$ and $T_{\text{Sizer}}(V_{\text{eq}}) = \tau$, we find that $V_{\text{target}} = V_{\text{eq}} e^{\lambda T_{\text{S/G2/M}}}$. The sizer response curve thus follows:

$$T_{\text{Sizer}}(V_C) = \frac{1}{\lambda} \ln \left( \frac{2V_{\text{eq}}}{V_C} \right) \tag{13}$$

We note again here that if the volume was measured at division or at birth, the equation for the sizer would be identical with the only difference being that the control volume $V_C$ and the equilibrium volumes $V_{\text{eq}}$ would correspond to division or birth volumes respectively.

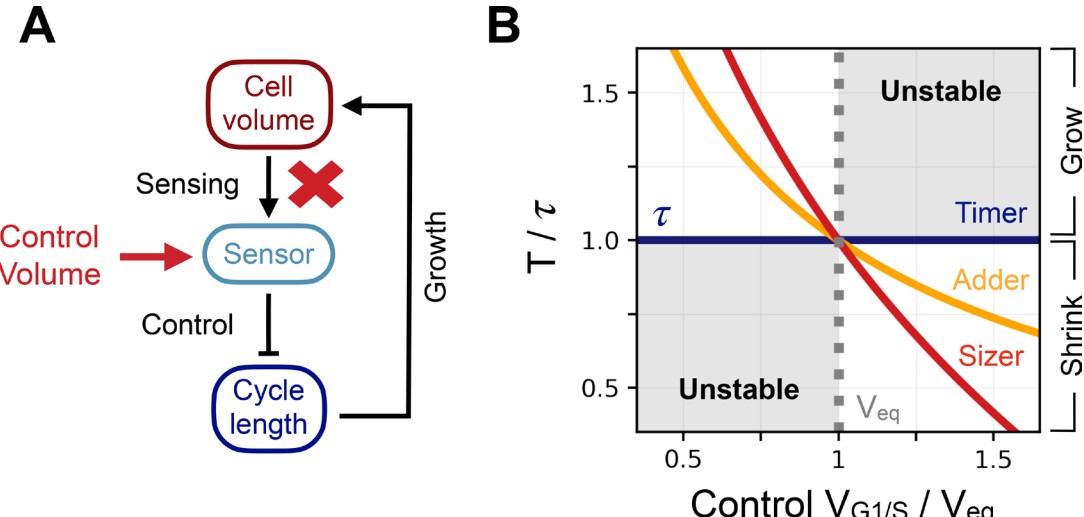

**Appendix 1—figure 2.** Control volume and archetype response curves. (**A**) Schematic representation of the way we break the feedback in the system and impose a control volume $V_C$ (red arrow) at the G1/S transition in order to record the induced cell cycle period $T(V_C)$. (**B**) Response curve of the 3 size control archetypes. X-coordinate is the control volume at G1/S $V_C$ normalized by the equilibrium volume $V_{eq}$. Y-coordinate is the response curve of the models $T(V_C)$ normalized by the doubling time $\tau$. The dark blue curve is the response curve for the timer of length $\tau$, the orange curve the response curve for the adder, and the red curve the response curve for the sizer. The dotted grey line indicates the equilibrium volume $V_{eq}$. The shaded region corresponds to the region where growth is unstable and volume diverges over successive generations.

The three archetypes' response curves are shown in *Appendix 1—figure 2B*. From these curves and given the particular cell cycle structures we examined, we can extract multiple relevant measures of size control such as the volume at birth, G1/S, and division from which we get the added volume during each phase of the cell cycle. It is noteworthy that the derivative of the added volumes $\Delta V$ with respect to the birth volume $V^{\text{Birth}}$ for the timers, adders, and sizers, are respectively 1, 0, and –1. Size control mechanisms are typically compared to the 3 archetypes by measuring the amount of added cell volumes over their cell cycles $\Delta V_{\text{Cycle}}$ and then fitting a linear model to these data points. The fitted slope of the linear model then informs what archetype this particular mechanism is more akin to. Some models evolved with added volume slopes lower than –1 and we call those super-sizers. Such mechanisms overcompensate for volume deviations about the equilibrium value which can increase variation in the size distribution instead of decreasing it.

## Response curve of the seed networks

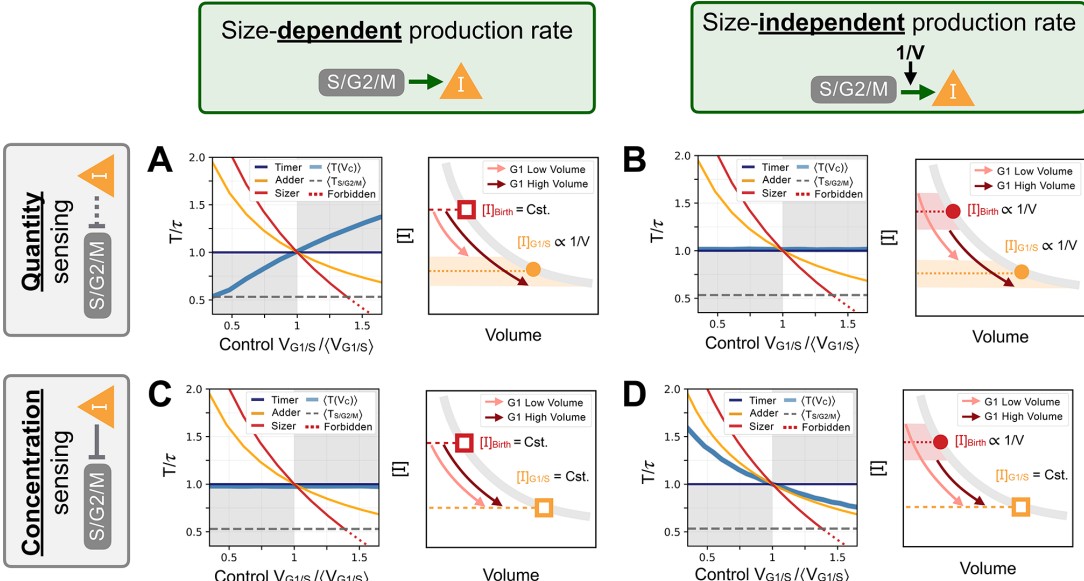

**Appendix 1—figure 3.** Response curves for the initial seed networks. Columns indicate the size scaling assumption of the protein production rates as indicated above the figure. Rows indicate quantity or concentration sensing of inhibitor $I$ at the G1/S transition assumption as indicated on the left side of the figure. In each panel, we first show the seed network's response curve $T(V_C)$ as a function of control volume $V_C$ at G1/S. Sizer (red), adder (orange) and timer (dark blue) archetypes are shown for comparison. Second, we provide a schematic representation of the $[\mathbf{I}]$ trajectory in G1. Schematic trajectories are shown for low volumes (light pink) and high volumes (dark red). (**A**) Quantity sensing of $I$ at G1/S with a size-dependent production rate in S/G2/M. Here, cells are born with a constant concentration $[\mathbf{I}]$. Because of the quantity sensing at G1/S, the concentration of $[\mathbf{I}]$ at G1/S scales as $1/V$. Thus, the time spent in G1 scales with $V$. This is the initial seed network we chose for most of our evolution experiments. (**B**) Quantity sensing of $I$ at G1/S with a size-independent production rate in S/G2/M. Here, cells are born with a concentration $[\mathbf{I}]$ at birth that scales as $1/V$. Because of the quantity sensing at G1/S, we again find that the concentration of $[\mathbf{I}]$ at G1/S scales as $1/V$. Thus, the time spent in G1 is constant. (**C**) Concentration sensing of $[\mathbf{I}]$ at G1/S with a size-dependent production rate in S/G2/M. Here, cells are born with a constant concentration $[\mathbf{I}]$. Because of the concentration sensing at G1/S, we find that the concentration of $[\mathbf{I}]$ at G1/S is constant. Thus, the time spent in G1 is constant. (**D**) Concentration sensing of $[\mathbf{I}]$ at G1/S with a size-independent production rate in S/G2/M. Here, cells are born with a concentration $[\mathbf{I}]$ that scales as $1/V$. Because of the concentration sensing at G1/S, we find that the concentration of $[\mathbf{I}]$ at G1/S is constant. Thus, the time spent in G1 scales as $1/V$.

In light of the control volume and response curve definitions from subsection 2.2, we can revisit the initial seed model and investigate how different assumptions alter the stability of growth and division in a cell lineage. Specifically, we investigate the size scaling assumption of the protein production rates and the concentration vs. quantity sensing of the transcriptional regulator $I$ at G1/S as previously described in Section 1. We summarize our results in *Appendix 1—figure 3*.

First, let us consider the G1 trajectory of the transcriptional regulator $[\mathbf{I}]$ who is solely produced during the S/G2/M timer. The dynamics of $[\mathbf{I}](t)$ in G1 will be described by the following equation:

$$[\mathbf{I}](t)|_{G1} = [\mathbf{I}]_0 e^{-(\delta+\lambda)t} \tag{14}$$

Here, the time variable $t$ represents the time since birth, $[\mathbf{I}]_0 = [\mathbf{I}](t = 0)$, $\delta$ is the protein's degradation rate and $\lambda$ is the growth rate of the cell volume. This equation holds until the G1/S transition where the S/G2/M Switch is turned on again and G1 ends. Because the degradation of the inhibitor does not yet depend on volume in any way, the time spent in G1 will only be dependent on the ratio between: (1) the initial condition at birth $[\mathbf{I}]_0$; (2) the final condition at the G1/S transition, $[\mathbf{I}]_{G1/S}$.

We found that the size scaling assumption of the protein production rates influences the initial condition at birth $[\mathbf{I}]_0$. When production rates scale with size, we find that $[\mathbf{I}]_0$ is independent of volume. This is expected as this assumption was chosen specifically to model proteins whose

concentrations are independent of size. Conversely, when we modify this assumption and consider that protein production rates are independent of size (e.g. like Whi5 in budding yeast), we find that the system produces a constant quantity of inhibitor instead of a constant concentration. This means that the initial concentration at birth $[\mathbf{I}]_0$ scales as $1/V$.

Similarly, when imposing quantity sensing of I at G1/S, we found that the concentration of inhibitor at G1/S $[\mathbf{I}]_{G1/S}$ scales as $1/V$. Finally, when imposing concentration sensing of I at G1/S, we found a constant concentration of inhibitor at G1/S $[\mathbf{I}]_{G1/S}$ as was expected by design.

Together, those assumptions alter the scaling of the duration of the G1 phase of the initial seed cycle. We summarize these results and present the models' response curves $T(V_C)$ in *Appendix 1—figure 3* where each row and column corresponds to a specific combination of assumptions. There, in *Appendix 1—figure 3A*, we see that for the combination of size-scaling production rate and quantity sensing at G1/S, we get a cell cycle period that is increasing with control volume $V_C$. This is undesirable and leads to unstable growth of the cell lineage towards 0 or $\infty$, but rewards the evolution of size control mechanisms that can prevent this unstable growth. We chose this initial seed model for most of our evolutionary simulations. In *Appendix 1—figure 3B,C*, we found that the two assumptions compensated each other to create size-independent timer models. The parameters of the network can be precisely fine-tuned to yield a response period of exactly $\tau$ as was done to produce the response curves shown here. Thus, it is technically possible to evolve a size control mechanism using these initial seed models, but we chose not to go down that path because we wanted to evolve an active size control mechanism. Finally, in *Appendix 1—figure 3D*, we see that if we assume that protein production rates do not scale with size and that the G1/S transition depends on the concentration of inhibitor $[\mathbf{I}]$, we get an initial seed model that already accomplishes size control as it displays a response curve that decreases with control volume $V_C$. This simple model loosely corresponds to the Whi5 inhibitor dilution model of budding yeast (*Schmoller et al., 2015*) where a constant quantity of inhibitor Whi5 is present at birth (and thus a concentration $[\text{Whi5}] \propto 1/V$) and is passively diluted in G1 until it reaches concentration threshold that triggers the G1/S transition.

## Evolutionary algorithm

Here we briefly describe the $\varphi$-evo evolutionary algorithm from *Henry et al., 2018* that we used to evolve size control networks. We refer the reader to the original publication's main text and supplementary material for a more thorough description of the algorithm. A schematic representation of the algorithm's architecture is shown in *Appendix 1—figure 4*.

First, an initial seed network is selected by the user as the starting point of the evolution simulation. $\varphi$-evo then clones this first individual to create a population of networks. At each epoch, mutations are randomly applied to the networks of the population. Those mutations vary from topological changes to the network, where biochemical species or interactions can be added or removed, to non-topological changes, where the networks' kinetic parameter values are modified. Following mutations, networks are ranked based on their performance at accomplishing the biological function we select for. This performance is encoded via a user-defined fitness function that is problem specific. We give details about the specific implementation of the fitness functions for cell size control in the following subsection. After ranking the networks, $\varphi$-evo proceeds to select the most fit half of the network population. The less fit half is then discarded and replaced by a copy of the most fit half to maintain a constant population size. With this, $\varphi$-evo completes the first epoch of the evolutionary process. We use the term *epoch* here rather than the term 'generation' which we retain to describe a cell lineage. A predetermined number of epochs of mutation and selection are then performed after which a final population of networks is extracted.

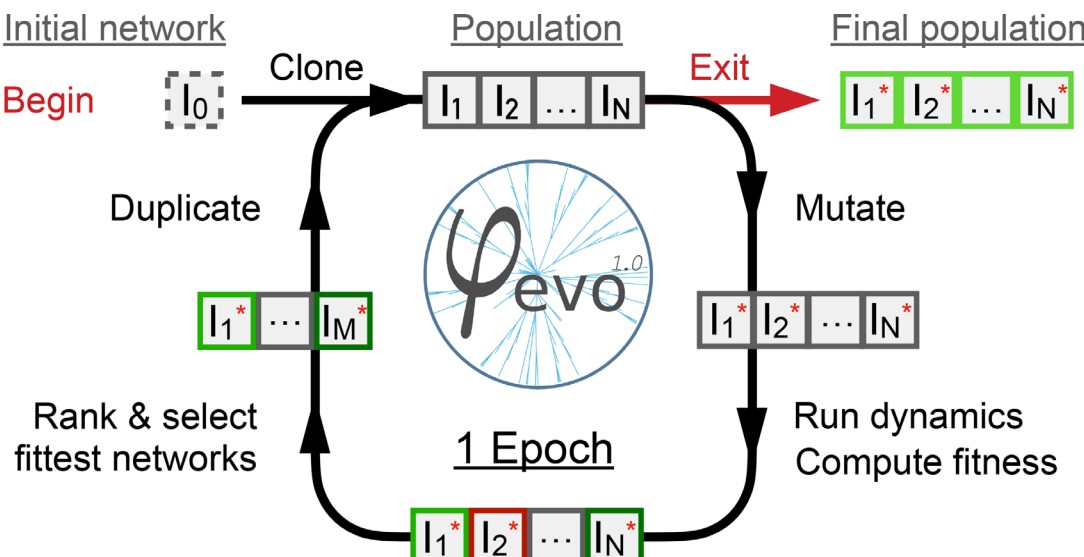

**Appendix 1—figure 4.** Schematic representation of the $\varphi$-evo algorithm. We begin with a user-defined initial seed network as starting point of the evolutionary process. The seed network is cloned to give a first population of networks. Individuals are then mutated randomly given the mutation parameters of the run. The dynamics and fitness scores of the networks are then computed and ranked. The best half of the population is selected and retained and the rest are discarded. The best half is then duplicated to maintain a constant population size $N$. We then repeat these instructions for a predefined number of epochs, after which a final population of networks is extracted and analyzed.

## Fitness

In order to rank and select networks based on their performance at accomplishing a specific biological function, we design a specific objective function that we call fitness. Here, we chose a fitness function that could quantify a model's ability to produce many viable descendants during a fixed time period of length $t$. We initially considered a simple fitness function to be minimized by $\varphi$-evo, $f_0 = -N_{Div}$. Here $N_{Div}$ is the number of divisions or *generations* in a cell lineage produced during a total time period of length $t$. Noise at the G1/S transition and in the S/G2/M timer duration act as a source of variation in volume at each generation which needs to be controlled by the evolved networks in order to prevent the cell volumes from diverging. Thus, networks that perform size control display a high number of divisions $N_{Div}$. The cycle duration distribution of a size control network will be centered around the doubling time $\tau$ in order to promote stable growth. Thus, on average, we expect a fit network to exhibit a maximum number of $\max(N_{Div}) = t/\tau$ divisions during a simulation of length $t$. Note that this number is mostly independent of the volume range selected by the evolutionary simulation as the doubling time $\tau$ is independent of the initial volume of the cell at the beginning of each cycle. There is a small effect on $N_{Div}$ from the initial conditions chosen for the system of ODEs modelling the cell cycle, but this effect is mostly negligible as long as the total time period $t \gg \tau$.

In our first evolution experiments, we found that the single objective function $f_0 = -N_{Div}$ was insufficient alone to evolve a cycling network. A possible reason for this is that the fitness landscape in parameter space defined by $f_0$ is mostly flat far from the optimum and is difficult to navigate as it doesn't incrementally guide the evolution process towards a proper size control phenotype. Indeed, a network that does not perform size control will display very few divisions before diverging towards sizes of 0 or $\infty$. In contrast, a network that *does* performs some size control, even if performed badly, will display mostly stable growth with many divisions and the volume will not diverge over successive generations. There is thus an all or nothing effect with this fitness function. We found that optimization process would often get stuck in a local optima with a low number of divisions and could not find a path to the global optima of size control. Because of this, we chose to turn towards multi-objective Pareto optimization which aims to simultaneously optimize several fitness functions. The idea here is that an additional fitness function can guide the evolutionary process through a different path in parameter space and could allow the evolutionary procedure to escape local optima.

In the Pareto optimization framework, we consider $N$ equally important objective functions $\vec{f} = (f_0, f_1, ..., f_N)$. Assuming that fitness functions are to be maximized, we say that individual $i$ (strictly) dominates individual $j$ if and only if their fitness $\vec{f}^i$ and $\vec{f}^j$ are such that:

$$\forall\, k \in \{1, N\}, f_k^i \geq f_k^j \text{ and } \exists\, k' \in \{1, N\} \mid f_{k'}^i > f_{k'}^j$$

The algorithm then selects half of the population of the highest rank using a fitness sharing algorithm to maximize population diversity. We refer the reader to *Warmflash et al., 2012* for more details on this procedure.

For this project, we chose to limit the optimization process to two fitness functions. We chose $f_0 = -N_{div}$ as the first fitness function. We tested different measures of size control for the second objective function which are described in the following subsections. In most of the cases, we chose the coefficient of variation of the size distribution at birth $CV_{\mathrm{Birth}}$ to be minimized as the second fitness function. In any case, we typically run 10 independent realizations of a network's performance and compute the average fitness score over those runs to buffer variations in fitness scores.

## Residuals

We first tested a least squared residuals fitness function to be minimized by $\varphi$-evo. This function yielded successful evolutionary runs but was abandoned due to requiring a user-defined target volume $V_t$. Indeed, we wanted to avoid the bias where we could select for biochemical networks matching a specific volume range. We used the following equation for the fitness function with $V_{\mathrm{Birth}}^{(n)}$ indicating the cell volume at birth at the $n$-th generation in a lineage.

$$f_1 = \frac{1}{N_{\mathrm{Div}} V_t} \sqrt{\sum_{n=1}^{N_{\mathrm{Div}}} (V_{\mathrm{Birth}}^{(n)} - V_t)^2} \tag{15}$$

We nevertheless present the result of a successful evolution run using Pareto optimization of $N_{Div}$ and $f_1$ in *Appendix 1—figure 11* where we obtain a version of the feedback-based network topology of Model A1.

## Coefficient of variation

We then considered the coefficient of variation of the size distribution at birth ($CV_{\mathrm{Birth}}$) to be minimized by $\varphi$-evo. The $CV_{\mathrm{Birth}}$ is a measure of size control that normalizes the variance of the size distribution at birth with respect to its mean and is thus mostly insensitive to the absolute volume range of the cell. We chose this second fitness function in most of the Pareto optimization evolution experiments as described in the main text.

$$f_2 = CV_{\mathrm{Birth}} = \frac{\sqrt{\mathbf{Var}[V_{\mathrm{Birth}}]}}{\mathbf{E}[V_{\mathrm{Birth}}]} \tag{16}$$

## Added volume slope in G1

We also considered directly optimizing the fitted slope of the volume added in G1 as a function of volume at birth to reinforce the sizer behavior in G1. As described in the main text, given a series of volume values at birth and at G1/S, $V_{\mathrm{Birth}}^{(i)}$ and $V_{\mathrm{G1/S}}^{(i)}$ for $i \in \{1, N_{\mathrm{Div}}\}$, the added volumes in G1 are defined as:

$$\Delta V_{\mathrm{G1}}^{(i)} = V_{\mathrm{G1/S}}^{(i)} - V_{\mathrm{Birth}}^{(i)}$$

The best fit slope $m$ of a linear model $\Delta V_{\mathrm{G1}} = m \cdot V_{\mathrm{Birth}} + b$ has a closed-form equation which is given as a function of the lineage data directly:

$$m = \frac{N_{\mathrm{Div}} \cdot \left( \sum_{i=1}^{N_{\mathrm{Div}}} V_{\mathrm{Birth}}^{(i)} \cdot \Delta V_{\mathrm{G1}}^{(i)} \right) - \left( \sum_{i=1}^{N_{\mathrm{Div}}} V_{\mathrm{Birth}}^{(i)} \right) \cdot \left( \sum_{i=1}^{N_{\mathrm{Div}}} \Delta V_{\mathrm{G1}}^{(i)} \right)}{N_{\mathrm{Div}} \cdot \sum_{i=1}^{N_{\mathrm{Div}}} \left( V_{\mathrm{Birth}}^{(i)} \right)^2 - \left( \sum_{i=1}^{N_{\mathrm{Div}}} V_{\mathrm{Birth}}^{(i)} \right)^2} \tag{17}$$

A slope of −1 corresponds to a sizer, a slope of 0 to an adder and a slope of +1 corresponds to a timer. In order to directly optimize the size control mechanism in G1 and to bias towards sizer mechanisms and to keep the fitness values positive, we considered the following fitness function to be minimized by $\varphi$-evo:

$$f_3 = m + 1 \qquad (18)$$

## Fitness penalties

In order to keep biochemical concentrations and cell volumes at reasonable levels, we chose to bound volume growth to a range $V \in [V_{min}, V_{max}]$ with $V_{min} = 0.1$ and $V_{max} = 100$. To guide the evolution of size control and to have a well defined steady-state distribution of cell sizes, we chose to impose fitness penalties on networks that would see their volumes reach one of those bounds at some point during a run. This applies to all evolution experiments performed in this study, but we will describe the case of the $N_{Div}$ and $CV_{\text{Birth}}$ fitness functions as they were used most of the time during this study.

Firstly, if the volume reached $V_{min}$, we considered the cell too small and declared it dead. At that point, any further cycles would not contribute to fitness scores. With this penalty, networks are guided towards preventing or at least delaying the time at which the volume becomes too small for the cell to remain viable.

Similarly, when volume reaches $V_{max}$, we penalize the fitness functions but in a different way. Because volume growth is restricted to the domain below $V_{max}$, we set the growth rate $\lambda = 0$ when $V = V_{max}$. In this case, we sometimes see networks make use of this growth arrest and exploit this artificial feature. This phenomenon can sometimes be seen in computational evolution where digital mirages are often exploited by optimization processes (**Lehman et al., 2020**). Here, such exploitative networks have initially long cycle periods $T \gg \tau$ and can sometimes spend the majority of their cell cycle in this growth arrest phase. Over successive epochs, this period $T$ gets shortened to get more and more divisions to take place during a run, shortening the time spent in growth arrest at each cycle. Eventually, this optimization leads to a particular type of model that has very sloppy size control but that scores highly with the fitness functions because of the artificial growth arrest. Such networks develop a timer with period $T \approx \tau$ and use the artificial growth arrest to buffer any volume variation incurred from late G1/S transition or noisy S/G2/M duration. These networks are optimal from an $N_{Div}$ perspective since they exactly double their mass at each cycle and perform the same number of division cycles during a run as an actual size control network. They are also more than optimal from a $CV_{\text{Birth}}$ since their growth is always stopped at $V_{max}$. Thus, without fail, their birth volume $V_{\text{B}irth} = V_{max}/2$ and there is no variation at all in the distribution. This size control illusion is a global optimum of the 2D-fitness space and must thus be heavily penalized to prevent the optimization from selecting this phenotype. Thus, when $V = V_{max}$, we only count 70% of the divisions $N_{Div}$ which is sufficient to distinguish this artificial phenotype from actual size control mechanisms. Additionally, we penalize the $CV_{\text{Birth}}$ score by adding to it a penalty of +10. This is significantly different from the usual range of coefficient of variations which lie between 0 and 0.5 typically, and prevents the optimization from selecting the artificial phenotype.

The introduction of these fitness penalties improved the convergence rate of the $\varphi$-evo algorithm significantly. Specifically, we believe the penalty on $V_{max}$ being somewhat less severe than the one for $V_{min}$ improved the convergence rate drastically. This is probably because the initial cell cycle model chosen for the evolution runs exhibits unstable growth and inevitably sees the cell volume grow to $V_{max}$ or shrink to $V_{min}$ rapidly as shown in **Appendix 1—figure 3**. Thus, the fitness landscape surrounding the initial cell cycle model is quite flat and is difficult to navigate from an optimization perspective. Gradual improvements to the $N_{Div}$ fitness function at $V_{max}$ by spending less and less time in growth arrest seemed to have helped the optimization process. This guided the algorithm towards better size control models more often than if penalties were absent.

Overall, even if the penalties somewhat biased the evolutionary process into following a phenotypic trajectory, they improved the convergence rate of the evolutionary process dramatically to the point that we decided to keep them for all experiments.

## Biochemical interactions

Many 'inverse-approach' approaches in systems biology have focused on purely transcriptional networks (**Cotterell and Sharpe, 2010**; **François et al., 2007**; **Fujimoto et al., 2008**; **Ten Tusscher and Hogeweg, 2011**;), because they are generic, easier to study and can efficiently describe many biological dynamics (**Alon, 2007**). In this project, we extend the biochemical interactions available for evolution: we not only model transcriptional activation and repression but also include complexation also known as protein-protein interaction (PPI), and assume there passive degradation. Adding PPIs is especially crucial because they are well known to lead to non-linear effects (**Buchler and Cross,**

*2009*; *Buchler and Louis, 2008*) allowing for the simple implementation of complex dynamics such as genetic oscillations (*François and Hakim, 2005*) observed e.g. in circadian clocks (*François, 2005*), and such non-linear effects indeed play crucial roles for control in our evolved model. The equations for these interactions are presented in this subsection.

We model transcriptional activation of arbitrary network species Y's production from species X using the following Hill equation:

$$\text{Activation}_{X \to Y}([\mathbf{X}], t_{X:Y}, n_{X:Y}) = \frac{([\mathbf{X}]/t_{X:Y})^{n_{X:Y}}}{1+([\mathbf{X}]/t_{X:Y})^{n_{X:Y}}} \tag{19}$$

Similarly, repression of arbitrary species Y's production from species X is modeled via the following Hill equation:

$$\text{Repression}_{X \to Y}([\mathbf{X}], t_{X:Y}, n_{X:Y}) = \frac{1}{1+([\mathbf{X}]/t_{X:Y})^{n_{X:Y}}} \tag{20}$$

In these equations, $t_{X:Y}$ is the threshold required for $[\mathbf{X}]$ to activate or repress $[\mathbf{Y}]$ at 50% of its capacity and $n_{X:Y}$ is the Hill coefficient. While both types of interactions are modeled using Hill equations, their combined effects on a network species' dynamics is computed differently. Indeed, only the maximum of all the activations is accounted for whereas the inhibitions are multiplicative and all are accounted for. Additionally, we allow some species to be produced at a basal rate $b$ independent of any activator which counts as an additional activation.

Altogether using an example, assuming multiple species $\mathbf{X}_1, ..., \mathbf{X}_q$ activate the production of species Z while multiples species $\mathbf{Y}_1, ..., \mathbf{Y}_r$ repress it, and assuming that Z has a basal production rate $b_Z$ and a maximum production rate $p_Z$, then the total contribution of these interactions to the ODE for the dynamics of $[\mathbf{Z}]$ is given by:

$$\frac{d[\mathbf{Z}]}{dt} = \max \left[ p_Z \cdot \max \left[ \frac{([\mathbf{X}_1]/t_{X_1:Z})^{n_{X_1:Z}}}{1+([\mathbf{X}_1]/t_{X_1:Z})^{n_{X_1:Z}}}, ..., \frac{([\mathbf{X}_q]/t_{X_q:Z})^{n_{X_q:Z}}}{1+([\mathbf{X}_q]/t_{X_q:Z})^{n_{X_q:Z}}} \right], b_Z \right] \cdot \prod_{i=1}^{r} \frac{1}{1+([\mathbf{Y}_i]/t_{Y_i:Z})^{n_{Y_i:Z}}} \tag{21}$$

We use the law of mass-action to model PPIs and passive degradation. Specifically, if arbitrary species X and Y interact together and form a complex Z given a forward rate $k_f$ and a backwards rate $k_b$, then the contribution of these interactions to the ODE for the dynamics of the system will be given by:

$$\frac{d[\mathbf{X}]}{dt} = \frac{d[\mathbf{Y}]}{dt} = -\frac{d[\mathbf{Z}]}{dt} = -k_f[\mathbf{X}] \cdot [\mathbf{Y}] + k_b[\mathbf{Z}] \tag{22}$$

Lastly, all network species are assumed to be degraded at a passive rate. Thus, if an arbitrary species X is solely degraded with a rate $\delta_X$, then the dynamic equation for the dynamics of $[\mathbf{X}]$ will be given by:

$$\frac{d[\mathbf{X}]}{dt} = -\delta_X[\mathbf{X}] \tag{23}$$

## Analysis of sources of noise

As mentioned in the main text and in this Appendix, we chose a hierarchical way of introducing noise in the system, starting with the biggest contributing factor and incrementally adding additional sources of noise in subsequent analyses. We first included noise in the cell cycle phases, specifically in the timing of the G1/S transition and in the length of the S/G2/M phase. Then in the later parts, we introduced protein production noise modeled as Langevin noise.

In the simulations presented in the main text, we chose not to include noise in the growth rate and in the division ratio as the recorded noise level for in experiments for these measures is lower than that for the timing of the cell cycle and the protein concentration noise (*Di Talia et al., 2007*; *Newman et al., 2006*; *Zatulovskiy et al., 2020*). Nevertheless, those are crucial assumptions that we made that we chose to investigate in more details here.

In subsection S/G2/M noise, we investigate how the level of noise in S/G2/M affects the conclusions drawn in the main text. Then, in subsection Growth rate noise we do the same for noise in the growth rate and in subsection Division ratio noise for the division ratio.

*Appendix 1—table 1* shows the values of $CV_{Birth}$ of the three models presented in the main text compared to the values reported in the literature for budding yeast (*Di Talia et al., 2007*), fission

yeast (*Sveiczer et al., 1996*) and mouse epidermal stem cell grown in the animal (*Xie and Skotheim, 2020*).

**Appendix 1—table 1.** Coefficients of variation: models and experiments.

| Models | | Data | | |
|---|---|---|---|---|
| Name | $CV_{\text{Birth}}$ | Cell type | Time of size measure | $CV$ |
| A1 | 0.098 | Haploid budding yeast | Budding | 0.17 |
| A2 | 0.095 | Haploid fission yeast | Fission | 0.06 |
| B | 0.235 | Mouse epidermal stem cell | Birth | 0.17 |

## S/G2/M noise

Here, we perform similar evolution experiments to those reported in *Figure 4* of the main text to examine the effect of modulating the noise in the S/G2/M timer. We thus perform three independent experiment where we set the CV in the timer period to 0%, 5%, and 8% corresponding to no, medium, and high noise respectively. For reference, the CV of the timer period in the control condition where Model A1 was evolved is 3%. Note that we maintain the average duration of the timer to be about half the time it takes to double the cell's volume. Having specified the S/G2/M timer parameters and starting from the initial seed network of Model A1, we perform evolution and select networks as previously. We compare ensembles of 60 networks for each noise level, half of them evolved under the Pareto optimization of $N_{\text{Div}}$ and $CV_{\text{Birth}}$ and the other half under the single objective optimization of $N_{\text{Div}}$. The results are shown in *Appendix 1—figure 5*.

Increasing the noise, progressively leads to a loss of the sizer signature and increases the $CV_{\text{Birth}}$. This is likely because the fixed duration of S/G2/M allows the system to accurately reset protein concentrations for the subsequent cell cycle to promote accurate G1 control (*Willis et al., 2020*). Thus, an increasing level of noise becomes associated with a worse accuracy in the size control mechanism which leads to loss of the sizer signature and increased $CV_{\text{Birth}}$ as can be seen in *Appendix 1—figure 5D,E*. Other results from the main text remain unchanged.

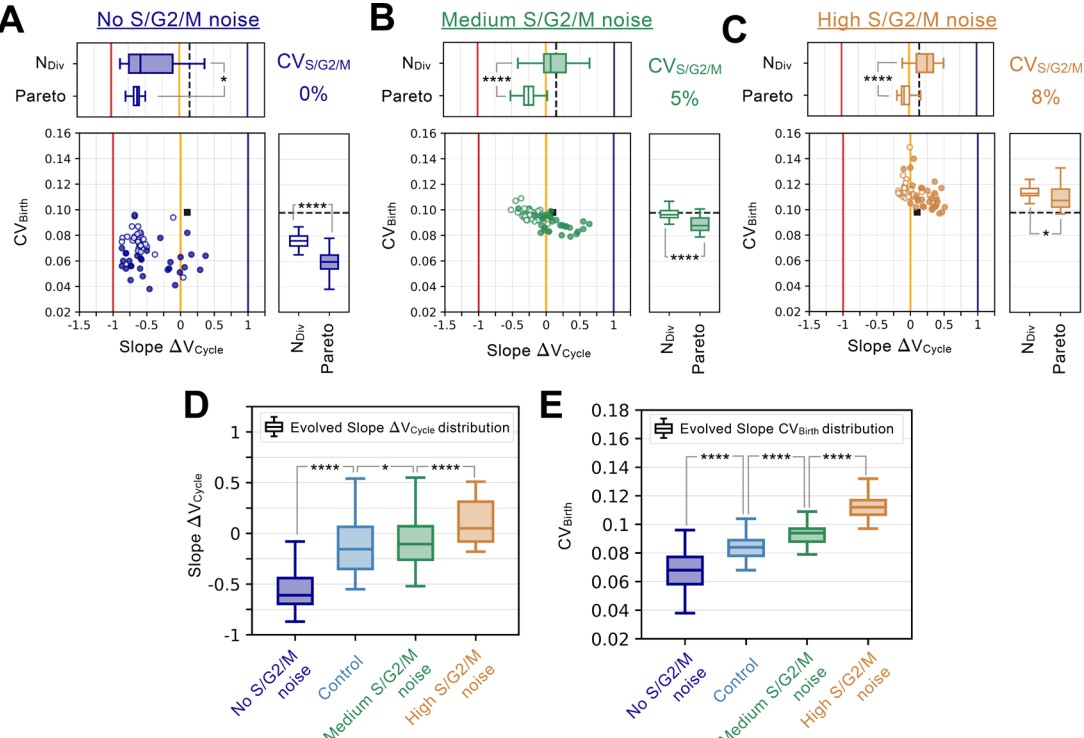

**Appendix 1—figure 5.** S/G2/M noise analysis. Summary statistics for evolutionary simulations each having 500 epochs. Model A1 was used as the initial seed network. 30 simulations were performed using Pareto optimization

*Appendix 1—figure 5 continued*

of the number of divisions ($N_{\text{Div}}$) and the CV of cell size at birth ($CV_{\text{Birth}}$), are labeled Pareto and are shown in full colors. 30 more simulations were performed using only the number of divisions as the fitness function, are labeled $N_{\text{Div}}$ and are shown in colored outlines only. Scatter plots show the coefficient of variation of the size distribution at birth ($CV_{\text{Birth}}$, Y-coordinate) as a function of the fitted added volume slope over the whole cycle as a function of volume at birth (Slope $\Delta V_{\text{Cycle}}$, X-coordinate) for the most fit models evolved during each of the 60 independent simulations. Horizontal box plots above the scatter plots in A-C display the distributions of the added volume slopes for the Pareto and $N_{\text{Div}}$ simulations. Timer (dark blue), adder (orange) and sizer (red) slopes are shown respectively at 1, 0, and –1 for comparison. Vertical box plots on the right of the scatter plots in A-C show the distributions of $CV_{\text{Birth}}$ for the Pareto and $N_{\text{Div}}$ simulations. Asterisks represent p-values for the Welch's t-Test between the distributions. For reference, *NS* indicates $p > 0.05$, * indicates $p < 0.05$, ** indicates $p < 10^{-2}$, *** indicates $p < 10^{-3}$ and **** indicates $p < 10^{-4}$. The values of $CV_{\text{Birth}}$ and Slope $\Delta V_{\text{Cycle}}$ for the initial seed Model A1 are shown as a black square in the scatter plot or as a dashed black line in the box plots. Each panel explores different S/G2/M noise levels. (**A**) Evolution results for no noise in S/G2/M duration. (**B**) Evolution results for a noise level in S/G2/M duration equal to 5%. (**C**) Evolution results for a noise level in S/G2/M duration equal to 8%. (**D**) Evolved Slope $\Delta V_{\text{Cycle}}$ distributions as a function of noise level in S/G2/M. For reference, noise level for the Control experiment from *Figure 4* corresponds to 3%. Box plots in D-E represent the distributions for both the Pareto and $N_{\text{Div}}$ evolution experiments. Here, increased S/G2/M noise leads to loss of the sizer signature. (**E**) Evolved $CV_{\text{Birth}}$ distributions as a function of noise level in S/G2/M. Here, increased S/G2/M noise leads to increased variability in the cell size distributions at birth.

## Growth rate noise

Here, we perform similar evolution experiments as we did for the noise in S/G2/M but this time by adding noise in the growth rate $\lambda$. Specifically, at each generation of a cell's lineage, we sample a growth rate from a Gaussian distribution centered around 0.25, the initial value we used in the rest of this project. We perform three evolution experiments with coefficient of variations for the growth rate distributions set to 3%, 5% and 8% corresponding to low, medium and high noise respectively. We perform 30 independent evolution runs with the Pareto optimization framework for each $\lambda$ noise level, each of them starting from the initial seed network of Model A1. The results are shown in *Appendix 1—figure 6*.

We note that on average, the growth rate will remain centered around the same value, thus not affecting the optimal fitness score networks are able to achieve. However, individual variations at each cycle perturb the ability of the system of accomplishing size control by always modifying the doubling time $\tau = \ln(2)/\lambda$. This leads to progressive loss of the sizer signature and also increases the $CV_{\text{Birth}}$. This increased noise in the system sometimes sends the cell volume towards 0 or $\infty$ as volume is kicked outside of the control mechanism's working range. Since this behavior is highly penalized in the fitness functions scoring, the evolution finds a way to prevent this from happening via different strategies. Interestingly, at higher noise levels, strong sizers can still evolve but are not the most common phenotype. Instead, evolution seems to favor timers that reliably ensure timely cell division generation after generation. There is however a trade-off and these models exhibit a higher $CV_{\text{Birth}}$ due to the lower amount of size control. Adders can also be evolved at all tested noise levels and provide good size control with reliably low $CV_{\text{Birth}}$.

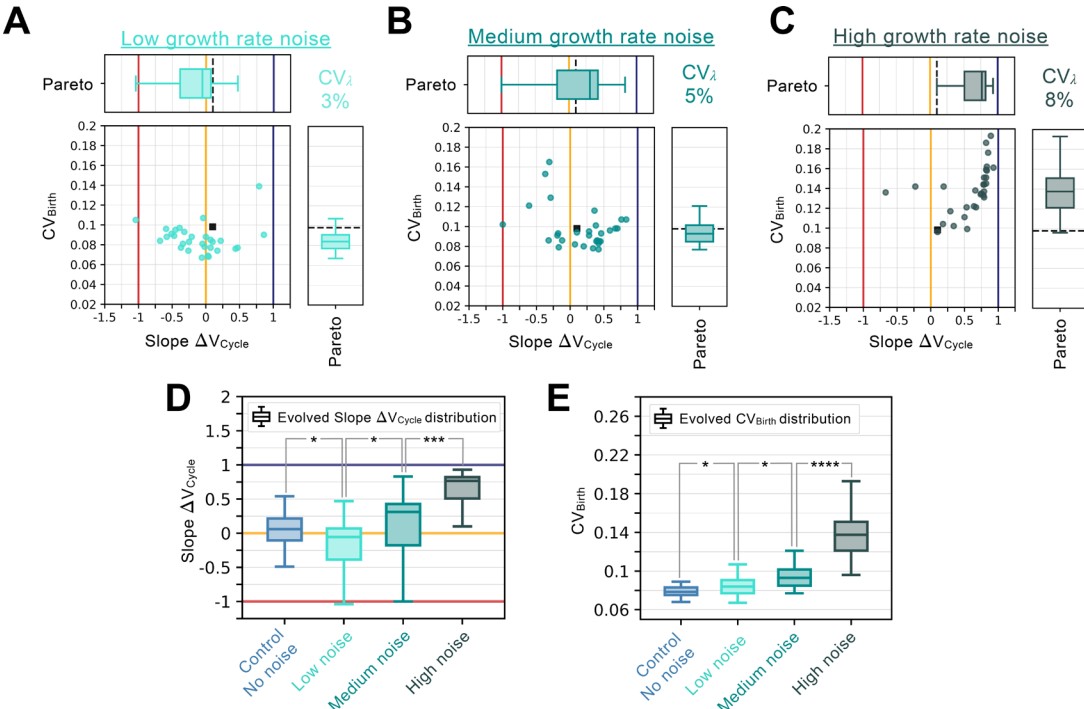

**Appendix 1—figure 6.** Growth rate noise analysis. Summary statistics for evolutionary simulations each having 500 epochs. Model A1 was used as the initial seed network. Only 30 simulations were performed using Pareto optimization of the number of divisions ($N_{\mathrm{Div}}$) and the CV of cell size at birth ($CV_{\mathrm{Birth}}$), are labeled Pareto. Scatter plots show the coefficient of variation of the size distribution at birth ($CV_{\mathrm{Birth}}$, Y-coordinate) as a function of the fitted added volume slope over the whole cycle as a function of volume at birth (Slope $\Delta V_{\mathrm{Cycle}}$, X-coordinate) for the most fit models evolved during each of the 30 independent simulations. Horizontal box plots above the scatter plots in A-C display the distributions of the added volume slopes. Timer (dark blue), adder (orange) and sizer (red) slopes are shown respectively at 1, 0, and –1 for comparison. Vertical box plots on the right of the scatter plots in A-C show the distributions of $CV_{\mathrm{Birth}}$. Asterisks represent p-values for the Welch's t-Test between the distributions. For reference, *NS* indicates $p > 0.05$, * indicates $p < 0.05$, ** indicates $p < 10^{-2}$, *** indicates $p < 10^{-3}$ and **** indicates $p < 10^{-4}$. The values of $CV_{\mathrm{Birth}}$ and Slope $\Delta V_{\mathrm{Cycle}}$ for the initial seed Model A1 are shown as a black square in the scatter plot or as a dashed black line in the box plots. Each panel explores different growth rate noise levels. (**A**) Evolution results for low noise in growth rate with associated coefficient of variation at 3%. (**B**) Evolution results for medium noise in growth rate with associated coefficient of variation at 5%. (**C**) Evolution results for high noise in growth rate with associated coefficient of variation at 8% (**D**) Evolved Slope $\Delta V_{\mathrm{Cycle}}$ distributions as a function of noise level in the growth rate. For reference, noise level for the Control experiment from **Figure 4** corresponds to no noise. Here, increased growth rate noise leads to rapid loss of the sizer signature. (**E**) Evolved $CV_{\mathrm{Birth}}$ distributions as a function of noise level in the growth rate. Here, increased growth rate noise leads to increased variability in the cell size distributions at birth.

## Division ratio noise

Here, we perform similar evolution experiments as we did for the noise in S/G2/M and in $\lambda$, but this time by adding noise in the division fraction $f$. Specifically, at each generation of a cell's lineage, we sample $f$ from a Gaussian distribution centered around 2, the initial value we used in the rest of this project for symmetrical divisions. We perform three evolution experiments with coefficient of variations for the growth rate distributions set to 2%, 4% and 8% corresponding to low, medium and high noise respectively. We perform 30 independent evolution runs with the Pareto optimization framework for each $f$ noise level, each of them starting from the initial seed network of Model A1. The results are shown in **Appendix 1—figure 7**.

The results of this experiment are very similar to those for noise in the growth rate $\lambda$ described in the previous subsection. Indeed, changing the division fraction $f$ does not change the doubling time $\tau$ directly like for $\lambda$. Instead, it changes the cycle period around which cells see their volume shrink or grow over successive generations. Indeed, for a division fraction $f$, this equilibrium time between shrinking and growth becomes $\tau_f = \ln(f)/\lambda$. Intuitively, if $f$ is bigger than 2, then cells need to spend

a little bit more time in their cell cycles for growth to occur in order to compensate for this increased division fraction. This increased noise in $f$ leads to progressive loss of the sizer signature as seen before and also increases the $CV_{\text{Birth}}$. The division fraction noise has a big effect on premature cell death. Indeed, at the highest noise level tested where $CV_f = 8\%$, we saw many runs end with premature cell death to the point where three evolution runs were completely unable to find a mechanism able to prevent this. As a result of this strong pressure to avoid premature cell death, evolution turns once again to timers instead of sizers in the noisier regime. As before, adders seem to be the most reliable size control phenotype exhibiting low $CV_{Birth}$.

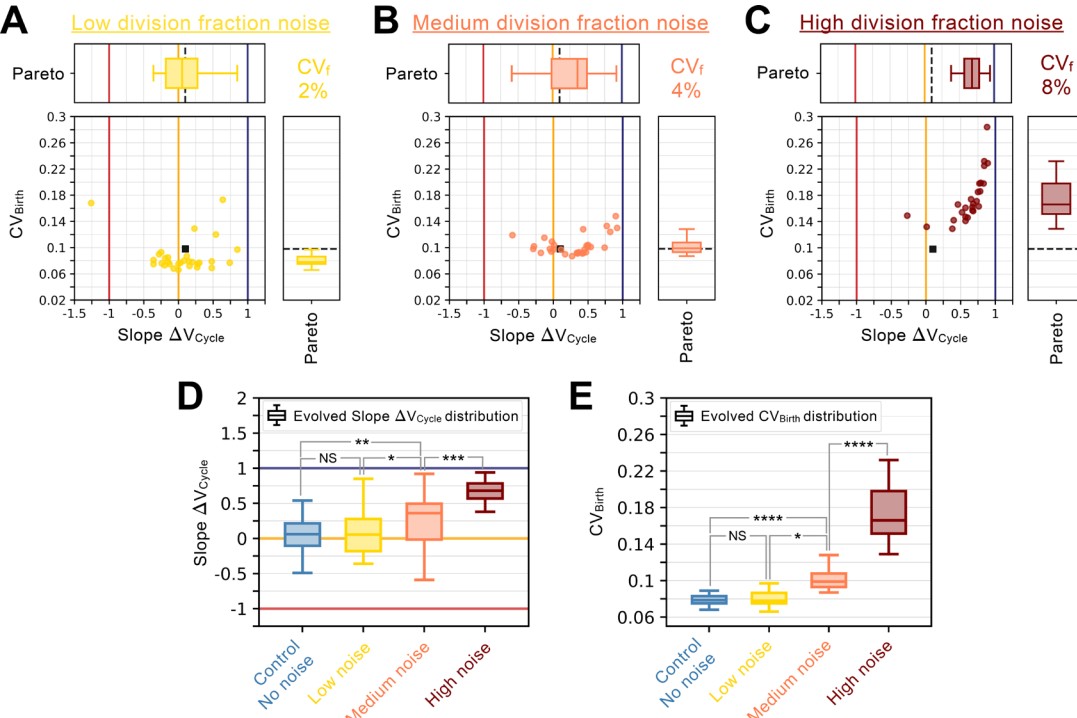

**Appendix 1—figure 7.** Division fraction noise analysis. Summary statistics for evolutionary simulations each having 500 epochs. Model A1 was used as the initial seed network. Only 30 simulations were performed using Pareto optimization of the number of divisions ($N_{\text{Div}}$) and the CV of cell size at birth ($CV_{\text{Birth}}$), are labeled Pareto. Scatter plots show the coefficient of variation of the size distribution at birth ($CV_{\text{Birth}}$, Y-coordinate) as a function of the fitted added volume slope over the whole cycle as a function of volume at birth (Slope $\Delta V_{\text{Cycle}}$, X-coordinate) for the most fit models evolved during each of the 30 independent simulations. Horizontal box plots above the scatter plots in A-C display the distributions of the added volume slopes. Timer (dark blue), adder (orange) and sizer (red) slopes are shown respectively at 1, 0, and –1 for comparison. Vertical box plots on the right of the scatter plots in A-C show the distributions of $CV_{\text{Birth}}$. Asterisks represent p-values for the Welch's t-Test between the distributions. For reference, *NS* indicates $p > 0.05$, * indicates $p < 0.05$, ** indicates $p < 10^{-2}$, *** indicates $p < 10^{-3}$ and **** indicates $p < 10^{-4}$. The values of $CV_{\text{Birth}}$ and Slope $\Delta V_{\text{Cycle}}$ for the initial seed Model A1 are shown as a black square in the scatter plot or as a dashed black line in the box plots. Each panel explores different division fraction $f$ noise levels. (**A**) Evolution results for low noise in division fraction with associated coefficient of variation at 2%. (**B**) Evolution results for medium noise in division fraction with associated coefficient of variation at 4%. (**C**) Evolution results for high noise in division fraction with associated coefficient of variation at 8%. 27/30 evolution runs succeeded and are shown here. (**D**) Evolved Slope $\Delta V_{\text{Cycle}}$ distributions as a function of noise level in the division fraction. For reference, noise level for the Control experiment from *Figure 4* corresponds to no noise. Here, increased noise leads to rapid loss of the sizer signature. (**E**) Evolved $CV_{\text{Birth}}$ distributions as a function of noise level in the division fraction. Here, increased noise leads to increased variability in the cell size distributions at birth.

## Model descriptions

In this section, we give the full set of equations and parameter values of the models from the main text. We remind the reader that we scale all our variables so that a concentration of one arbitrary

unit corresponds roughly to 1000 proteins in a 100fL cell (**Milo et al., 2010**). Additionally, we scale the time variable such that 1 arbitrary time unit corresponds roughly to 30 min (**Di Talia et al., 2007**).

## Model A1

The parameter values of the Model A1 are shown in **Appendix 1—table 2** along with the corresponding differential equations in **Equation 24**.

**Appendix 1—table 2.** Model A1 parameter values.

| Parameter | Value | Parameter | Value |
|---|---|---|---|
| $p_2$ | 0.369408 | $\delta_3$ | 0.4574764 |
| $t_{1:2}$ | 1.104619 | $k_{f1}$ | 6.783001 |
| $n_{1:2}$ | 3.215727 | $k_{b1}$ | 0.195168 |
| $\delta_2$ | 0.083827 | $\delta_4$ | 4.244007 |
| $p_3$ | 0.703658 | $k_{f2}$ | 0.021174 |
| $t_{2:3}$ | 0.044938 | $k_{b2}$ | 1.075146 |
| $n_{2:3}$ | 3.266732 | $\delta_5$ | 1.1010876 |

$$
\begin{aligned}
\frac{d[\mathbf{I}]}{dt} &= p_2 \frac{(\mathbf{S/G2/M\ Switch}/t_{1:2})^{n_{1:2}}}{1+(\mathbf{S/G2/M\ Switch}/t_{1:2})^{n_{1:2}}} - k_{f1}[\mathbf{I}] \cdot [\mathbf{R}] + k_{b1}[\mathbf{S}_4] - k_{f2}[\mathbf{I}]^2 + k_{b2}[\mathbf{S}_5] \\
&\quad -(\delta_2 + \lambda)[\mathbf{I}] \\
\frac{d[\mathbf{R}]}{dt} &= p_3 \frac{1}{1+([\mathbf{I}]/t_{2:3})^{n_{2:3}}} - k_{f1}[\mathbf{I}] \cdot [\mathbf{R}] + k_{b1}[\mathbf{S}_4] - (\delta_3 + \lambda)[\mathbf{I}] \\
\frac{d[\mathbf{S}_4]}{dt} &= k_{f1}[\mathbf{I}] \cdot [\mathbf{R}] - k_{b1}[\mathbf{S}_4] - (\delta_4 + \lambda)[\mathbf{S}_4] \\
\frac{d[\mathbf{S}_5]}{dt} &= k_{f2}[\mathbf{I}]^2 - k_{b2}[\mathbf{S}_5] - (\delta_5 + \lambda)[\mathbf{S}_5]
\end{aligned}
\tag{24}
$$

## Model A2

The parameter values of the Model A2 are shown in **Appendix 1—table 3** along with the corresponding differential equations in **Equation 25**.

**Appendix 1—table 3.** Model A2 parameter values.

| Parameter | Value | Parameter | Value |
|---|---|---|---|
| $p_2$ | 1.915601 | $n_{1:3}$ | 2.751652 |
| $t_{1:2}$ | 0.17872 | $t_{2:3}$ | 0.441106 |
| $n_{1:2}$ | 2.09054 | $n_{2:3}$ | 2.780297 |
| $t_{3:2}$ | 0.962612 | $\delta_3$ | 0.045051 |
| $t_{3:2}$ | 2.144666 | $k_{f1}$ | 0.839879 |
| $\delta_2$ | 0.019495 | $k_{b1}$ | 0.381067 |
| $p_3$ | 1.944803 | $\delta_4$ | 0.913992 |
| $t_{1:3}$ | 0.422939 | | 0 |

$$
\begin{aligned}
\frac{d[\mathbf{I}]}{dt} &= p_2 \frac{(\mathbf{S/G2/M\ Switch}/t_{1:2})^{n_{1:2}}}{1+(\mathbf{S/G2/M\ Switch}/t_{1:2})^{n_{1:2}}} \frac{1}{1+([\mathbf{R}]/t_{3:2})^{n_{3:2}}} - k_{f1}[\mathbf{I}] \cdot [\mathbf{R}] + k_{b1}[\mathbf{S}_4] \\
&\quad -k_{f2}[\mathbf{I}]^2 + k_{b2}[\mathbf{S}_5] - (\delta_2 + \lambda)[\mathbf{I}] \\
\frac{d[\mathbf{R}]}{dt} &= p_3 \frac{1}{1+([\mathbf{I}]/t_{2:3})^{n_{2:3}}} \frac{1}{1+(\mathbf{S/G2/M\ Switch}/t_{1:3})^{n_{1:3}}} - k_{f1}[\mathbf{I}] \cdot [\mathbf{R}] + k_{b1}[\mathbf{S}_4] \\
&\quad -(\delta_3 + \lambda)[\mathbf{I}] \\
\frac{d[\mathbf{S}_4]}{dt} &= k_{f1}[\mathbf{I}] \cdot [\mathbf{R}] - k_{b1}[\mathbf{S}_4] - (\delta_4 + \lambda)[\mathbf{S}_4]
\end{aligned}
\tag{25}
$$

## Model B

The parameter values of the fluctuation-sensing Model B are shown in *Appendix 1—table 4* along with the corresponding stochastic differential equations in *Equation 26*. We note that the $C_0$ parameter in the noise terms of the equation is a concentration scaling factor.

**Appendix 1—table 4.** Model B parameter values.

| Parameter | Value | Parameter | Value |
|---|---|---|---|
| $p_2$ | 4.968896 | $n_{3:3}$ | 3.189190 |
| $t_{1:2}$ | 2.569361 | $\delta_3$ | 0.246561 |
| $n_{1:2}$ | 0.952178 | $p_4$ | 4.674459 |
| $t_{3:2}$ | 0.131057 | $\delta_4$ | 1.010978 |
| $n_{3:2}$ | 9.219961 | $k_f$ | 3.194116 |
| $\delta_2$ | 0.521437 | $k_b$ | 4.634009 |
| $p_3$ | 0.113586 | $\delta_5$ | 1.165439 |
| $t_{3:3}$ | 2.183079 | $C_0$ | 1 |

$$
\begin{aligned}
\frac{d[\mathbf{I}]}{dt} &= p_2 \cdot \max\left( \frac{(\mathbf{S/G2/M\ Switch}/t_{1:2})^{n_{1:2}}}{1+(\mathbf{S/G2/M\ Switch}/t_{1:2})^{n_{1:2}}}, \frac{([\mathbf{A}]/t_{3:2})^{n_{3:2}}}{1+([\mathbf{A}]/t_{3:2})^{n_{3:2}}} \right) - (\delta_2 + \lambda)[\mathbf{I}] \\
&\quad + \mathcal{N}_1(0,1) \cdot \sqrt{\frac{p_2 \cdot \max\left( \frac{(\mathbf{S/G2/M\ Switch}/t_{1:2})^{n_{1:2}}}{1+(\mathbf{S/G2/M\ Switch}/t_{1:2})^{n_{1:2}}}, \frac{([\mathbf{A}]/t_{3:2})^{n_{3:2}}}{1+([\mathbf{A}]/t_{3:2})^{n_{3:2}}} \right) + \delta_2[\mathbf{I}]}{V \cdot C_0 \cdot \Delta t}} \\
\frac{d[\mathbf{A}]}{dt} &= p_3 \cdot \frac{([\mathbf{A}]/t_{3:3})^{n_{3:3}}}{1+([\mathbf{A}]/t_{3:3})^{n_{3:3}}} - k_f[\mathbf{A}] \cdot [\mathbf{S}_4] + k_b[\mathbf{S}_5] - (\delta_3 + \lambda)[\mathbf{I}] \\
&\quad + \mathcal{N}_2(0,1) \cdot \sqrt{\frac{p_3 \cdot \frac{([\mathbf{A}]/t_{3:3})^{n_{3:3}}}{1+([\mathbf{A}]/t_{3:3})^{n_{3:3}}} + \delta_3[\mathbf{A}]}{V \cdot C_0 \cdot \Delta t}} - \mathcal{N}_3(0,1) \cdot \sqrt{\frac{k_f[\mathbf{A}] \cdot [\mathbf{S}_4]}{V \cdot C_0 \cdot \Delta t}} \\
&\quad + \mathcal{N}_4(0,1) \cdot \sqrt{\frac{k_b[\mathbf{S}_5]}{V \cdot C_0 \cdot \Delta t}} \\
\frac{d[\mathbf{S}_4]}{dt} &= p_4 - k_f[\mathbf{A}] \cdot [\mathbf{S}_4] + k_b[\mathbf{S}_5] - (\delta_4 + \lambda)[\mathbf{S}_4] - \mathcal{N}_3(0,1) \cdot \sqrt{\frac{k_f[\mathbf{A}] \cdot [\mathbf{S}_4]}{V \cdot C_0 \cdot \Delta t}} \\
&\quad + \mathcal{N}_4(0,1) \cdot \sqrt{\frac{k_b[\mathbf{S}_5]}{V \cdot C_0 \cdot \Delta t}} + \mathcal{N}_5(0,1) \cdot \sqrt{\frac{p_4 + \delta_4[\mathbf{S}_4]}{V \cdot C_0 \cdot \Delta t}} \\
\frac{d[\mathbf{S}_5]}{dt} &= k_f[\mathbf{A}] \cdot [\mathbf{S}_4] - k_b[\mathbf{S}_5] - (\delta_5 + \lambda)[\mathbf{S}_5] + \mathcal{N}_3(0,1) \cdot \sqrt{\frac{k_f[\mathbf{A}] \cdot [\mathbf{S}_4]}{V \cdot C_0 \cdot \Delta t}} \\
&\quad - \mathcal{N}_4(0,1) \cdot \sqrt{\frac{k_b[\mathbf{S}_5]}{V \cdot C_0 \cdot \Delta t}} + \mathcal{N}_6(0,1) \cdot \sqrt{\frac{\delta_5[\mathbf{S}_5]}{V \cdot C_0 \cdot \Delta t}}
\end{aligned}
\tag{26}
$$

## Additional models

Here we present additional models that were evolved but not discussed in the main text to provide more examples of size control mechanisms.

### Model A3

Model A3 is similar to Model A1, but lacks the homodimerization interaction of $I$ (see *Equation 24*). This specific model results in a weak adder/timer that displays a non-linear size control response curve. This is similar to Model A2 where we see a sizer/adder behavior in the low control volume regime and adder/timer behavior in the high control volume regime.

This model's behavior is summarized in *Appendix 1—figure 8*. The parameter values of the model are shown in *Appendix 1—table 5* along with the corresponding differential equations in *Equation 27*. This model was evolved with the Pareto fitness optimization framework to maximize $N_{Div}$ and minimize $CV_{\text{Birth}}$. The initial seed model topology for this evolutionary simulation was the quantity sensing oscillator shown in *Appendix 1—figure 3A* and was optimized during 3000 epochs.

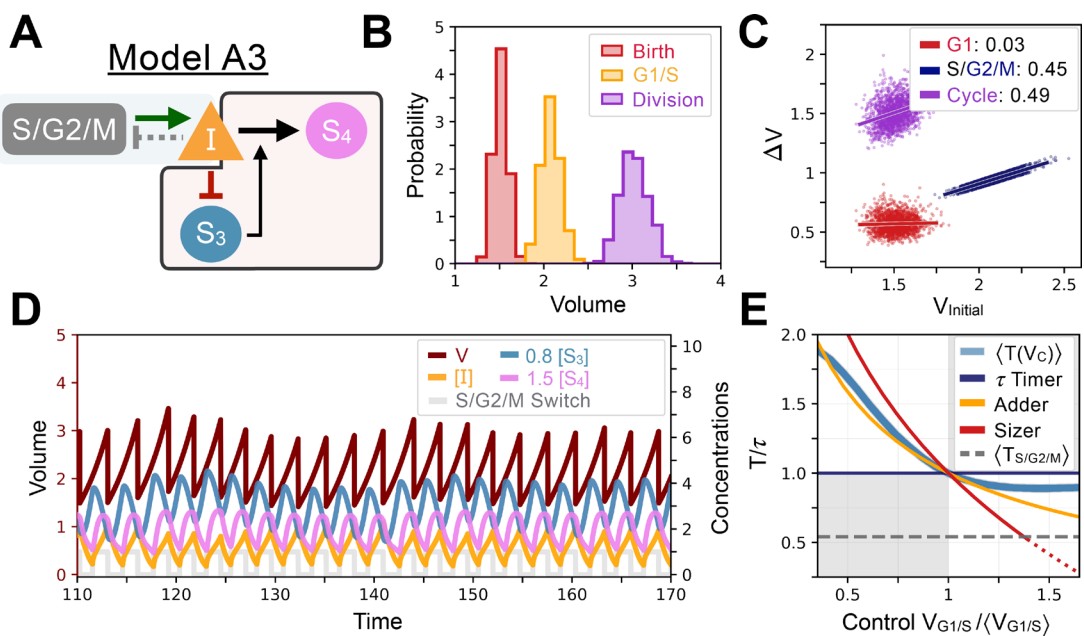

**Appendix 1—figure 8.** Model A3's behavior. (**A**) Network topology of the evolved Model A3. $S_3$ is as a size sensor and titrates $I$ in a size-dependent manner. (**B**) Size distributions at birth (red), G1/S (orange), and division (purple). (**C**) Added volumes in G1 (red), S/G2/M (blue) and over the whole cycle (purple) as a function of the volume at the beginning of those phases. (**D**) Temporal dynamics of the model, colors correspond to the variables in A. (**E**) Response curve $T(V_C)$ as a function of control volume $V_C$ at the G1/S transition.

**Appendix 1—table 5.** Model A3 parameter values.

| Parameter | Value | Parameter | Value |
|---|---|---|---|
| $p_2$ | 5.606156 | $t_{2:3}$ | 1.410420 |
| $t_{1:2}$ | 0.066136 | $n_{2:3}$ | 3.434213 |
| $n_{1:2}$ | 8.177965 | $k_f$ | 1.930909 |
| $b_2$ | 0.911279 | $k_b$ | 3.871610 |
| $\delta_2$ | 0.257920 | $\delta_3$ | 0.013294 |
| $p_3$ | 5.887584 | $\delta_4$ | 1.803926 |

$$
\begin{aligned}
\frac{d[\mathbf{I}]}{dt} &= \max\left(p_2 \cdot \frac{(\mathbf{S/G2/M\ Switch}/t_{1:2})^{n_{1:2}}}{1+(\mathbf{S/G2/M\ Switch}/t_{1:2})^{n_{1:2}}}, b_2\right) - k_f[\mathbf{I}] \cdot [\mathbf{S}_3] + k_b[\mathbf{S}_4] \\
&\quad - (\delta_2 + \lambda)[\mathbf{I}] \\
\frac{d[\mathbf{S}_3]}{dt} &= \frac{p_3}{1+(\mathbf{I}/t_{2:3})^{n_{2:3}}} - k_f[\mathbf{I}] \cdot [\mathbf{S}_3] + k_b[\mathbf{S}_4] - (\delta_3 + \lambda)[\mathbf{S}_3] \\
\frac{d[\mathbf{S}_4]}{dt} &= k_f[\mathbf{I}] \cdot [\mathbf{S}_3] - k_b[\mathbf{S}_4] - (\delta_4 + \lambda)[\mathbf{S}_4]
\end{aligned}
\tag{27}
$$

## Model A4

Model A4 is similar in essence to Model A1, albeit more unstable. In this model, $\mathbf{S}_4$ is the size sensor. Instead of using a PPI to titrate $I$ in a size-dependent manner in G1, this model leverages a transcriptional repression to modulate the production of $I$ directly rather than its effective degradation. The model's behavior is summarized in *Appendix 1—figure 9*. The parameter values of the model are shown in *Appendix 1—table 6* along with the corresponding differential equations in *Equation 28*. Notably, *Appendix 1—figure 9E* shows that the model's response curve in the low control volume regime is ill-defined. In this model specifically, when the low volume regime is reached, the concentration of inhibitor $I$ at G1/S increases due to quantity sensing $[I] \propto 1/V$. Then, because of the homodimerization of $I$ into $S_3$, we see the concentration $[S_3]$ also rise. We

can see both of these curves spike up momentarily in the trajectories of *Appendix 1—figure 9D* around $t \approx 136$. The problem arises if this increase is too strong. Then, $S_3$ activates the production of additional $I$, kick-starting a positive feedback loop that creates more and more inhibitor $I$, effectively interrupting the oscillator and making the period ill-defined. In practice, due to intrinsic noise at the G1/S transition and in the S/G2/M timer length, we've seen a cell lineage terminate prematurely before it can reach the maximum number of $N_{Div}$ allowed in a simulation because of this problem. One could say this model is thus less fit than those presented in the main text, although it displays an added volume slope over its cycle of –0.41 and is close to a sizer.

This model was evolved using a Pareto fitness optimization that maximizes $N_{Div}$ and minimizes $CV_{Birth}$. The initial model topology was the quantity sensing oscillator of *Appendix 1—figure 3A* which was optimized over 4000 epochs.

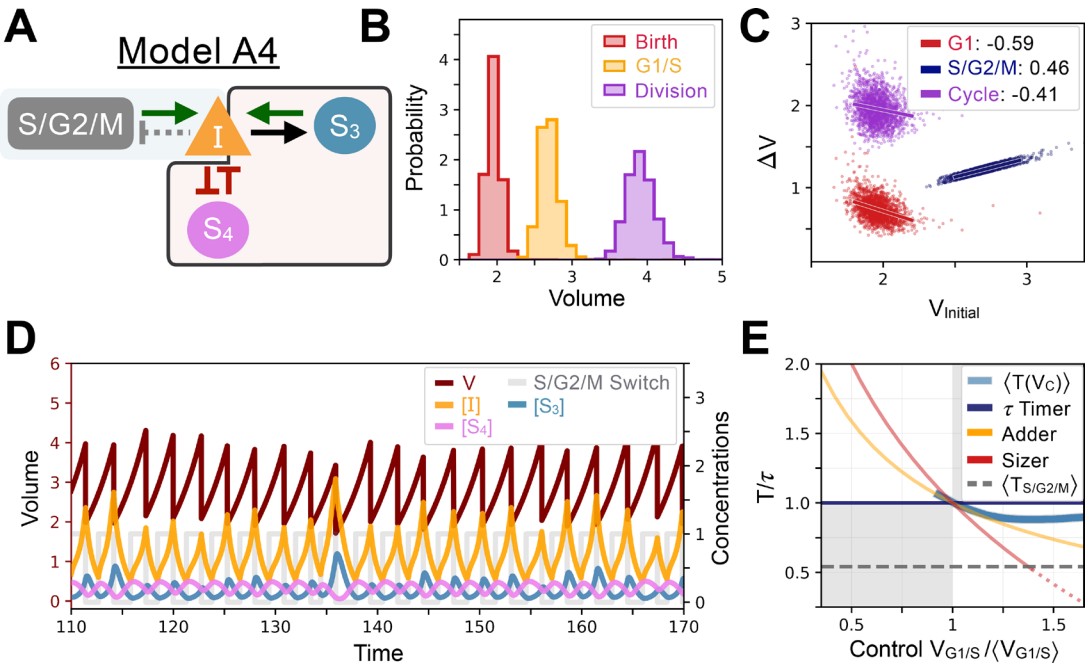

**Appendix 1—figure 9.** Model A4's behavior. (**A**) Network topology of the evolved Model A4. $S_4$ is the size sensor and represses the production of $I$ in a size-dependent manner. (**B**) Size distributions at birth (red), G1/S (orange) and division (purple). (**C**) Added volumes in G1 (red), S/G2/M (blue) and over the whole cycle (purple) as a function of initial volume at the start of those phases. (**D**) Temporal dynamics of the model, colors correspond to the variables in A. (**E**) Response curve $T(V_C)$ as a function of control volume $V_C$ at the G1/S transition.

**Appendix 1—table 6.** Model A4 parameter values.

| Parameter | Value | Parameter | Value |
|---|---|---|---|
| $p_2$ | 5.6604334 | $\delta_2$ | 0.001059 |
| $t_{1:2}$ | 0.408208 | $k_f$ | 0.843408 |
| $n_{1:2}$ | 1.769233 | $k_b$ | 1.680321 |
| $t_{3:2}$ | 0.887392 | $\delta_3$ | 0.825150 |
| $n_{3:2}$ | 6.437683 | $p_4$ | 4.674459 |
| $t_{4:2}$ | 0.197495 | $t_{2:4}$ | 0.744119 |
| $n_{4:2}$ | 3.238633 | $n_{2:4}$ | 3.451469 |
| $b_2$ | 0.800255 | $\delta_4$ | 1.929584 |

$$\frac{d[\mathbf{I}]}{dt} = \max\left(p_2 \cdot \max\left(\frac{(\text{S/G2/M Switch}/t_{1:2})^{n_{1:2}}}{1+(\text{S/G2/M Switch}/t_{1:2})^{n_{1:2}}}, \frac{([\mathbf{S}_3]/t_{3:2})^{n_{3:2}}}{1+([\mathbf{S}_3]/t_{3:2})^{n_{3:2}}}\right), b_2\right)$$
$$\cdot \frac{1}{1+(\mathbf{S}_4/t_{4:2})^{n_{4:2}}} - k_f[\mathbf{I}]^2 + k_b[\mathbf{S}_3] - (\delta_2 + \lambda)[\mathbf{I}]$$
$$\frac{d[\mathbf{S}_3]}{dt} = k_f[\mathbf{I}]^2 - k_b[\mathbf{S}_3] - (\delta_3 + \lambda)[\mathbf{S}_3] \tag{28}$$
$$\frac{d[\mathbf{S}_4]}{dt} = \frac{p_4}{1+(\mathbf{I}/t_{2:4})^{n_{2:4}}} - (\delta_4 + \lambda)[\mathbf{S}_4]$$

## Model A5

Model A5 is another variation on Model A1 but here within the *S. pombe* cell cycle framework where *I* controls the timing of division directly and the Switch is turned on in G1 instead of S/G2/M. Here, $\mathbf{S}_3$ directly senses size and does so via the PPI linking I, $\mathbf{S}_3$, and $\mathbf{S}_7$. Indeed, since I is inversely proportional to the volume at G1/S due to quantity sensing and since $\mathbf{S}_7$ is solely produced via complex formation of I with $\mathbf{S}_3$, $\mathbf{S}_7$ is also inversely proportional to volume. Consequently, $\mathbf{S}_3$, which is almost solely produced via the dissociation of $\mathbf{S}_7$, becomes a direct sensor of the size of the cell.

Then, instead of using a PPI to titrate *I* in a size-dependent manner as is done in Models A1 and A2, Model A5 leverages a transcriptional repression mechanism to modulate the production of *I* directly instead of its degradation, precisely like in Model A4. The model's behavior is summarized in *Appendix 1—figure 10*. The parameter values of the model are shown in *Appendix 1—table 7* along with the corresponding differential equations in *Equation 29*.

This model was evolved using a Pareto fitness optimization that maximizes $N_{\text{Div}}$ and minimizes $CV_{\text{Birth}}$. The seed model topology was the quantity sensing oscillator of *Appendix 1—figure 3A* which was optimized over 2500 epochs.

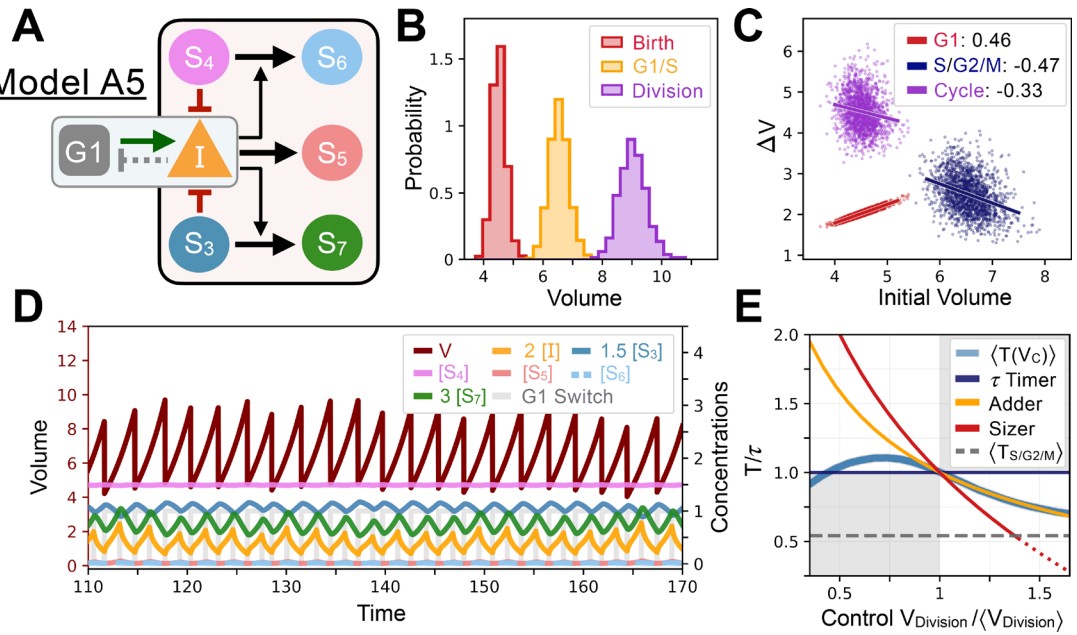

**Appendix 1—figure 10.** Model A5's behavior. (**A**) Network topology of the evolved Model A5. $S_3$ is the size sensor and represses the production of *I* in a size-dependent manner. (**B**) Size distributions at birth (red), G1/S (orange) and division (purple). (**C**) Added volumes in G1 (red), S/G2/M (blue) and over the whole cycle (purple) as a function of initial volume at the start of those phases. (**D**) Temporal dynamics of the model, colors correspond to the variables in A. (**E**) Response curve $T(V_C)$ as a function of control volume $V_C$ at division.

**Appendix 1—table 7.** Model A5 parameter values.

| Parameter | Value | Parameter | Value |
|---|---|---|---|
| $p_2$ | 2.379635 | $b_4$ | 2.624939 |
| $t_{1:2}$ | 0.142770 | $\delta_4$ | 1.499653 |
| $n_{1:2}$ | 0.383384 | $\delta_5$ | 0.006318 |
| $t_{3:2}$ | 0.714386 | $\delta_6$ | 0.983140 |
| $n_{3:2}$ | 8.642875 | $\delta_7$ | 0.481592 |
| $t_{4:2}$ | 1.754776 | $k_{f1}$ | 0.226150 |
| $n_{4:2}$ | 7.783080 | $k_{b1}$ | 3.793388 |
| $b_2$ | 0.195855 | $k_{f2}$ | 1.608325 |
| $\delta_2$ | 0.576282 | $k_{b2}$ | 3.322565 |
| $b_3$ | 0.582449 | $k_{f3}$ | 2.999118 |
| $\delta_3$ | 0.318466 | $k_{b3}$ | 0.906997 |

$$
\begin{aligned}
\frac{d[\mathbf{I}]}{dt} &= p_2 \cdot \max\left( \frac{(\mathbf{G1\ Switch}/t_{1:2})^{n_{1:2}}}{1+(\mathbf{G1\ Switch}/t_{1:2})^{n_{1:2}}}, b_2 \right) \cdot \frac{1}{1+(\mathbf{S}_3/t_{3:2})^{n_{3:2}}} \cdot \frac{1}{1+(\mathbf{S}_4/t_{4:2})^{n_{4:2}}} \\
&\quad - k_{f1}[\mathbf{I}] \cdot [\mathbf{S}_4] + k_{b1}[\mathbf{S}_6] - k_{f2}[\mathbf{I}]^2 + k_{b2}[\mathbf{S}_5] - k_{f3}[\mathbf{I}] \cdot [\mathbf{S}_3] + k_{b3}[\mathbf{S}_7] \\
&\quad - (\delta_2 + \lambda)[\mathbf{I}] \\
\frac{d[\mathbf{S}_3]}{dt} &= b_3 - k_{f3}[\mathbf{I}] \cdot [\mathbf{S}_3] + k_{b3}[\mathbf{S}_7] - (\delta_3 + \lambda)[\mathbf{S}_3] \\
\frac{d[\mathbf{S}_4]}{dt} &= b_4 - k_{f1}[\mathbf{I}] \cdot [\mathbf{S}_4] + k_{b1}[\mathbf{S}_6] - (\delta_4 + \lambda)[\mathbf{S}_4] \\
\frac{d[\mathbf{S}_5]}{dt} &= k_{f2}[\mathbf{I}]^2 - k_{b2}[\mathbf{S}_5] - (\delta_5 + \lambda)[\mathbf{S}_5] \\
\frac{d[\mathbf{S}_6]}{dt} &= k_{f1}[\mathbf{I}] \cdot [\mathbf{S}_4] - k_{b1}[\mathbf{S}_6] - (\delta_6 + \lambda)[\mathbf{S}_6] \\
\frac{d[\mathbf{S}_7]}{dt} &= k_{f3}[\mathbf{I}] \cdot [\mathbf{S}_3] - k_{b3}[\mathbf{S}_7] - (\delta_7 + \lambda)[\mathbf{S}_7]
\end{aligned}
\tag{29}
$$

## Model A6

Model A6 is yet another version of Model A1 evolved with slightly different fitness functions. This model was evolved using a Pareto fitness optimization that maximizes $N_{Div}$ and minimizes the sum of squared residuals from a target volume at birth as described in *Equation 15*. For reference, the target volume at birth chosen for this simulation was $V_t = 10$. The initial model topology was the quantity sensing oscillator of *Appendix 1—figure 3A* which was optimized over 3000 epochs.

The model's behavior is summarized in *Appendix 1—figure 11*. The parameter values of the model are shown in *Appendix 1—table 8* along with the corresponding differential equations in *Equation 30*.

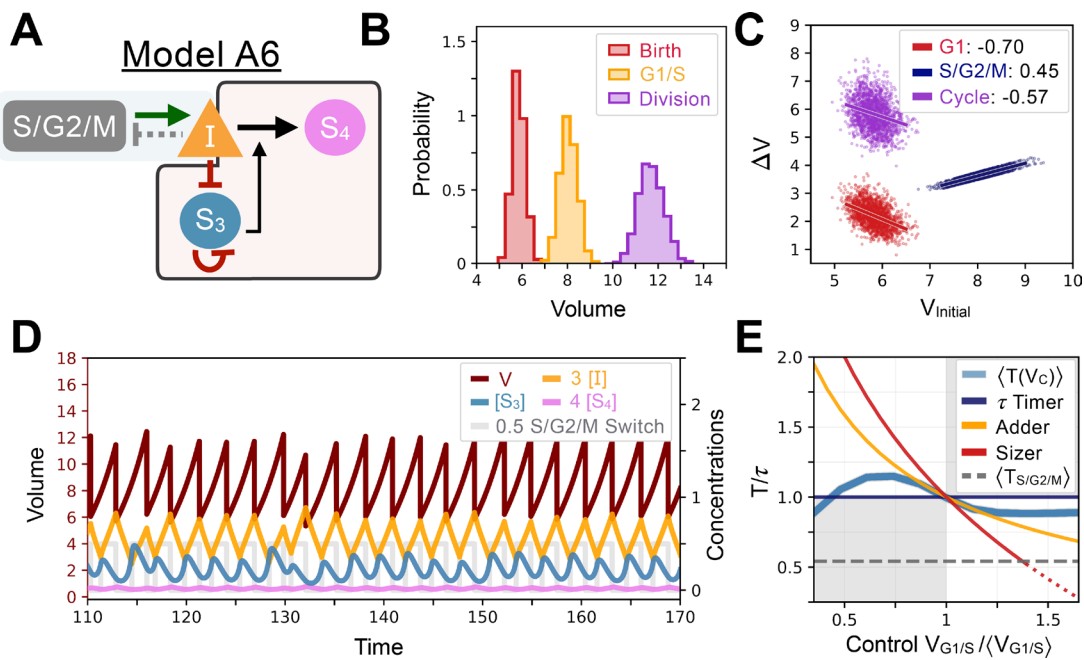

**Appendix 1—figure 11.** Model A6's behavior. (**A**) Network topology of the evolved Model A6. $S_3$ is the size sensor and represses the production of $I$ in a size-dependent manner. (**B**) Size distributions at birth (red), G1/S (orange) and division (purple). (**C**) Added volumes in G1 (red), S/G2/M (blue) and over the whole cycle (purple) as a function of initial volume at the start of those phases. (**D**) Temporal dynamics of the model, colors correspond to the variables in A. (**E**) Response curve $T(V_C)$ as a function of control volume $V_C$ at the G1/S transition.

**Appendix 1—table 8.** Model A6 parameter values.

| Parameter | Value | Parameter | Value |
|---|---|---|---|
| $p_2$ | 0.369408 | $\delta_2$ | 0.093159 |
| $t_{1:2}$ | 0.748731 | $k_{f1}$ | 0.533415 |
| $n_{1:2}$ | 3.850889 | $k_{b1}$ | 3.141514 |
| $p_3$ | 3.868044 | $\delta_3$ | 0.626507 |
| $t_{2:3}$ | 0.082081 | $k_{f2}$ | 3.911233 |
| $n_{2:3}$ | 3.638939 | $k_{b2}$ | 1.885120 |
| $t_{3:3}$ | 0.599375 | $\delta_4$ | 0.821609 |
| $n_{3:3}$ | 6.300643 | $\delta_5$ | 1.882484 |

$$
\begin{aligned}
\frac{d[\mathbf{I}]}{dt} &= p_2 \frac{(\mathbf{S/G2/M\ Switch}/t_{1:2})^{n_{1:2}}}{1+(\mathbf{S/G2/M\ Switch}/t_{1:2})^{n_{1:2}}} - k_{f1}[\mathbf{I}]^2 + k_{b1}[\mathbf{S_4}] - k_{f2}[\mathbf{I}] \cdot [\mathbf{S_3}] + k_{b2}[\mathbf{S_5}] \\
&\quad - (\delta_2 + \lambda)[\mathbf{I}] \\
\frac{d[\mathbf{S_3}]}{dt} &= p_3 \frac{1}{1+(([\mathbf{I}]/t_{2:3})^{n_{2:3}})} \frac{1}{1+(([\mathbf{S_3}]/t_{3:3})^{n_{3:3}})} - k_{f2}[\mathbf{I}] \cdot [\mathbf{S_3}] + k_{b2}[\mathbf{S_5}] - (\delta_3 + \lambda)[\mathbf{S_3}] \\
\frac{d[\mathbf{S_4}]}{dt} &= k_{f1}[\mathbf{I}]^2 - k_{b1}[\mathbf{S_4}] - (\delta_4 + \lambda)[\mathbf{S_4}] \\
\frac{d[\mathbf{S_5}]}{dt} &= k_{f2}[\mathbf{I}] \cdot [\mathbf{S_3}] - k_{b2}[\mathbf{S_5}] - (\delta_5 + \lambda)[\mathbf{S_5}]
\end{aligned}
\tag{30}
$$

