## [Editor Report]

This paper develops evolutionary simulations to identify the type of molecular networks that can give rise to size control. The authors propose an evolutionary framework to find which factors select for particular mechanisms in cell size control. They show that the evolution of a specific cell size control mechanism is dependent on the cell cycle structure.

---

## [Decision Letter]

**Decision letter after peer review:**

Thank you for submitting your article "Evolution of cell size control is canalized towards adders or sizers by cell cycle structure and selective pressures" for consideration by *eLife*. Your article has been reviewed by 3 peer reviewers, and the evaluation has been overseen by a Reviewing Editor and Aleksandra Walczak as the Senior Editor. The following individuals involved in the review of your submission have agreed to reveal their identity: Kabir Husain (Reviewer #1); Shiladitya Banerjee (Reviewer #2).

Essential revisions:

The reviewers liked many aspects of the manuscript but have suggested a number of revisions, which include showing the robustness of the results, a better justification of the assumptions/methods, a comparison with existing data and mathematical approaches, including clarification of how the evolutionary approach differs from other approaches, and a more careful rewording of the conclusions. Please address all of the reviewer comments (see below).

*Reviewer #1 (Recommendations for the authors):*

Figures and text:

I think the paper would be greatly strengthened by less dense Figures. As it stands, I found it difficult to figure out what the main points are. As an example: the main point of Figure 5, as I understand it, might be best served by a time-series plot of Slope \Δ-V_{cycle}, or some other summary statistic that shows the transient evolution of a sizer. Otherwise, this point is buried in the legend of panels B, E, and H.

On feedback control mechanisms:

There are no statistical summaries of the simulations described in Figures 2 and 3 of the main text -- Only Figure 4 (whose simulations are initialised with the evolved Model A1) contains statistics on evolved networks. If I understand the text, all the simulations resulted in topologically similar networks -- is this the case. What is the range of CV_births, and does the achieved CV_birth depend on the molecular implementation of the feedback control (PPI, dimerisation, transcriptional control), or do these molecular details affect the \Δ-V_{slope}?

On adders vs sizers (Figures4, 5, and lines 557 to 561 in the discussion):

Figure 4, 5, and the text around it, suggest that the CV_{birth} for adders is lower than that of sizers, in contrast to the common view that sizers are better at controlling cell size. I wonder if the distinction is between the *steady-state CV_{birth}* and the time taken to return to equilibrium from a large 'perturbation' of the cell size?

If it is true that adders are better at the former, but sizers are better at the latter, then would this also explain why cell size control late in the cycle tends to favour sizers? The intuition perhaps being that size control later in the cycle needs to deal with perturbations that occurred earlier in the cell cycle (as suggested by Lines 557 to 561)?

*Reviewer #2 (Recommendations for the authors):*

We have the following specific comments and recommendations for the authors.

1. Lines 74-77: This is the one piece of the introduction where readers unfamiliar with the eukaryotic cell cycle would be confused. Including background information about G1/S and S/G2/M phases would help expand the target audience of the paper since the techniques discussed therein are widely applicable.

2. Lines 101-106: The "poisson rate that corresponds to the deterministic rate" is unclear. These two sentences could be elaborated on further since significant prior knowledge of the reader is currently assumed.

3. Line 115: "While size-dependent growth mechanisms exist and do support size homeostasis" – This assertion should be backed up with relevant citations.

4. Lines 209-210: "Upon passing the G1/S transition, we assume cells are committed to division and there is a fixed time delay before they divide thus modeling S/G2/M as a timer." – What motivates this assumption? A citation or further discussion is warranted.

5. Line 285: This would be a good place for a citation to direct readers to sources discussing concentration vs quantity sensing. Note that in the bacterial size control literature, quantity sensing of division initiators has been shown to regulate adder behavior (Si et al., Curr Biol 2019).

6. Figure 1B: The dashed grey line is defined afterwards in 1C. It would be good to include it in the defined interactions here instead. In addition, the clarity of this schematic would be better with a line showing how the final network becomes the initial network in the next epoch.

7. Lines 301-302: "… its production is completely shut down in S/G2/M" does not come through clearly in the associated figure. You should clearly describe the chosen dynamics for the inhibitor protein in different phases of the cell cycle, and justify why the choices are different from known inhibitors such as Whi5.

8. Figure 2A: The message of this figure is not presented clearly. There is clustering with high CV volume and low N, another with negligible CV volume and widespread N division, and then the circled optimum. However, the trajectory of how a network evolves is not clear in this picture. Do all of them converge to the optimum eventually? Do they move to low CV before high N division or at the same time? How many epochs does it take to cross the large gap between the clustered networks and the optimum? Recommend somehow indicating sample evolutionary trajectories in addition to the aforementioned clarifications to remedy this issue. Additionally, why are there no numerical values in the axes? This makes it very difficult to assess the degree to which original values have changed.

9. Lines 309-322: This paragraph is somewhat confusing, in particular lines 314-316. The motivation for the control volume is unclear, especially in the physical sense of why a cell would use a non-physical volume to control a transition. While the idea makes sense later in the supplementary material, it needs to be clear from the very start that the goal here is that the control volume is a tool to examine how size at G1/S affects the cycle time.

10. Figures 3A,3C: While intuitive, specifying the role of the dashed red line would improve clarity.

11. Figure 3D: The predictions for the cell cycle scatter appear much stronger than the scatter itself. Can you comment on this?

12. Figure 3B, D: Compare how the model predictions compare with binned means for the scatter.

13. Lines 357-361: The numbers of 120 simulations and 500 epochs appear to be chosen arbitrarily. Why did you choose these initial conditions, and are the results of your paper robust with respect to higher/lower values? If so, including that point here would strengthen the argument, especially with a brief discussion on the lower limit. In addition, roughly how long in time is an epoch? The speed at which evolution is occurring would be of interest to many readers.

14. Figure 4: 4B is created to resemble *S. pombe*. Do the other panels have real-life analogies or are they arbitrarily chosen for qualitative representations of the discussed effects in the main text?

15. Figures 2F, 2I, 5A, 5D, 5G, 6C: The second zoomed-in panel of 6C is essential to understand the inner-generational dynamics of your modeling. The first many-generation panel shows stability in V but fails to address the other variables and the multi-phase dynamics. The other figures (2F, 2I, 5A, 5D, 5G) would benefit greatly from either a similar treatment or just fewer generations. The stability can be shown with significantly fewer divisions than are currently used.

16. Figure 5B: The ΔV scatters for S/G2/M and the whole cycle could be grouped into two – a positive correlation and a negative correlation. A best fit to the entire scatter is misleading therefore and does not describe the correlation trend.

17. Lines 539-542: Why is a one-step implementation of size control discarded? Surely a simpler control mechanism could be preferred naturally despite being a lesser theoretical interest in evolution simulations.

18. Lines 793-794: In this paper, you consider parallels to organisms that divide asymmetrically, such as budding yeast. Have you run simulations considering asymmetric division? Surely that would impact cell size distribution and variability.

19. Lines 794-795: Wouldn't disregarding one of the two daughter cells add a bias against faster dividing cells? I.e. if the number of divisions is one of your fitness functions, doesn't this method eliminate the natural advantage of a relatively larger population size for multi-cell level exponential growth? Also an issue at S130-132.

20. Lines 801-802: How is the extraction of the nullcline performed?

21. 829: Recommend providing the conversion from the arbitrary units used to physical values, here and all other figures.

22. 838-839: Model A2 does not receive an explanation comparable to A1 in this figure; either move to supplemental materials or explain it clearly as well.

23. Figures 6H, 6I, 6J: These subfigures are not very clear, together with captions for 6I and 6J that do not sufficiently explain to the reader how they are read. Why is the Burst Amplitude axis extended so far beyond the heatmap?

24. S120-S121: How do these theta and n theta values come to be? Currently, it seems like they are chosen with no supporting reasoning or explanation.

25. S239: V target missing a capital T.

26. S328-S329: Need to use a left apostrophe rather than two right apostrophes for epoch and generation.

27. Eq. (S18): Why do you minimize m+1 rather than just m?

28. S406-S408: Why do you choose these interactions to include? How much of all biochemical interactions do they encompass together? Are there others that you are aware of that you are choosing to neglect, and why? Can you provide citations and/or an argument to motivate this choice? This is another crucial ansatz for your modeling that needs to be discussed more carefully.

29. Supplementary Section 4: Translating the arbitrary units into physical values (when possible) would be immensely useful/helpful here.

30. Figure S6E: Why does cut off just below 1 (here, and not in any other plots)?

31. Since the model is generally applicable to any organism, comparisons to size control in bacterial cells (even qualitative) would be useful to widen the appeal. For example, could you predict why almost all bacterial cells (even evolutionary divergent ones) behave as adders? It has been shown that adder is regulated by threshold accumulation of an initiator protein that is produced at a rate proportional to cell volume, which your model could perhaps capture. Furthermore, many bacterial cells also exhibit biphasic size regulation during the cell cycle. It has been shown that *Bacillus subtilis* behave as sizers during the first phase, followed by a timer phase till division (DOI:10.1016/j.cub.2020.04.030). By contrast, Caulobacter crescentus cells implement a timer first, followed by an adder phase of size control (DOI:10.1038/nmicrobiol.2017.116). Both these organisms behave as approximate adders overall.

---

## [Author Response]

Reviewer #1 (Recommendations for the authors):Figures and text:I think the paper would be greatly strengthened by less dense Figures. As it stands, I found it difficult to figure out what the main points are. As an example: the main point of Figure 5, as I understand it, might be best served by a time-series plot of Slope \Δ-V_{cycle}, or some other summary statistic that shows the transient evolution of a sizer. Otherwise, this point is buried in the legend of panels B, E, and H.

We have clarified the main text and the captions to better explain Figure 5 but otherwise have kept the figure mostly unchanged as we find useful to see the actual temporal dynamics of the networks throughout evolution to illustrate the sloppy sizer evolving into a weak adder in order to reduce the CV of the size distribution at birth. This key point has now been added to the discussion as described above. To re-emphasize its importance, and clarify the purpose of Figure 5, we have modified the top of the figure caption, which now includes the following sentence: “Evolutionary dynamics continually reduce the selected for CVextBirth and proceed through a noisy sizer to a less noisy adder”

On feedback control mechanisms:There are no statistical summaries of the simulations described in Figures 2 and 3 of the main text -- Only Figure 4 (whose simulations are initialised with the evolved Model A1) contains statistics on evolved networks. If I understand the text, all the simulations resulted in topologically similar networks -- is this the case.

It is the case as all successful evolution simulations ended with some version of Model A. In Figure 2, we present two versions of Model A that have slightly different network topologies but perform the feedback mechanism in the same way. In the Supplement, we also give additional examples of evolved models in Figures S8, S9 and S10. We rewrote the second paragraph of the Results section, which now begins as: “Evolution simulations are in part reproducible and most often lead to similar network topologies. The evolution trajectory leading to Model A1 is a typical example in which all simulations produced similar networks (Figure 2B).”

What is the range of CV_births, and does the achieved CV_birth depend on the molecular implementation of the feedback control (PPI, dimerisation, transcriptional control), or do these molecular details affect the \Δ-V_{slope}?

We have now included a Table in the Supplement where we show the ranges of CV_births obtained by our evolved models (see new Table S1). We’ve seen different combinations of biochemical interactions evolve and perform feedback in different ways. Yet, they generally result in similar CV_Birth as seen in the 5 different versions of Model A shown in Figures 2, 3, S8, S9, and S10. Our understanding is that the resulting \Δ V_Slope is a direct consequence of the strength of the feedback mechanism which is in itself dictated by the molecular interactions of the network. But, the feedback control can be implemented in different ways with similar results. We now describe these results in the last paragraph of the section ‘Evolution of quantity-based size control mechanisms’ which reads as: “We give additional examples of similarly evolved networks in Figures S8-S10 where we can see the sensing and the feedback mechanism being implemented in different ways. Yet, despite these mechanistic differences in feedback regulation the resulting function of the evolved networks were similar as indicated by their CV_Birth_ (Table S1).”

On adders vs sizers (Figures4, 5, and lines 557 to 561 in the discussion):Figure 4, 5, and the text around it, suggest that the CV_{birth} for adders is lower than that of sizers, in contrast to the common view that sizers are better at controlling cell size. I wonder if the distinction is between the *steady-state CV_{birth}* and the time taken to return to equilibrium from a large 'perturbation' of the cell size?If it is true that adders are better at the former, but sizers are better at the latter, then would this also explain why cell size control late in the cycle tends to favour sizers? The intuition perhaps being that size control later in the cycle needs to deal with perturbations that occurred earlier in the cell cycle (as suggested by Lines 557 to 561)?

As we discussed above, a noisy sizer can have a higher CV at steady state than an accurate adder. However, in line with the reviewers intuition, expect sizers to be better at reducing the effect of large fluctuations where the system is far from steady state. This is because a sizer will return the system to the steady state in one cell cycle. We now emphasize this point in the discussion, where we write:

‘However, we anticipate even noisy sizers will be better than adders at controlling cell size in response to large deviations away from the steady state distribution. This is because sizers will always return the cell size to be within the steady state distribution within a cell cycle.’

The argument here is agnostic to the distribution of cell size control in the different phases of the cell cycle and independent of the fact that G2 size control promotes sizers (which is discussed at length in the text already).

Reviewer #2 (Recommendations for the authors):We have the following specific comments and recommendations for the authors.1. Lines 74-77: This is the one piece of the introduction where readers unfamiliar with the eukaryotic cell cycle would be confused. Including background information about G1/S and S/G2/M phases would help expand the target audience of the paper since the techniques discussed therein are widely applicable.

We have added some basic information on the cell cycle and cited the main book of the field by David Morgan. The introduction now contains the following lines of text: “Cell size is regulated through the cell cycle control network that governs transitions from one phase of the cell cycle to the next. The division cycle can be broken up into distinct phases that are characterized by different molecular activities (D. O. Morgan, 2007). While it is typically considered that there are 4 phases of the cell cycle (G1, S, G2, and M), we here consider a two phase model based on a G1 phase and a composite S/G2/M phase. This is because size control in general has been associated with either the G1/S transition or mitosis at the end of the cell cycle.”

2. Lines 101-106: The "poisson rate that corresponds to the deterministic rate" is unclear. These two sentences could be elaborated on further since significant prior knowledge of the reader is currently assumed.

We now write: “Thus, each biochemical reaction takes place with a rate that corresponds to the deterministic rate, to which we add one white Gaussian noise with a variance equal to that rate. For example, given a deterministic biochemical rate k and a time interval of size extΔt, we consider a tau-leaping change of kΔt+N(0,kΔt) where N(0,kΔt) is a random gaussian variable of mean 0 and variance extkΔt.”

3. Line 115: "While size-dependent growth mechanisms exist and do support size homeostasis" – This assertion should be backed up with relevant citations.

We now cite Miettinen and Bjorklund (2016; https://doi.org/10.1016/j.devcel.2016.09.004) and Ginzburg et al. (2018; https://doi.org/10.7554/*eLife*.26957).

4. Lines 209-210: "Upon passing the G1/S transition, we assume cells are committed to division and there is a fixed time delay before they divide thus modeling S/G2/M as a timer." – What motivates this assumption? A citation or further discussion is warranted.

This assumption is justified by what happens in *S. cerevisiae*’s cell cycle. Upon passing the G1/S transition, during an event called Start, the cell becomes irreversibly committed to division. This means that at that point, even when treated with drugs that would disable the late cell cycle machinery, cells will divide no matter what. We now cite Doncic et al. (2011) as a reference for the commitment point, and Chandler-Brown et al. (2017) as a reference for the time nature of S/G2/M phase.

5. Line 285: This would be a good place for a citation to direct readers to sources discussing concentration vs quantity sensing. Note that in the bacterial size control literature, quantity sensing of division initiators has been shown to regulate adder behavior (Si et al., Curr Biol 2019).

As suggested, we now refer the reader to references discussing mechanisms to sense protein quantities rather than concentrations. We now discuss quantity sensing when it first is introduced in the methods section. The subsection on initial cell cycle model now contains the following sentences: “One way the amount rather than the concentration of a molecule could be sensed is through its titration against a fixed cellular quantity such as the genome, which is part of a general class of titration-based cell size sensing mechanisms (Amodeo et al., 2015; Heldt et al., 2018; Si et al., 2019; Wang et al., 2009).”

6. Figure 1B: The dashed grey line is defined afterwards in 1C. It would be good to include it in the defined interactions here instead. In addition, the clarity of this schematic would be better with a line showing how the final network becomes the initial network in the next epoch.

The dashed grey line is a special interaction that cannot be evolved or mutated by the PhiEvo algorithm. This grey dashed line always connects the inhibitor I to the S/G2/M Switch to indicate that I is in fact an inhibitor of the S/G2/M cell cycle phase. Because of this distinction, we want to separate this interaction from the evolvable interactions (transcriptional activation, transcriptional repression and complexation) that are shown in Figure 1B. To avoid confusion, we have removed the dashed grey line from the cartoon networks of Figure 1B as the focus of this panel is on the general evolution algorithm itself and not the specific implementation used in this study.

As for a line showing how the final network becomes the initial network in the next epoch, we have included a figure in the Supplement showing this process in more detail (Figure S4).

7. Lines 301-302: "… its production is completely shut down in S/G2/M" does not come through clearly in the associated figure. You should clearly describe the chosen dynamics for the inhibitor protein in different phases of the cell cycle, and justify why the choices are different from known inhibitors such as Whi5.

Here in these lines, we were talking about the R protein and not the I inhibitor that plays the role of Whi5 in our system. We apologize for this confusion and have clarified this point in the text, which now reads as: “For example, Model A2 contains extra interactions for the volume sensing gene [R], where [R] is repressed by the S/G2/M Switch (meaning its production is completely shut down in S/G2/M leading to sawtooth-like dynamics).”

8. Figure 2A: The message of this figure is not presented clearly. There is clustering with high CV volume and low N, another with negligible CV volume and widespread N division, and then the circled optimum. However, the trajectory of how a network evolves is not clear in this picture. Do all of them converge to the optimum eventually? Do they move to low CV before high N division or at the same time? How many epochs does it take to cross the large gap between the clustered networks and the optimum? Recommend somehow indicating sample evolutionary trajectories in addition to the aforementioned clarifications to remedy this issue. Additionally, why are there no numerical values in the axes? This makes it very difficult to assess the degree to which original values have changed.

We have completely remade this figure panel to render it more transparent (shown below) and have updated the caption accordingly. We have also added additional details about the evolutionary trajectory in the subsection S3A – Fitness of the Supplement.

Evolution happens in several stages. First, there are several epochs without any size control; networks then cluster in two regions of the Pareto front, essentially corresponding to volume going to 0 ([ii] in the new panel) and volume going to maximum volume ([i] in the new panel) both cases which are highly penalized in their fitness score (as is now explained in the Supplement). In Figure S3A, we show that the initial cell cycle network for the evolution process leads to unstable growth which is why we inevitably see the cell volume crash to 0 or maximum volume and cluster in [i] or [ii]. Evolution goes back and forth between those two clusters with a slow increase of the number of divisions (the fitness on the x axis). Eventually, some volume control evolves, most of the time corresponding to the Model 1 architecture described in the main text. Then, both the number of divisions N_div and the CV_Birth of those networks are optimized considerably, which is what we described in the main text as an “all or nothing” fitness score. Finally, CV_Birth keeps decreasing slowly until it reaches a plateau of 6-9% at the last generation. Not all networks go to optimum: in fact our Pareto evolution favours population diversity so makes sure that some networks are always relatively far from optimum. Also, notice that all simulations are different: those are stochastic simulations. Some simulations converge while others get lost on their way to the optimum as is expected. However, we implement the equivalent of a Drake’s rule: on average each network is mutated once per epoch. So the number of epochs before an evolutionary jump is a proxy of the typical number of mutations needed to select for one “good” mutation (changing Pareto front). We have added more details on all of this.

9. Lines 309-322: This paragraph is somewhat confusing, in particular lines 314-316. The motivation for the control volume is unclear, especially in the physical sense of why a cell would use a non-physical volume to control a transition. While the idea makes sense later in the supplementary material, it needs to be clear from the very start that the goal here is that the control volume is a tool to examine how size at G1/S affects the cycle time.

The reviewer is absolutely correct and phrases this statement better than we did. We have now redone Figure 3 rewritten this part of the text, which now reads as: “We then numerically integrate the differential equations of the model and measure the period T(VC) of the simulated cell-cycle for this control volume. Use of the control volume allows us to break the size feedback system and distinguish its input, V_C_, from its output, the induced cycle period T (Angeli et al., 2004).”

10. Figures 3A,3C: While intuitive, specifying the role of the dashed red line would improve clarity.

We clarified this and have updated the figure accordingly.

11. Figure 3D: The predictions for the cell cycle scatter appear much stronger than the scatter itself. Can you comment on this?

We think there is some confusion here. We note that we are predicting the average response using the control volume framework described above. We do not predict the scatter, which results from the stochastic simulations in steady state. We have clarified this in the caption of Figure 3B which now reads: “Added volumes extΔV for different phases of the cell cycles for simulations of Model A1. Individual dots correspond to different cell cycles for a simulation at steady-state. The full line corresponds to the extrapolation from the T(VC) curve shown in A for a restricted range of VC relevant to the scatter. The black cross, star and square indicate the average added volumes corresponding to when the system senses a volume corresponding to ⟨VG1/Sangle at the G1/S transition. We see that the model is predicted to follow an adder over a large range of volumes.”

12. Figure 3B, D: Compare how the model predictions compare with binned means for the scatter.

In evolutionary simulations, fitness typically improves rapidly in the beginning but then plateaus and only very gradually increases after a couple of hundred epochs. This is a general property of optimization simulations, as is also generally seen in machine learning. Based on our extensive previous experience with PhiEvo simulations, 500 epochs typically is sufficient to find the fitness plateau, but not so much that the network extensively explores the neutral mutations around the plateau. An epoch has no length per se, it just is a cycle of mutation/selection, however, the evolutionary algorithm adjust its mutation rates to have on average 1 mutation per generation (this is in fact an experimental fact called Drake’s law, and practically it prevents the so called “code-bloat” that can be observed in evolutionary simulations). This means that if a network evolves within, say, 100 generations, it is at most 100 mutations away from the initial state, and practically much less than that because most mutations are either neutral or deleterious (and in that case are not kept). To clarify this point in the text, we now write: “We chose to use 500 epochs in our simulations because in our previous experience this was sufficient for networks to evolve to be near the optimum, but not so much that they were forced to extensively explore the effects of neutral mutations near the optimum.”

13. Lines 357-361: The numbers of 120 simulations and 500 epochs appear to be chosen arbitrarily. Why did you choose these initial conditions, and are the results of your paper robust with respect to higher/lower values? If so, including that point here would strengthen the argument, especially with a brief discussion on the lower limit. In addition, roughly how long in time is an epoch? The speed at which evolution is occurring would be of interest to many readers.

For clarification, as suggested by the reviewer, we have redone the figure panels on the dynamics with fewer generations to allow for better visualization of the multi-generational protein and volume dynamics.

14. Figure 4: 4B is created to resemble *S. pombe.* Do the other panels have real-life analogies or are they arbitrarily chosen for qualitative representations of the discussed effects in the main text?

Panel A was designed to resemble *S. cerevisiae*, at least qualitatively, which was discussed in the text as having a size control at the G1/S transition.

15. Figures 2F, 2I, 5A, 5D, 5G, 6C: The second zoomed-in panel of 6C is essential to understand the inner-generational dynamics of your modeling. The first many-generation panel shows stability in V but fails to address the other variables and the multi-phase dynamics. The other figures (2F, 2I, 5A, 5D, 5G) would benefit greatly from either a similar treatment or just fewer generations. The stability can be shown with significantly fewer divisions than are currently used.

For clarification, as suggested by the reviewer, we have redone the figure panels on the dynamics with fewer generations to allow for better visualization of the multi-generational protein and volume dynamics.

16. Figure 5B: The ΔV scatters for S/G2/M and the whole cycle could be grouped into two – a positive correlation and a negative correlation. A best fit to the entire scatter is misleading therefore and does not describe the correlation trend.

This is a good point, we have done this as there are indeed two distinct behaviors as pointed out by the referee depending on whether or not the cell is small or large. The figure panel now looks as follows and we have adjusted the caption accordingly:

17. Lines 539-542: Why is a one-step implementation of size control discarded? Surely a simpler control mechanism could be preferred naturally despite being a lesser theoretical interest in evolution simulations.

As explained above in response to the reviewers major comment #4, we chose to do the analysis the way we did because we want to explore how cell size control can be done by a network with multiple feedbacks rather than just the concentration of a single protein, such as the budding yeast Whi5 protein, that has a special dedicated synthesis mechanism to make its concentration directly reflect cell size.

18. Lines 793-794: In this paper, you consider parallels to organisms that divide asymmetrically, such as budding yeast. Have you run simulations considering asymmetric division? Surely that would impact cell size distribution and variability.

It is definitively of interest to explore the effects of asymmetric divisions, but this is outside the scope of this already dense manuscript and will be the subject of future investigations.

19. Lines 794-795: Wouldn't disregarding one of the two daughter cells add a bias against faster dividing cells? I.e. if the number of divisions is one of your fitness functions, doesn't this method eliminate the natural advantage of a relatively larger population size for multi-cell level exponential growth? Also an issue at S130-132.

Here, cell growth is exponential on the single cell level since we were assessing size control mechanisms that take size as an input to cell cycle control. We are not exploring the very interesting case where growth deviates from the exponential. In that case, what the reviewer says is absolutely essential because then size homoeostasis would have a contribution from some cells in the population outcompeting others in terms of their growth. We intend to explore this possibility in future work. We have now added this text to the supplementary material which reads: “Here, cell growth is exponential on the single cell level since we were assessing size control mechanisms that take size as an input to cell cycle control. We are not exploring the very interesting case where growth deviates from the exponential. In that case, size homoeostasis would have a contribution from some cells in the population outcompeting others in terms of their growth and we would have to simulate the entire cell population and not disregard one of the daughter cells as we do here.”

20. Lines 801-802: How is the extraction of the nullcline performed?

We clarify in the text how the fictitious nullcline is extracted. We write:

“An intermediate fictitious nullcline is shown as a line that connects the average concentration [I] at G1/S and at division.”

21. 829: Recommend providing the conversion from the arbitrary units used to physical values, here and all other figures.

Our arbitrary units are described in the methods where it states that: “Note that we scale all our variables so that a concentration of one arbitrary unit corresponds roughly to 1000 proteins in a 100fL cell (Milo et al., 2010). Additionally, we scale the time variable so that 1 arbitrary time unit corresponds roughly to 30 min (Di Talia et al., 2007).” However, we prefer to have the figures in AU since this leads to a numerically simpler presentation and corresponds directly to what we have evolved.

22. 838-839: Model A2 does not receive an explanation comparable to A1 in this figure; either move to supplemental materials or explain it clearly as well.

We think the shorter explanation of A2 is fine for the figure caption because there is a longer explanation in the main text to explain this model. We now refer the reader to see the text in the figure caption.

23. Figures 6H, 6I, 6J: These subfigures are not very clear, together with captions for 6I and 6J that do not sufficiently explain to the reader how they are read. Why is the Burst Amplitude axis extended so far beyond the heatmap?

To make these figure panels more clear, we truncated the Burst Amplitude axis as suggested by the reviewer, and moved the inset to be its own panel. We also modified the caption to reflect these changes.

24. S120-S121: How do these theta and n theta values come to be? Currently, it seems like they are chosen with no supporting reasoning or explanation.

This is an important point. We chose these values because they give a similar amount of noise in the G1/S transition as observed experimentally (*e.g.*, Di Talia et al. 2007; Chandler-Brown et al. 2017). We have added this explanation in the supporting information.

25. S239: V target missing a capital T.

This correction has been made.

26. S328-S329: Need to use a left apostrophe rather than two right apostrophes for epoch and generation.

This correction has been made.

27. Eq. (S18): Why do you minimize m+1 rather than just m?

In principle, it is the same. We chose to optimize m+1 rather than m in order to keep the fitness function positive. This helps when imposing multiplicative penalties on the fitness when cell volume is too low or too high and does not affect the optimization process.

28. S406-S408: Why do you choose these interactions to include? How much of all biochemical interactions do they encompass together? Are there others that you are aware of that you are choosing to neglect, and why? Can you provide citations and/or an argument to motivate this choice? This is another crucial ansatz for your modeling that needs to be discussed more carefully.

We chose these interactions because they are commonly used in the systems biology literature. Moreover, we have found in our previous evolutionary simulations that a combination of transcriptional interactions with protein-protein interactions is sufficient to account for many standard mechanisms in systems biology (e.g., switches, oscillators, biochemical adaptation). We note that other work in this area often restricts itself to purely transcriptional networks because they are easier to study and understand, and we added references to those works. Our approach is more flexible and generic.

We now write in the Supplement: “Many “inverse-approach” approaches in systems biology have focused on purely transcriptional networks (Francois et al., 2007, Fujimoto et al., 2008, Cotterell et al., 2010, Ten Tusscher et al., 2011), because they are generic, easier to study and can efficiently describe many biological dynamics (Alon, 2007).

In this project, we extend the biochemical interactions available for evolution: we not only model transcriptional activation and repression but also include complexation also known as protein-protein interaction (PPI), and assume there passive degradation. Adding PPIs is especially crucial because they are well known to lead to non-linear effects (Buchler et al., 2008, Buchler et al., 2009) allowing for the simple implementation of complex dynamics such as genetic oscillations (François et al., 2005) observed e.g. in circadian clocks (François, 2005), and such non-linear effects indeed play crucial roles for control in our evolved model. The equations for these interactions are presented in this subsection.”

29. Supplementary Section 4: Translating the arbitrary units into physical values (when possible) would be immensely useful/helpful here.

We have made the requested modification.

30. Figure S6E: Why does cut off just below 1 (here, and not in any other plots)?

For smaller Vc than the cutoff, Model A4 does not produce an oscillation, but instead reaches a steady state so there is no defined T/tau. We now note this fact in the new S9E (old S6E) caption and include a more detailed explanation for this model in the Supplement.

We write: “Notably, Figure S9E shows that the model's response curve in the low control volume regime is ill-defined. In this model specifically, when the low volume regime is reached, the concentration of inhibitor *I* at G1/S increases due to quantity sensing [I] ∝ 1/V. Then, because of the homodimerization of *I* into S_3_, we see the concentration [S_3_] also rise. We can see both of these curves spike up momentarily in the trajectories of Figure S9D around t ≈ 136. The problem arises if this increase is too strong. Then, S activates the production of additional *I*, kick-starting a positive feedback loop that creates more and more inhibitor *I,* effectively interrupting the oscillator and making the period ill-defined.”

31. Since the model is generally applicable to any organism, comparisons to size control in bacterial cells (even qualitative) would be useful to widen the appeal. For example, could you predict why almost all bacterial cells (even evolutionary divergent ones) behave as adders? It has been shown that adder is regulated by threshold accumulation of an initiator protein that is produced at a rate proportional to cell volume, which your model could perhaps capture. Furthermore, many bacterial cells also exhibit biphasic size regulation during the cell cycle. It has been shown that *Bacillus subtilis* behave as sizers during the first phase, followed by a timer phase till division (DOI:10.1016/j.cub.2020.04.030). By contrast, Caulobacter crescentus cells implement a timer first, followed by an adder phase of size control (DOI:10.1038/nmicrobiol.2017.116). Both these organisms behave as approximate adders overall.

We thank the reviewer for indicating those references, that we now cite in the manuscript. We have a generic argument for why adders might arise, ie, if size control takes place early in the division cycle, but do not have any specific things to say about bacteria in comparison to eukaryotic cells.